# A stochastic multicellular model identifies biological watermarks from disorders in self-organized patterns of phyllotaxis

Yassin Refahi[1,2], Géraldine Brunoud[1], Etienne Farcot[3,4], Alain Jean-Marie[5], Minna Pulkkinen[6], Teva Vernoux[1]*, Christophe Godin[7]*

[1]Laboratoire de Reproduction de développement des plantes, Lyon, France; [2]Sainsbury Laboratory, University of Cambridge, Cambridge, United Kingdom; [3]School of Mathematical Sciences, The University of Nottingham, Nottingham, United Kingdom; [4]Center for Integrative Plant Biology, The University of Nottingham, Notthingam, United Kingdom; [5]INRIA Project-Team Maestro, INRIA Sophia-Antipolis Méditerranée Research Center, Montpellier, France; [6]UMR Lerfob, AgroParisTech, Nancy, France; [7]INRIA Project-Team Virtual Plants, CIRAD, INRA and INRIA Sophia-Antipolis Méditerranée Research Center, Montpellier, France

**Abstract** Exploration of developmental mechanisms classically relies on analysis of pattern regularities. Whether disorders induced by biological noise may carry information on building principles of developmental systems is an important debated question. Here, we addressed theoretically this question using phyllotaxis, the geometric arrangement of plant aerial organs, as a model system. Phyllotaxis arises from reiterative organogenesis driven by lateral inhibitions at the shoot apex. Motivated by recurrent observations of disorders in phyllotaxis patterns, we revisited in depth the classical deterministic view of phyllotaxis. We developed a stochastic model of primordia initiation at the shoot apex, integrating locality and stochasticity in the patterning system. This stochastic model recapitulates phyllotactic patterns, both regular and irregular, and makes quantitative predictions on the nature of disorders arising from noise. We further show that disorders in phyllotaxis instruct us on the parameters governing phyllotaxis dynamics, thus that disorders can reveal biological watermarks of developmental systems.

*For correspondence: teva. vernoux@ens-lyon.fr (TV); Christophe.Godin@inria.fr (CG)

**Competing interests:** The authors declare that no competing interests exist.

## Introduction

Developmental systems strikingly produce regular patterns and analysis of eukaryote development has classically been focused on regularities as the main source of information to understand these complex systems. However, it is becoming increasingly evident that intrinsic molecular noise is an inherent property of biological systems (*Elowitz et al., 2002*; *Kupiec, 1997*; *Lander, 2011*). This noise can be buffered, e.g. (*Okabe-Oho et al., 2009*), but can also theoretically propagate through scales and generate patterning disorders e.g. (*Itoh et al., 2000*). In this case, disorders observed during development could be informative not only on the origin of noise but also on the underlying developmental mechanisms that propagate the noise. Here we address this question theoretically using phyllotaxis, the remarkably regular geometric organization of plant aerial organs (such as leaves and flowers) along the stem, as a model system (Appendix section 1).

Phyllotaxis primarily arises at the shoot apical meristem, a specialized tissue containing a stem cell niche and located at the tip of growing shoots. Rooted in early works of pioneers such as (*Bonnet, 1754*; *Braun, 1831*; *Bravais and Bravais, 1837*) and after decades of research, the idea that

**eLife digest** Plants grow throughout their lifetime, forming new flowers and leaves at the tips of their stems through a patterning process called phyllotaxis, which occurs in spirals for a vast number of plant species. The classical view suggests that the positioning of each new leaf or flower bud at the tip of a growing stem is based on a small set of principles. This includes the idea that buds produce inhibitory signals that prevent other buds from forming too close to each other. When computational models of phyllotaxis follow these 'deterministic' principles, they are able to recreate the spiral pattern the buds form on a growing stem.

In real plants, however, the spiral pattern is not always perfect. The observed disturbances in the pattern are believed to reflect the presence of random fluctuations – regarded as noise – in phyllotaxis. Here, using numerical simulations, Refahi et al. noticed that the patterns of inhibitory signals in a shoot tip pre-determine the locations of several competing sites where buds could form in a robust manner. However, random fluctuations in the way cells perceive these inhibitory signals could greatly disturb the timing of organ formation and affect phyllotaxis patterns.

Building on this, Refahi et al. created a new computational model of bud patterning that takes into account some randomness in how cells perceive the inhibitory signals released by existing buds. The model can accurately recreate the classical spiral patterns of buds and also produces occasional disrupted patterns that are similar to those seen in real plants. Unexpectedly, Refahi et al. show that these 'errors' reveal key information about how the signals that control phyllotaxis might work.

These findings open up new avenues of research into the role of noise in phyllotaxis. The model can be used to predict how altering the activities of genes or varying plant growth conditions might disturb this patterning process. Furthermore, the work highlights how the structure of disturbances in a biological system can shed new light on how the system works.

phyllotactic patterns emerge from simple physical or bio-chemical lateral inhibitions between successive organs produced at the meristem has become largely prevalent, (*Adler et al., 1997*; *Jean, 1995*; *Kuhlemeier, 2007*; *Pennybacker et al., 2015*; *Reinhardt, 2005*). Microscopic observations and modeling led to propose that this self-organizing process relies on five basic principles: *i*) organs can form only close to the tip of growing shoots, *ii*) no organ can form at the very tip, *iii*) pre-existing organs prevent the formation of new organs in their vicinity (*Hofmeister, 1868*), forming altogether an inhibitory field that covers the organogenetic zone, *iv*) due to growth, organs are progressively moved away from the organogenetic zone, *v*) a new organ is formed as soon as the influence of the inhibitory field produced by the existing organs fades away at the growing tip, (*Schoute, 1913*; *Snow and Snow, 1962*; *Turing, 1952*). Computer simulations were used to analyze the dynamical properties of an inhibitory field model relying on these assumptions (*Mitchison, 1977*; *Thornley, 1975*; *Veen and Lindenmayer, 1977*; *Young, 1978*) and many others after them, including (*Chapman and Perry, 1987*; *Douady and Couder, 1996a*; *Green et al., 1996*; *Meinhardt, 2003*; *Schwabe and Clewer, 1984*; *Smith et al., 2006b*). In a detailed computational analysis (*Douady and Couder, 1996a*; *1996b*; *1996c*), Douady and Couder demonstrated the ability of such models to recapitulate a wide variety of phyllotactic patterns in a parsimonious way and that these patterns are under the control of a simple geometric parameter corresponding to the ratio between the radius of organ inhibitory fields and the radius of the central zone (area at the very tip where no organ can form). This modeling framework thus provides a deterministic theory of self-organizing patterns in the meristem characterized by a global geometric parameter, capturing macroscopic symmetries and orders emerging from lateral inhibitions, e.g. (*Adler, 1975*; *Atela, 2011*; *Newell et al., 2008*; *Smith et al., 2006b*). In the sequel, we will refer to this widely accepted view as the *classical model* of phyllotaxis.

In recent years, plausible molecular interpretations of the abstract concepts underlying the classical model have been proposed. They mainly rely on distribution in the meristem of the plant hormone auxin, a central morphogenetic regulator. Auxin is actively transported at the meristem surface, notably by both PIN-FORMED1 (PIN1) polar efflux carriers and non-polar influx carriers (AUX/LAX family). These transporters form a dynamic network that permanently reconfigures and

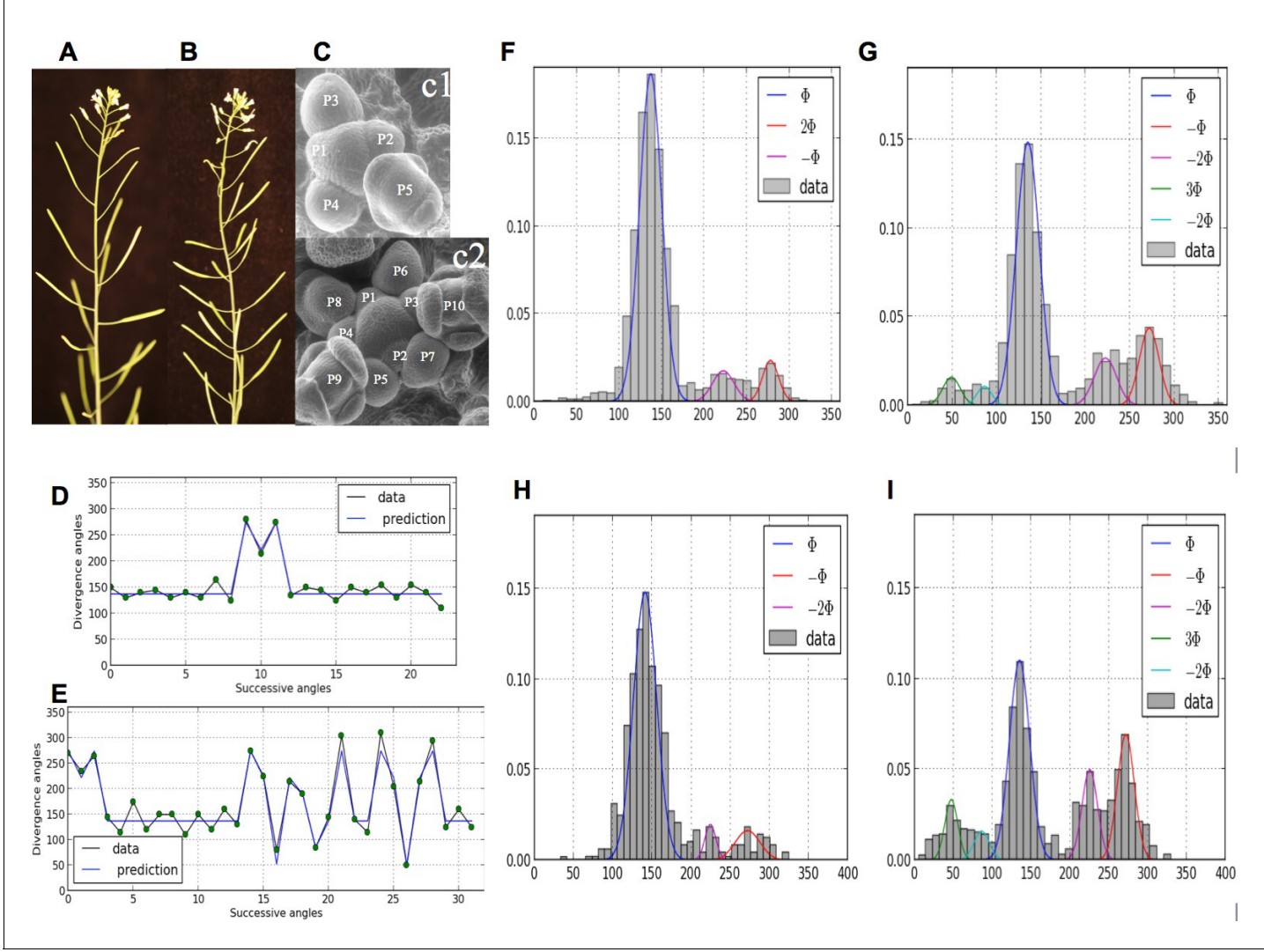

**Figure 1.** Irregularity in phyllotaxis patterns. (**A**) wild type inflorescence of *Arabidopsis thaliana* showing regular spiral phyllotaxis. (**B**) *aph6* mutant inflorescence showing an irregular phyllotaxis: both the azimuthal angles and the distances between consecutive organs are largely affected. (**C1**) Organ initiation in the wild type: the size of organs is well hierarchized, initiations spaced by regular time intervals. (**C2**) Organ initiation in the *ahp6* mutant: several organs may have similar sizes, suggesting that they were initiated simultaneously in the meristem (co-initiations). (**D**) A typical sequence of divergence angles in the WT: the angle is mainly close to (≈137°) with possible exceptions (M-Shaped pattern). (**E**) In *ahp6*, a typical sequence embeds more perturbations involving typically permutations of 2 or 3 organs. (**F–I**) Frequency histogram of divergence angle: wild type (**F**); *ahp6* mutant (**G**); WS-4, long days (**H**); WS-4 short days - long days (**I**).

that periodically accumulates auxin at specific locations on the meristem flanks (the organogenetic domain), initiating organ primordia (*Reinhardt et al., 2003*). By attracting auxin, the growing primordium depletes auxin in its vicinity, thus preventing organ formation in this region. This mechanism is now thought to be at the origin of the predicted inhibitory fields in the meristem (*Barbier de Reuille et al., 2006*; *Brunoud et al., 2012*; *Jönsson et al., 2006*; *Smith et al., 2006a*; *Stoma et al., 2008*). The range of this inhibition corresponds to one of the two key parameters of the classical model: as primordia get away from the tip, inhibition is relaxed and auxin can accumulate again to initiate new primordia. For the second key parameter, i.e. the size of the apical domain in which no organ can form, it has been suggested that the very tip of the meristem contains significant quantities of auxin but is actually insensitive to auxin due to a down-regulation of the effectors of transcriptional auxin signaling (*Barbier de Reuille et al., 2006*; *Vernoux et al., 2011*). A low auxin sensitivity then

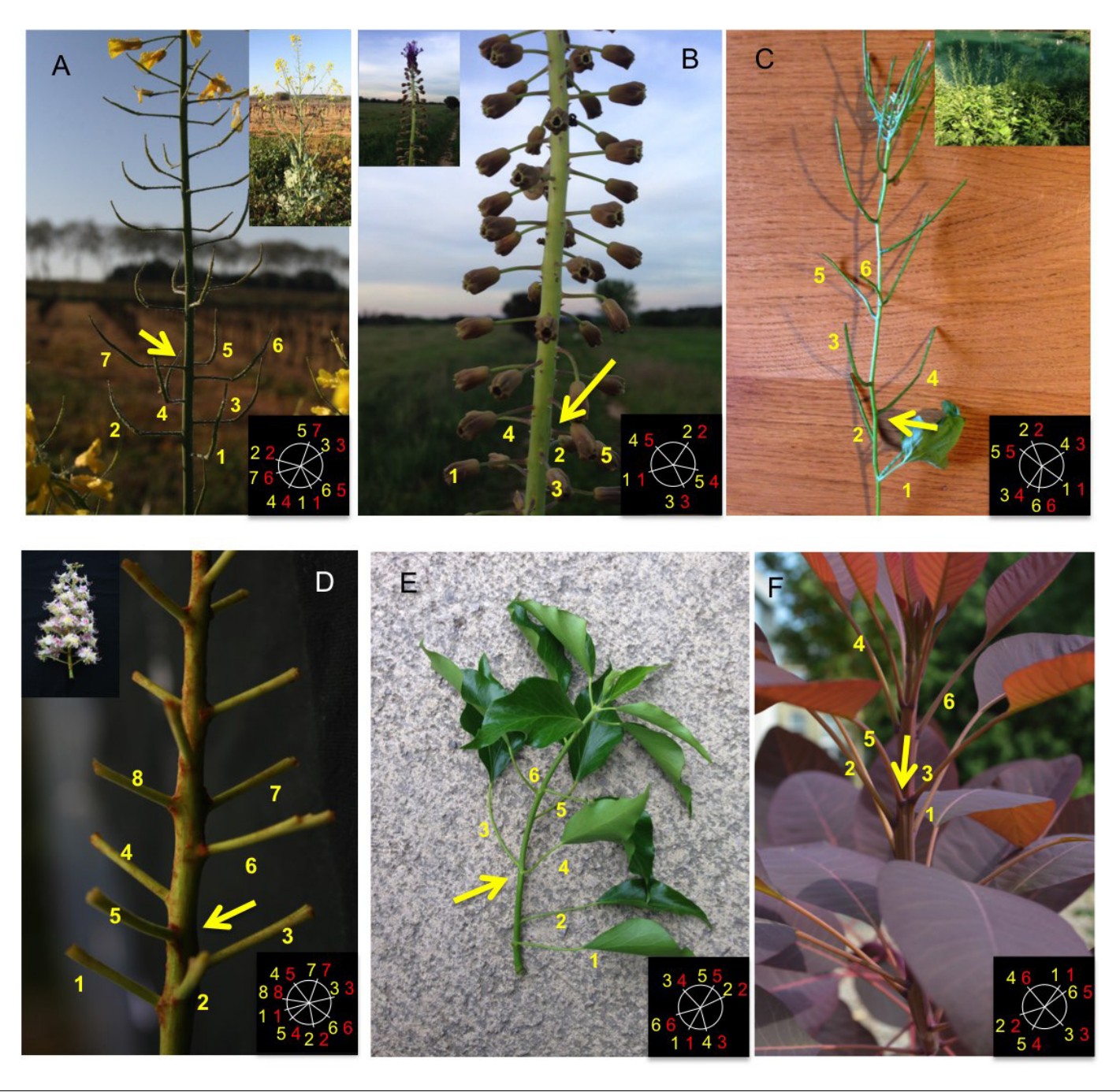

**Figure 2.** Permutations can be observed in various species with spiral phyllotaxis. A schema in the bottom right corner of each image indicates the rank and azimuthal directions of the lateral branches. The first number (in yellow, also displayed on the picture) indicates the approximate azimuthal angle as a multiple of the plant's divergence angle (most of the times close to 137° or 99°). The second number (in red) corresponds to the rank of the branch on the main stem. (A) *Brassica napus* (Inflorescence) (B) *Muscari comosum* (Inflorescence) (C) *Alliara petiolata* (D) *Aesculus hippocastanum* (Inflorescence) (E) *Hedera Helix* (F) *Cotinus Dummeri*.

participates in blocking organ initiation in the central domain (where the stem cells are located) of the meristem. These molecular insights support the hypothesized structure of the classical model.

Comparatively, little attention has been paid as of today to disorders in phyllotaxis (*Jean, 2009*; *Jeune and Barabé, 2004*). However, in the recent years, the presence of irregularities in phyllotactic

patterns has been repetitively observed in various genetic backgrounds (*Besnard et al., 2014*; *Couder, 1998*; *Douady and Couder, 1996b*; *Guédon et al., 2013*; *Itoh et al., 2000*; *Landrein et al., 2015*; *Leyser and Furner, 1992*; *Mirabet et al., 2012*; *Peaucelle et al., 2011*; *Prasad et al., 2011*; *Refahi et al., 2011*), suggesting that phyllotaxis has a non-deterministic component. In some cases, the departure from any known regular pattern is so strong that plant phyllotaxis is considered random, e.g. (*Itoh et al., 2000*). Recently, strong disorders have been observed and quantified in spiral patterns of *Arabidopsis thaliana* wild-type and *arabidopsis histidine phosphotransfer protein (ahp6)* mutants (*Besnard et al., 2014*), *Figure 1*. Surprisingly, a structure could be found in these disorders that corresponds to either isolated or series of permutations in the order of lateral organs along the stem when taking a perfect spiral with divergence angle 137.5° (golden angle $\phi$) as a reference. Live-imaging of meristems showed that organs are sometimes co-initiated in the meristem, leading randomly to post-meristematic organ order permutations in around half of the cases. This phenomenon showed that, while the azimuthal directions of the organs are highly robust, the time between consecutive organ initiation (or plastochron) is variable, *Figure 1G*. Taken together, these observations called for revisiting in depth models of phyllotaxis to account for disorders in this self-organizing developmental system.

Here, we show that the same disorders, the permutations, occur in various plant species, suggesting noisy plastochrons are a characteristic of phyllotactic systems at the origin of pattern disorders. In addition, we demonstrate that inhibitory fields pre-specify a number of organogenesis sites, suggesting noise on inhibition perception as the most likely origin of disorders. Building on this observation, we developed a stochastic model of organ initiation that is fully local and relies on a stochastic modeling of cell responses to inhibitory fields. Our stochastic model fully and precisely captures the observed dynamics of organogenesis at the meristem, recapitulating both regular and irregular phyllotactic patterns. We show that the stochastic model also makes quantitative predictions on the nature of the perturbations that may arise due to different genetic and growth manipulations. Most importantly, we demonstrate that disorders in phyllotactic patterns instruct us on the parameters governing the dynamics of phyllotaxis. Disorders can thus provide access to the biological watermarks corresponding to the parameter values of this self-organizing system, providing a striking example where disorders inform on mechanisms driving the dynamics of developmental systems.

## Results

### The shoot architecture of a variety of plant species suggests that disorder is a common phenomenon in phyllotaxis

As permutations have been notably reported in Arabidopsis (*Besnard et al., 2014*; *Guédon et al., 2013*; *Landrein et al., 2015*; *Refahi et al., 2011*) and in sunflower (*Couder, 1998*), we sampled a variety of unrelated species in the wild and searched for permutations. We could easily find permutations in several other Brassicaceae showing spiral phyllotaxis as well as in either monocotyledonous or dicotyledonous species from more distant families such as Asparagaceae, Sapindaceae or Araliaceae (*Figure 2*) (Appendix section 1). As suggested by the results on Arabidopsis, these observations raise the possibility that these organ permutations result from a noise on the plastochron and that such perturbation could be a common feature of phyllotactic systems that occurs in meristems with different geometries. In addition this disorder is probably under complex genetic control as different unrelated genetic modifications (including the Arabidopsis *ahp6* mutant) can modulate its intensity.

### The classical deterministic model suggests inhibition perception as the most likely origin of disorders

To understand how the timing of organ initiation could be affected during meristem growth, we first analyzed the relative stability of inhibitory field minima in the classical deterministic model. For this, we implemented a computational version of the classical model based on (*Douady and Couder, 1996b*) (Appendix section 2). Primordia are created on the meristem surface at the periphery of the meristem central zone, at a fixed distance $R$ from the meristem center. Once created, primordia drift away radially from the central zone at a speed proportional to their distance from the meristem center. As soon as created, a primordium $q$ of radius $r_0$, generates an inhibitory field, $E^{(q)}$ in its

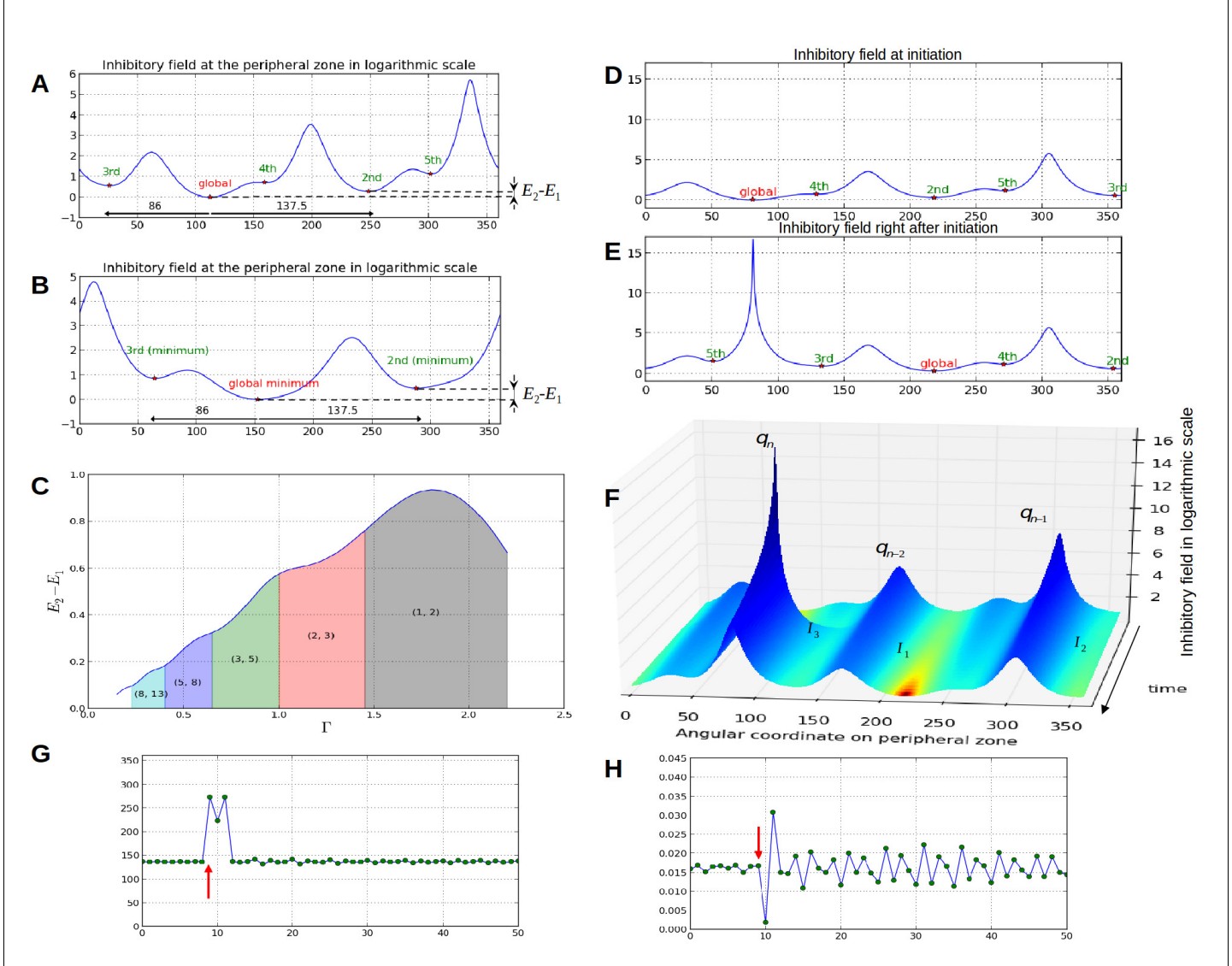

**Figure 3.** Properties of the inhibition profiles in the classical model and effect of a forced perturbation on divergence angles and plastochrons. (**A**) Inhibition variation (logarithmic scale) along the peripheral circle and its global and local minima for a control parameter $\Gamma_1 = 0.975$. $E_k - E_1$ is the difference in inhibition levels between the $k$th local minimum and the global minimum. The angular distances between the global minimum and the $k$th primordiu inhibition m are multiple of the canonical angle ($\alpha = 137°$). (**B**) Similar inhibition profile for a control parameter $\Gamma_2 = 0.675 < \Gamma1$. The difference $E_k - E_1$ in inhibition levels is higher than in A. (**C**) Variation of the distance $E_2 - E_1$ between the global minimum of the inhibition landscape and the local minimum with closest inhibition level (i.e. the second local minimum), as a function of $\Gamma$. (**D**) Inhibition profile just before an initiation at azimuth 80° and (**E**) just after. (**F**) Variation of inhibition profile in time. As the inhibition levels of local minima decrease, their angular position does not change significantly, even if new primordia are created (peak $q_n$), color code: dark red for low inhibition and dark blue for high inhibition values. (**G**) Sequence divergence angles between initiations simulated with the classical model (control parameter $\Gamma_1$). At some point in time (red arrow), the choice of the next initiation is forced to occur at the 2th local minimum instead of the global minimum. After the forcing, the divergence angle makes a typical M-shaped pattern and returns immediately to the $\alpha$ baseline. (**H**) Corresponding plastochrons: the forcing (red arrow) induces a longer perturbation of the time laps between consecutive organs.

The following figure supplement is available for figure 3:

**Figure supplement 1.** Divergence angle of a series of simulations of the classical model with control parameter $\Gamma_1$=0.975 and for which the choice of the $j$th local minimum (instead of the global minimum, i.e. $j$=1) has been forced at a given time-point (red arrow).

neighborhood such that at a point $x$, at a distance $d(q,x) > r_0$, the inhibition due to $q$ decreases with the distance to $q$: $E^{(q)}(x) = (\frac{r_0}{d(q,x)})^s$, where $s$ is a geometric stiffness parameter ($E^{(q)}$ is often regarded as an *inhibitory energy* emitted by primordium $q$, e.g. **Douady and Couder, 1996b**). As a result, at any moment and for any point $x$ of the meristem surface, the existing primordia create altogether a cumulated inhibition $E(x)$ that is the sum of the individual primordium contributions: $E(x) = \sum\limits_{q \in Q} E^{(q)}(x)$, where $Q$ denotes the set of all preexisting primordia. High inhibition levels on the peripheral circle prevent the initiation of new primordia at corresponding locations. However, as the preexisting primordia are moving away from the central zone during growth, the inhibitory level tends to decrease at each point of the peripheral circle. A new primordium initiates where and when the inhibitory field is under a predefined threshold on the peripheral circle.

We then performed a systematic analysis of the inhibition profiles and their dynamics along the peripheral circle and observed the following properties (*Figure 3*, *Video 1,2*):

- Property 1: let $i$ and $j$, $i < j$, denote the numbers of contact spirals that one can observe on a stem in clockwise and counter-clockwise directions (Appendix section 1), the number of local minima $l$ in the inhibition profile is bounded by $i$ and $j$ : $i \leq l \leq j$: (*Figure 3A–B*).
- Property 2: The angular distance between local minima is a multiple of the divergence angle (*Figure 3A–B*).
- Property 3: The difference between the inhibition values of local minima decreases monotonically with the control parameter $\Gamma = r_0/R$ (*Figure 3C*)

In addition, we noticed that depending on time, the difference in inhibition level between consecutive local minima may markedly vary. In some cases this difference is so small that a biological noise may lead the biological system to perceive the ordering between two or more local minima differently from the ordering of the actual inhibition levels. Such errors would lead to initiate several primordia together or to change the temporal order of their initiation, thus inducing perturbations in the sequence of divergence angle. Also, as suggested by property 3, the number of primordia initiation events affected by these errors would decrease with the $\Gamma$ parameter and would thus depend on the geometry of the meristem.

To investigate this possibility, we started to induce a perturbation in a stationary spiral pattern by forcing at a given time the system to initiate a primordium at the site of second local minimum instead of that of the global minimum (i.e. at an angle $2\phi$, *Figure 3G*, red arrow). The system was then left free to self-organize. We observed that the next primordium was always initiated at the site of the original global minimum, resulting in a divergence angle $-\phi$ and a quasi-null plastochron (*Figure 3H*). The system was then able to recover from the perturbation by initiating the next primordium at the originally expected site (i.e. with a divergence angle $2\phi$), with a long plastochron, leading to a M-shaped pattern (*Figure 3G*, *Video 3*). The rest of the divergence angle sequence was then not affected and remained at a value close to $\phi$ while long oscillatory perturbations were observed on plastochrons (*Figure 3H*). Stronger perturbations induced by forcing various other local minima to initiate instead of the global minimum (*Figure 3—figure supplement 1* and Appendix section 6.6 for related control simulation experiments) similarly demonstrated that the system spontaneously makes a short distorted pattern and then returns to the normal $\phi$ baseline in every case. We concluded that *i*) a noise in the perception of the local primordium order does not propagate far in the divergence angle sequence and that angle specification in the classical model patterning system is highly robust to perturbations in local minima initiation ordering and *ii*) however, the plastochron itself is affected during a much longer time span.

Together, these results suggest that time and space in primordium initiation are largely decoupled in inhibitory field-driven self-organization as locations of primordia are strongly pre-specified and relatively stable, while plastochrons are not. This observation is in line with the observations from live-imaging of *Arabidopsis* shoot apical meristem that demonstrated that despite variability in the plastochron (with almost 30% of organs co-initiated) the specification of initiation sites was extremely robust (*Besnard et al., 2014*).

## Organ initiation can be modeled as a stochastic process

Our previous results suggest that high variability in the timing of organ initiation could result from the joint effect of noise in the perception of inhibitory fields and of the decoupling between space

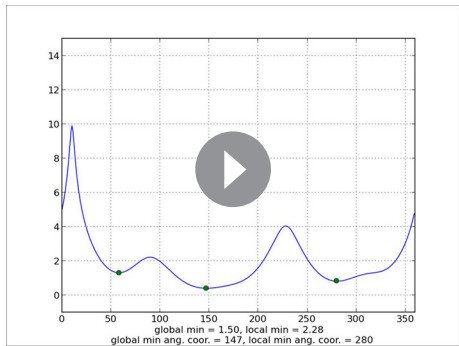

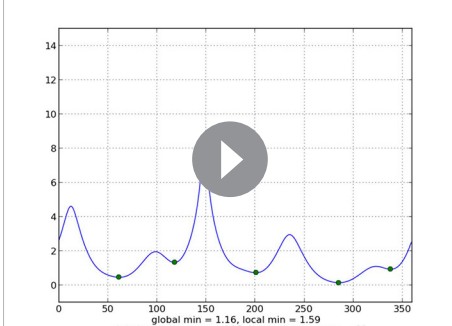

**Video 1.** Temporal variation of the inhibitory profile around the central zone in the classical model for a large value of the parameter Γ. The number of inhibition mimima is stable (3) in time. When the absolute minimum reaches the initiation threshold (here $E = 0$), a primordium is created that instantaneously creates a strong inhibition locally, which suddenly increases the inhibition level at its location. Between initiations, local minima regularly decrease in intensity due to the fact that growth is moving existing primordia away from the center. This movement is accompanied by a slight drift in position common to all primordia (here to the right).

**Video 2.** Temporal variation of the inhibitory profile around the central zone in the classical model for a small value of the parameter Γ. The dynamics is similar to that of small Γ except that the number of local minima of inhibition is higher (here 5) and that the distance between two consecutive minima is lower.

and time in this self-organizing system. We therefore decided to revisit the inhibitory field models to integrate locality and stochasticity as central components in the patterning system. For this, we kept from the classical deterministic model the assumptions related to the movement of primordia at the meristem through growth

and to the definition of inhibitory fields. However, we completely reformulated the way primordia are initiated as local stochastic processes.

At any time $t$, the $K$ cells that make up the periphery of the central zone potentially may take the identity of a primordium, depending on the local value of the inhibitory field (signaling) in each cell. We assume that this switch in cell identity does not depend on a threshold effect as in the classical model but, rather, depends on the cellular perception of the inhibitory signal which, in essence, is stochastic (**Eldar and Elowitz, 2010**). We thus assumed that each cell $k$ reads out the local inhibitory field value, $E_k(t)$, and switches its state to primordium identity with a probability that (*i*) depends on the level of inhibition $E_k(t)$; (*ii*) is proportional to the amount of time δt the cell is exposed to this level of signaling (**Gillespie, 1976**; **1977**):

$$P(X_k(t, \delta t) = 1) = \lambda(E_k(t))\delta t, \quad (1)$$

where $X_k(t, \delta t)$ denotes the number of primordia initiated at cell $k$ in the time interval $[t, t + \delta t]$ ($X_k(t, \delta t)$ will typically have a value 0 or 1), and $\lambda$ is a rate parameter that depends on the local inhibition value $E_k(t)$ at cell $k$ and that can be interpreted as a temporal density of initiation. To express the influence of inhibitory fields on the probability of initiation, the dependence of $\lambda$ on the local inhibition $E_k(t)$ must respect a number of general constraints: *i*) $\lambda$ must be a decreasing function of the inhibition as the higher the inhibition level at one site, the lower the probability to observe an initiation at this site during δt; *ii*) for small δt, for $P(X_k(t, \delta t) = 1)$ to be a probability, we must have $0 \leq \lambda(E_k(t))\delta t \leq 1$; *iii*) the ratio of the probabilities

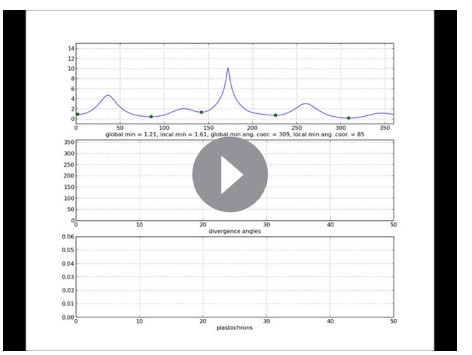

**Video 3.** Temporal variation of the inhibitory profile around the central zone in the stochastic model for a small value of the parameter Γ. Here, due to stochasticity, global minimum is not always the one that triggers an initiation. The dynamics of the divergence angle and of the plastochron are shown in the bottom graphs to interpret the model's initiations based on the inhibition levels.

to trigger an initiation at two different sites is a function of the difference in inhibition levels between these sites. Under all these constraints, the rate parameter takes the following form (see Appendix section 3 for a detailed derivation of the model):

$$\lambda(E_k(t)) = e^{-\beta(E_k(t) - E^*)}, \tag{2}$$

where $E^*$ is a parameter controlling the sensitivity of the system to inhibition, and $\beta$ is a parameter controlling the ability of the system to discriminate between inhibition levels, *i.e.* to respond differently to close inhibition levels (acuity). Therefore, for each cell $k$, the probability to initiate a primordium during a small time interval $\delta t$ can be expressed as:

$$P(X_k(t, \delta t) = 1) = e^{-\beta(E_k(t) - E^*)} \delta t. \tag{3}$$

If we now assume that the probabilities to observe an initiation at a site $k$ in disjoint time intervals are independent, the process described by *Equation 3* is known as a non-homogeneous Poisson process of intensity $\lambda(E_k(t))$ (e.g. *Ross, 2014*). Therefore for each cell $k$ of the periphery, our model assumes that the probability to initiate a primordium is a non-homogeneous Poisson process, whose parameter is regulated by the local level of the inhibitory field at that site.

This stochastic formulation of the model at the level of cells, called SMPmicro (Stochastic Model of Phyllotaxis at microscopic level), makes it possible to develop the calculus of different key quantities or properties of the system. For example, if we assume that recruitments of cells for organ initiation are stochastically independent from each other (the probability to draw an initiation at a site $k$ only depends on the value of the local parameters at site $k$ but not on what may be drawn at other places), then we can estimate the expected number of cells independently recruited for organ initiation during the timespan $\delta t$. Let us denote $X(t, \delta t)$ the number of cells initiated along the peripheral circle, and for a given time $t$ and a small time span $\delta t$, $X(t, \delta t) = \sum_{k=1}^{K} X_k(t, \delta t)$. Its expectation is simply the sum of the expectations of the individual independent Poisson processes:

$$\begin{aligned} \mathrm{E}(X(t, \delta t)) &= \sum_{k=1}^{K} \mathrm{E}(X_k(t, \delta t)) = \sum_{k=1}^{K} P(X_k(t, \delta t) = 1) \\ &= \delta t \sum_{k=1}^{K} e^{-\beta(E_k(t) - E^*)}, \end{aligned} \tag{4}$$

therefore giving us an estimate of the expected number of peripheral cells initiated during time $\delta t$ and for the inhibition profile $E(t)$.

So far we have considered peripheral cells as independent sites that may independently switch to primordium identity with a probability that depends on their local level of inhibition. As a local inhibition valley may span over several cells, one might expect that the probability to trigger a primordium initiation in a valley is increased by the fact that several founder cells can potentially contribute to this initiation during time $\delta t$. To formalize this, we then upscaled our stochastic model at the level of valleys where a stochastic process is now attached to each local minimum $l$ instead of to each cell $k$. This upscaled model is called SMPmacro. The idea is that the stochastic processes of all the cells $k$ spanned by a local inhibition valley $l$ sum up and together define a stochastic process $N_l$ that a primordium is initiated in $l$ at a higher level. Being the sum of independent Poisson processes, this upscaled process is also a Poisson process with intensity $\Lambda_l = \sum_{k \in K_l} \lambda_k(E_k(t))$, where $k$ varies over the set of cells spanned by the valley of the $l$ th local minimum and indexed by $K_l$. If $L$ denotes the number of local minima, then the expected number of primordia initiations during a small time laps $\delta t$ is thus:

$$\mathrm{E}(N_l(t, \delta t)) = \delta t \sum_{l=1}^{L} \Lambda_l(t). \tag{5}$$

Therefore, at microscopic scale, SMPmicro couples *i*) a deterministic part inherited from the classical model and related to the geometry and the dynamics of the fields and *ii*) a new stochastic part related to the perception of this inhibitory field, *i.e.* representing signal perception capacities. It relies on the assumption that the perception sites, corresponding to the cells surrounding the central

zone periphery, are stochastically independent of each other. Decisions regarding primordium initiation are taken in a cell-autonomous manner, thus reflecting more realistically the outcome of the initiation signaling pathway in each cell. At a macroscopic level, in each inhibition valley several cells may trigger primordium initiation. The probability to trigger an initiation increases in SMPmacro with the size of the valley when more than one founder cell are likely to contribute to initiation. A variant of this upscaled model consists of defining the probability for a valley to initiate a primordium by the probability of the cell with lowest level of inhibition in this valley. In this variant, called SMPmacro-max, $\Lambda_l = \max_{k \in K_l} \lambda_k(E_k(t))$.

## Stochastic modeling simulates realistic phyllotaxis sequences

To study the emergent properties of this system, we implemented a computational version of SMPmacro-max and tested its sensitivity to parameter changes. In addition to the geometrical parameters of the classical model, two new parameters, $\beta$ and $E^*$ now reflect the ability of the system to perceive the inhibitory signal from the fields (or equivalently the initiation signal).

As expected from a phyllotaxis model, the stochastic model is able to produce both spiral and whorl modes, from either imposed initial distributions of organs (*Figure 4A–C*), or random starting points (Appendix section 6.1). However, the great majority of the sequences of divergence angle generated for different values of $\Gamma = r_0/R$ displayed divergence angle perturbations (*i.e.* angles different from those predicted by the classical model with the same $\Gamma$) of the type observed in *Figure 4D–E*. As suggested by their typical distributions (*Figure 4F–H* left), these perturbations correspond to permutations very similar to those observed on real plants in previous studies (*Besnard et al., 2014*; *Landrein et al., 2015*) with the appearance of secondary modes at multiples of 137, *Figure 1F–I*. The amplitude of these secondary modes is correlated with the amount of perturbations in the sequences.

To complete this analysis, we looked at plastochron distributions in the simulated sequences (*Figure 4F,G,H* right). In all cases, distributions were displaying a single mode largely spread along the $x$-axis. Interestingly, the more sequences were perturbed, the more negatively skewed the distributions, showing thus a higher occurrence of short plastochrons (*Figure 4H*). This is reminiscent of the observation of co-initiations in growing meristems associated with perturbed phyllotaxis (*Besnard et al., 2014*; *Landrein et al., 2015*). The stochastic model is thus able to produce perturbed sequences with realistic series of divergence angles and corresponding realistic distributions of plastochrons.

## The proportion of complex versus simple disorders in phyllotaxis sequences depends on the global amount of phyllotaxis disorders

We then aimed to quantitatively assess the complexity level of permutations as a proxy for plastochron noise. For this we focused on spiral phyllotaxis modes and used two measures: the density of permuted organs respectively involved in 2 and 3-permutations $\pi_2 = 2.\sigma_2/\Sigma$ and $\pi_3 = 3.\sigma_3/\Sigma$ where $\sigma_2$ and $\sigma_3$ are respectively the number of 2- and 3-permutations in the sequence and $\Sigma$ the total number of organs in the sequence. The quantities $\sigma_2$ and $\sigma_3$ were estimated *a posteriori* from simulated sequences using the algorithm described in (*Refahi et al., 2011*). The total density of permuted organs involved either in 2- or 3 permutations is denoted by $\pi = \pi_2 + \pi_3$.

We first explored the intensity of permutations of different natures (2- and 3-permutations) in simulated phyllotaxis sequences. For this, we carried out simulations for a range of values for each parameter $\Gamma, \beta, E^*$ (*Figure 5—source data 1*). These values can be regarded as predictions of the model for each triplet $(\Gamma, \beta, E^*)$. Remarkably, the simulations show that the values of $\pi, \pi_2$ and $\pi_3$ are not linearly related to each other. To make this relationship explicit, we plotted the proportion of organs involved in 2-permutations $\pi_2$ as a function of the total proportion of perturbed organs $\pi$ (*Figure 5*). Surprisingly, the points, when put together on a graph, were organized in a narrow crescent showing a convex curve-like relationship between $\pi$ and $\pi_2$, revealing a remarkable property of the stochastic model: the more perturbations there are in the simulated sequences, the higher the proportion of 3-permutations, and this independently of the model parameters. The fact that this non-linear relationship emerges from a fairly large sampling of the parameter space suggests that it can be considered as a key observable property characterizing the model's underlying structure.

We tested this by gathering all measured values of $\pi_2$ and $\pi_3$ published in the literature for *Arabidopsis* (*Besnard et al., 2014*; *Landrein et al., 2015*) and plotted the corresponding points on the original graph showing the model's simulations (*Figure 5*, red crosses). The measured points fall within the range of predicted values and show that the measured values follow the same non-linear variation as the one predicted by the model: the larger the total percentage of permutations, the larger the proportion of 3-permutations in the sequences. This confirms a first prediction from the stochastic model and indicates that disorder complexity increases non-linearly with the frequency of disorders.

## The amount of disorders in a sequence depends on both geometry and signal perception

Based on our simulations, we then investigated the variations of these perturbations as a function of the model parameters $\Gamma, \beta.E^*$ and $s$. As a general trend, for a value of $s$ fixed to 3 as in (*Douady and Couder, 1996b*), we noticed that for a given value of $\Gamma$, an increase in $\beta$ was roughly counteracted in terms of permutation intensity by a decrease in $E^*$. Likewise, an increase of $\Gamma$ was canceled by a decrease in $\beta$. We gathered all these observations in one graph and plotted the global amount of perturbation $\pi$ in a sequence as a function of a combination of the three original model parameters and reflecting the observed trend: $\Gamma_P = \Gamma\beta E^*$. In the resulting graph (*Figure 6A*), each point corresponds to a particular instance of the three model parameters. The points form a narrow and decreasing band associating a small set of possible perturbation intensities $\pi$ with each value of $\Gamma_P$. For combinations of the three model parameters leading to a small $\Gamma_P$, the perturbations can affect up to 50% of the organs whereas for high values of $\Gamma_P$, there may be no perturbation at all. Interestingly, $\Gamma_P$ being a combination of the three elementary model parameters, plants having identical geometrical parameter $\Gamma$ may show substantially different intensities of perturbations if their perception for different values of parameters $\beta, E^*$. Consequently, since the intensity of perturbation in the system is more simply reflected by $\Gamma_P$ than by values of $\Gamma, \beta, E^*$ taken independently, $\Gamma_P$ can be considered to be a control parameter for perturbations.

## Both divergence angles and plastochrons are controlled by a unique combination of the geometric and perception parameters

To further investigate the structure of the stochastic model, we then studied how the usual observable quantities of a phyllotaxis system, i.e. divergence angles and plastochrons, depend on the model parameters $\Gamma, \beta, E^*$ and $s$.

In the classical deterministic model, divergence angles are a function of a unique control parameter $\Gamma = r_0/R$ (*Douady and Couder, 1996b*), meaning that the same divergence angle can be obtained in the model for different couples of $r_0$ and $R$ provided that their ratio is unchanged. We thus checked whether $\Gamma$ could also serve as the control parameter for the divergence angles and plastochrons in the stochastic model. For this, we simulated various spiral phyllotaxis sequences (from the Fibonacci branch where divergence angles are close to 137°) by varying $\Gamma, \beta, E^*$ and estimated the corresponding divergence angle $\alpha$ and plastochron $T$, *Figure 6B,D*. We observed that, although the point clouds evoke the corresponding curves in the standard deterministic model (*Appendix 1—figure 2*), significantly different values of the divergence angles (or plastochrons) can be observed for many particular values of $\Gamma$. This phenomenon suggests that $\Gamma$ is not a satisfactory control parameter in the stochastic model: for a given value of $\Gamma$, varying $\beta$ or $E^*$ can significantly modify the divergence angle or the plastochron. We therefore tested various combinations of these parameters (*Figure 6—figure supplement 1*), and finally found that for $\Gamma_D = \Gamma^{s/3}\frac{1}{\beta^{1/6}E^{*1/2}}$ both cloud of points collapse into a single curves (Figure 6B–C, D–E and Appendix section 6.3). For this definition of $\Gamma_D$, each value of the control parameter can be associated with quasi-unique divergence angle and plastochron, i.e. defines a precise observable state of the system.

We then checked whether parastichies were also controlled by $\Gamma_D$. For this, we computed the parastichies $(i,j)$ corresponding to each simulated sequence and plotted the sum, $i + j$, as a function of $\Gamma_D$ (*Figure 6F*). The resulting points were arranged on a stepwise curve, where each step corresponds to a Fibonacci mode: (1,2), (2,3), (3,5), (5,8). Each value of $\Gamma_D$ thus defines a precise value of the mode.

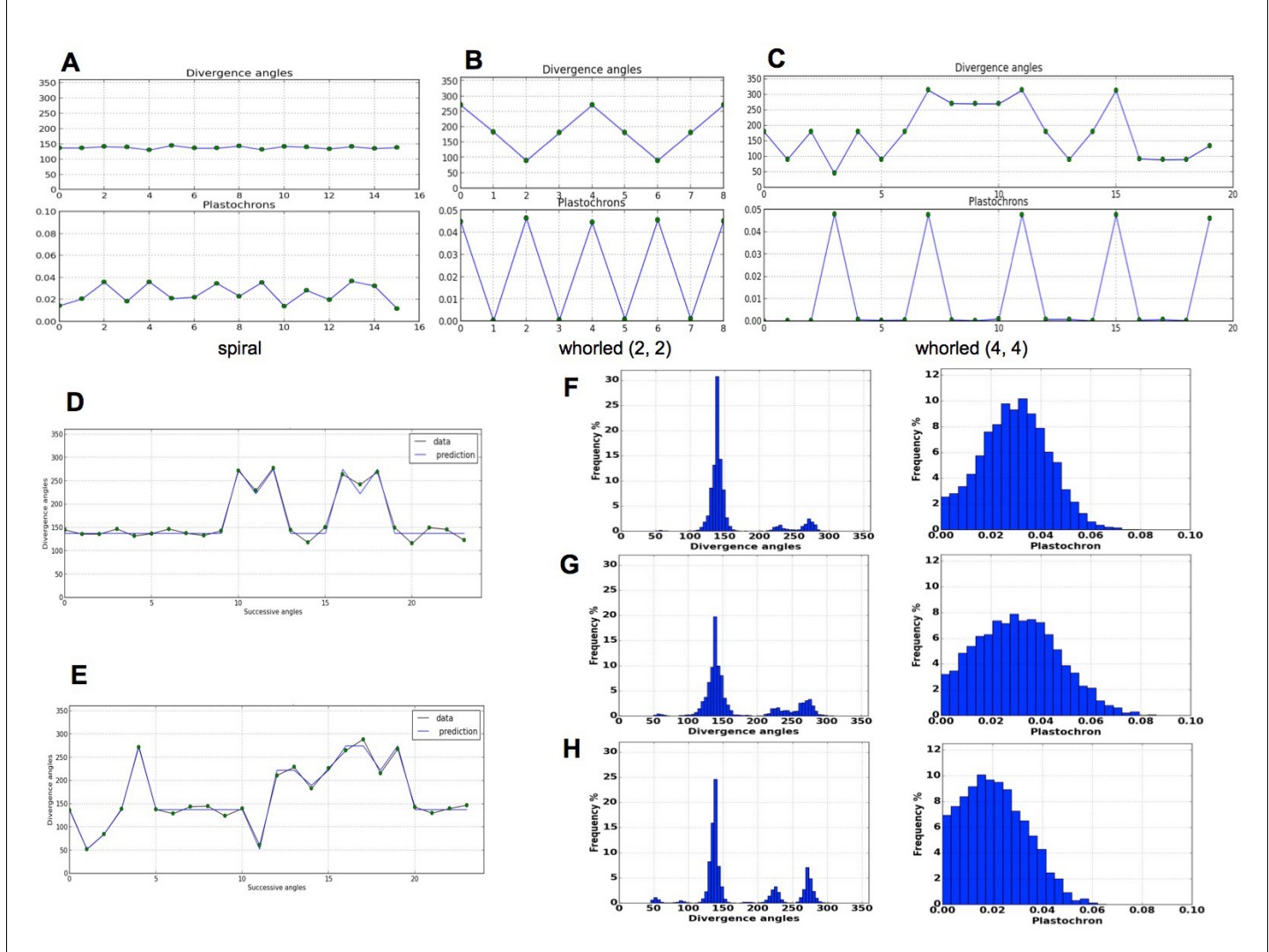

**Figure 4.** Patterns generated by the stochastic model. (A) The model generates spiral patterns (in (A–C), up: sequence of simulated divergence angles, down: corresponding plastochrons). (B–C) and whorled patterns. (D) Simple M-shaped permutations simulated by the stochastic model ($\beta$ = 10.0, $E^*$ = 1.4, $\Gamma$ = 0.625). (E) More complex simulated permutations involving 2- and 3-permutations ($\beta$ = 10.0, $E^*$= 1.4, $\Gamma$ = 0.9). The permutations are here: [4, 2, 3], [14, 13, 12], [16, 15], [19, 18]. (F) Typical histogram of simulated divergence angles and corresponding plastochron distribution for $\beta$ = 11.0, $E^*$ =1.2, $\Gamma$ = 0.8. (G) Histogram of simulated divergence angles and corresponding plastochron distribution for $\beta$ = 9.0, $E^*$ = 1.2, $\Gamma$ = 0.8 (H) Histogram of simulated divergence angles and corresponding plastochron distribution for $\beta$ = 9.0, $E^*$ = 1.2, $\Gamma$ = 0.625.

We concluded that $\Gamma_D$ can be considered as a second control parameter of the stochastic model, relating the system's state to the observable variables $\alpha$ and $T$ through a unique combination of the system's parameters.

## Observable variables convey key clues on the state of the phyllotaxis system

According to the stochastic model, a particular phyllotaxis system is characterized by a particular set of values of the parameters $\Gamma, \beta, E^*$. Upon growth, a specific spatio-temporal dynamics emerges that is characterized by observable variables: divergence angle, plastochron and frequency of permutations. In the current state of our knowledge and measuring means, the parameters $\Gamma, \beta, E^*$ are not directly observable. Therefore, we investigated what can be learned about them from the observable variables.

For this, we can use the relationships established above between the control parameters $\Gamma_P$ and $\Gamma_D$ and the observable variables $\alpha, T, \pi_2, \pi_3, \ldots$. For a genotype $G$, we have:

$$
\begin{aligned}
\Gamma_P &= \Gamma\beta E^* \\
\Gamma_D &= \Gamma^{s/3}\frac{1}{\beta^{1/6}E^{*1/2}}
\end{aligned}
\tag{6}
$$

Using the characteristic curves of *Figure 6C,E*, both measurements of $\alpha$ and $T$ give possible estimates for $\Gamma_D$. In a consistent model, these estimates should be compatible. Similarly, an estimate of $\Gamma_P$ can be derived from the observation of $\pi$. In the case of the WT for example, $\Gamma_D$ is estimated to be in [0.425, 0.5] and $\Gamma_P$ in [8.30, 11.49] according to the observed value of the plastochron, divergence angle and permutation intensity. This value is itself derived from the measurement of the plastochron ratio $\rho$ (Appendix section 4.1.3).

Based on estimated values of $\Gamma_D$ and $\Gamma_P$, equations 6 define a system of 2 equations and 3 unknowns ($s$ was fixed to 3 in the simulations). This system is underdetermined and does not allow us to identify exactly the values of $\Gamma, \beta, E^*$. However, being underdetermined in one dimension only, this system plays the role of a generalized control parameter: a given value of the generalized control parameter ($\Gamma_D$, $\Gamma_P$) determines the parameters $\Gamma, \beta, E^*$ up to one degree of freedom. If one of the parameters is given, then the others are automatically determined according to *equation 6*.

## Experimental observation of anti-correlated variations in disorder and plastochron is interpreted by the model as a change in inhibitory field geometry

Several recent works demonstrated that mutations or changes in growth conditions could alter phyllotaxis and disorder patterns. Our model predicts correlations between main observable phyllotaxis variables. According to the model, plastochron positively correlates with $\Gamma_D$ (*Figure 6E*). If signaling is not altered by the experimental setup ($\beta$ and $E^*$ are unchanged) then $\Gamma_D$ positively correlates with $\Gamma = r_0/R$ (*Equation 6*). Therefore, the model predicts that, like in the classical model, plastochron positively correlates with $r_0$ (size of the primordia inhibitory fields) and negatively correlates with $R$ (size of the central zone). However, the model makes also in this case the new prediction that plastochron negatively correlates with the frequency of observed permutations (*Figure 6A*).

A series of recent observations support this prediction. By changing growth conditions (plants first grown in different day-length conditions and then in identical conditions) or by using different accessions or mutants with markedly different meristem sizes from that of the wild type (*Landrein et al., 2015*), changes in the size of the meristem could be induced. The authors hypothesized that this change affected the size of the central zone only and not the size of primordia inhibitory fields. Corresponding changes in $\Gamma$ were observed to positively correlate with the frequency of organ permutations and negatively correlate with plastochron, as predicted by our model. In addition, the stochastic model makes it possible to quantitatively estimate the changes in central zone sizes from the measured phyllotaxis disorders with an error less than 5% (Appendix section 4.3).

In a previous study on *ahp6* mutants, *Besnard et al. (2014)* showed that the frequency of disorders could markedly augment while the size of meristems did not significantly change like in (*Landrein et al., 2015*). As discussed above, this change in disorder intensity could theoretically be due to an alteration of initiation perception in the mutant. However, the stochastic model suggests that it is not the case. Indeed, we re-analyzed plastochrons of the mutants and could observe that, although the change is limited, mutant plastochrons are significantly smaller than those of the wild type (Appendix section 4.2). According to *Figure 6E*, this means that $\Gamma_D$ is reduced in the mutant. If this reduction was due to an increase in either $\beta$ or $E^*$, then according to the model, one should expect a corresponding increase of $\Gamma_P$ (*Equation 6*) and, thus, a decrease of disorders (*Figure 6A*). On the contrary a significant increase of disorders was actually observed, suggesting that perception is not altered and that, rather, a decrease in $\Gamma = r_0/R$ could be the source of variation. Since the size $R$ of the meristems did not change, the model suggests that *ahp6* is altered in the size $r_0$ of the primordia inhibitory fields.

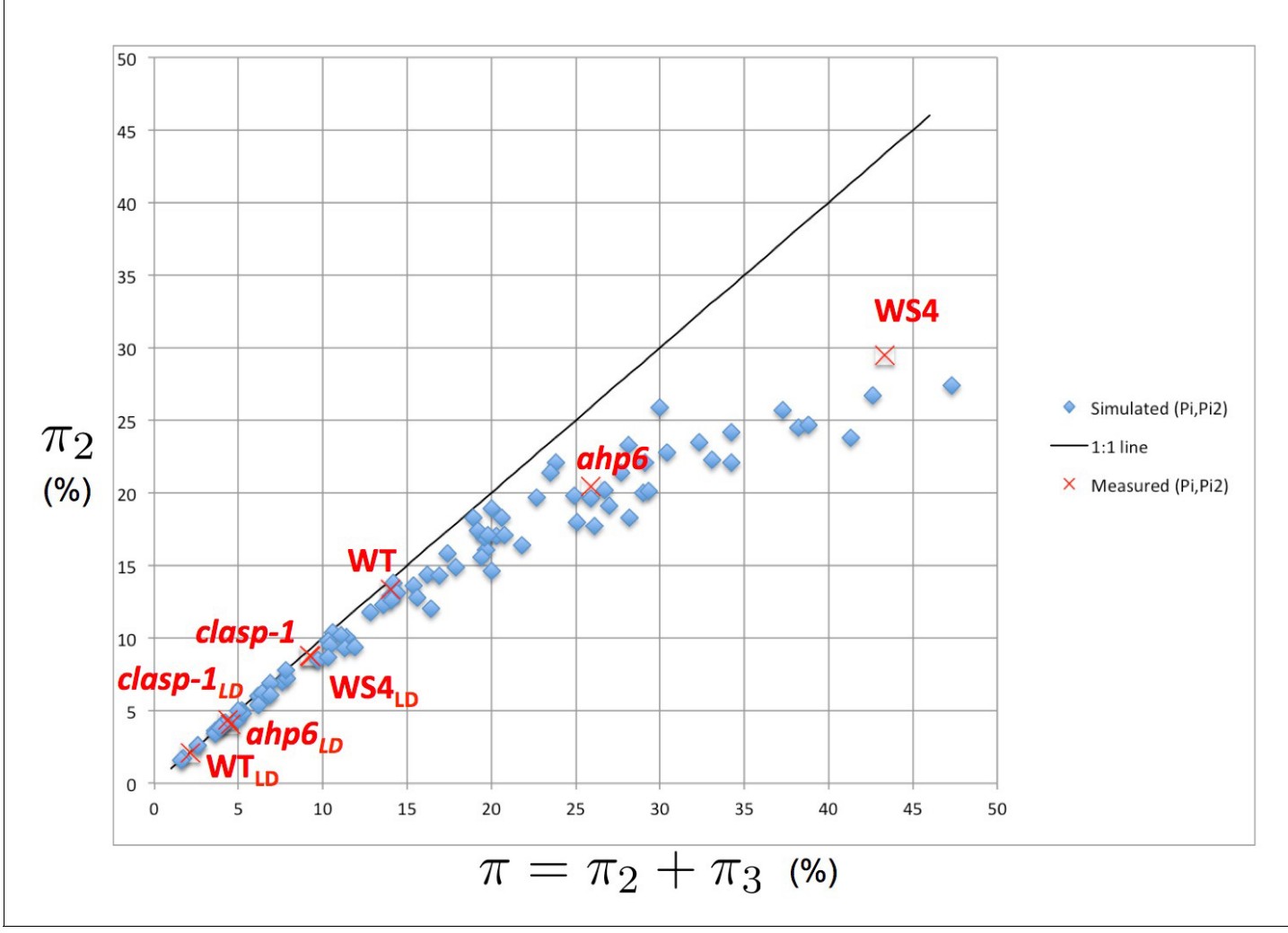

**Figure 5.** Intensity of 2-permutations as a function of the total amount of perturbations. As the perturbation intensity $\pi$ increases, the percentage of 2-permutations decreases in a non-linear way to the benefit of more complex 3-permutations. The diagonal line denotes the first bisector. In red: values of 2- and 3-permutations observed in different mutants and ecotypes of Arabidopsis thaliana (*Besnard et al., 2014*; *Landrein et al., 2015* and this study) placed on the plot of values predicted by the stochastic model.

The following source data is available for figure 5:

**Source data 1.** Source files for simulated permutation intensities.

## The stochastic model leads to interpret previously unexplained sequences as higher order permutations

In both the previous analysis and in related works (*Besnard et al., 2014*; *Guédon et al., 2013*; *Landrein et al., 2015*; *Refahi et al., 2011*), the permutation detection was restricted to 2- and 3-permutations. However, the stochastic model potentially predicts the existence of higher order permutations, i.e. 4- and 5-permutations, in Arabidopsis thaliana especially for small values of the control parameter $\Gamma_P$. Following this prediction, we revisited the measured divergence angles in (*Landrein et al., 2015*) on the WS4 mutant grown in standard conditions ($\pi_2$= 29.45%, $\pi_3$= 14.91%) and for which 7.7% of angles were left unexplained when seeking for 2- and 3-permutations (*Figure 7A*). When higher order permutation are allowed in the detection algorithm (Appendix sections 4.1 and 5), most of the unexplained angles for WS4 can be interpreted as being part of 4- and even 5-permutations (*Figure 7B*).

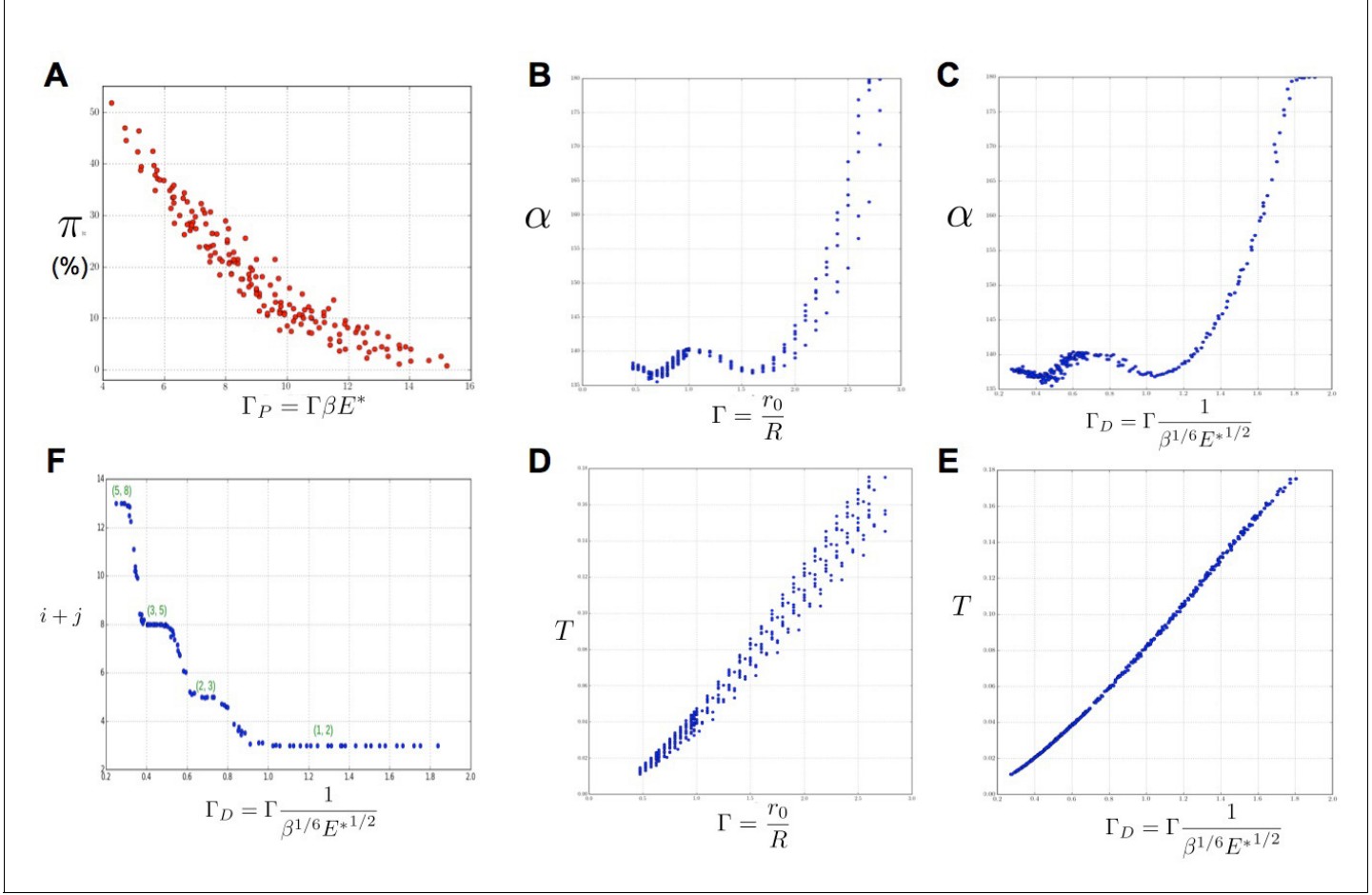

**Figure 6.** Key parameters controlling phyllotaxis phenotypes in the stochastic model. Phyllotaxis sequences were simulated for a range of values of each parameter $\beta$, $E^*$, $\Gamma$. Each point in the graph corresponds to a particular triplet of parameter values and represents the average value over 60 simulated sequences for this triplet. (A) Global amount of perturbation $\pi$ as a function of the new control parameter $\Gamma_P = \Gamma\beta E^*$. (B) Divergence angle $\alpha$ as a function of the control parameter $\Gamma$ of the classical model on the Fibonacci branch. (C) Divergence angle $\alpha$ as a function of the new control parameter $\Gamma_D = \Gamma\frac{1}{\beta^{1/6}E^{*1/2}}$ on the Fibonacci branch (here, we assume $s = 3$, see *Appendix 1—figure 6* for more details). (D) Plastochron $T$ as a function of control parameter of the classical model $\Gamma$. (E) Plastochron $T$ as a function of the new control parameter $\Gamma_D$. (F) Parastichy modes $(i,j)$ identified in simulated sequences as a function of $\Gamma_D$. Modes $(i,j)$ are represented by a point $i+j$. The main modes $(1,2)$, $(2,3)$ … correspond to well marked steps. (*Figure 5—source data 1*)

The following figure supplement is available for figure 6:

**Figure supplement 1.** New control parameter $\Gamma_D$ for divergence angle and plastochrons.

## The stochastic model predicts dynamic behaviors not yet observed

Previous observations (*Landrein et al., 2015*) point to the existence of positive correlations between meristem size and intensities of perturbations. As discussed above, the stochastic model explains this: if a mutation, or a change in growth conditions only affects the geometry of the system $\Gamma = r_0/R$, then $\Gamma_D$ and $\Gamma_P$ are both affected in the same sense (*Equation 6*). However, it also allows predicting new observable facts.

Assume that a mutation or a change in growth conditions affects the ability of the plant to perceive initiation signals without modifying the geometry of the system, i.e. $\beta$ and/or $E^*$ are modified while $\Gamma$ is left unchanged. Then, according to *equation 6*, this induces opposite variations in $\Gamma_D$ and $\Gamma_P$ that can be detected with the observed variables. For example, assuming that a mutation decreases the sensitivity of the system to the initiation signal (decrease of $E^*$), $\Gamma_P$ is decreased while $\Gamma_D$ is increased. The decrease of $\Gamma_P$ induces an increase of the perturbation intensity $\pi$ while the

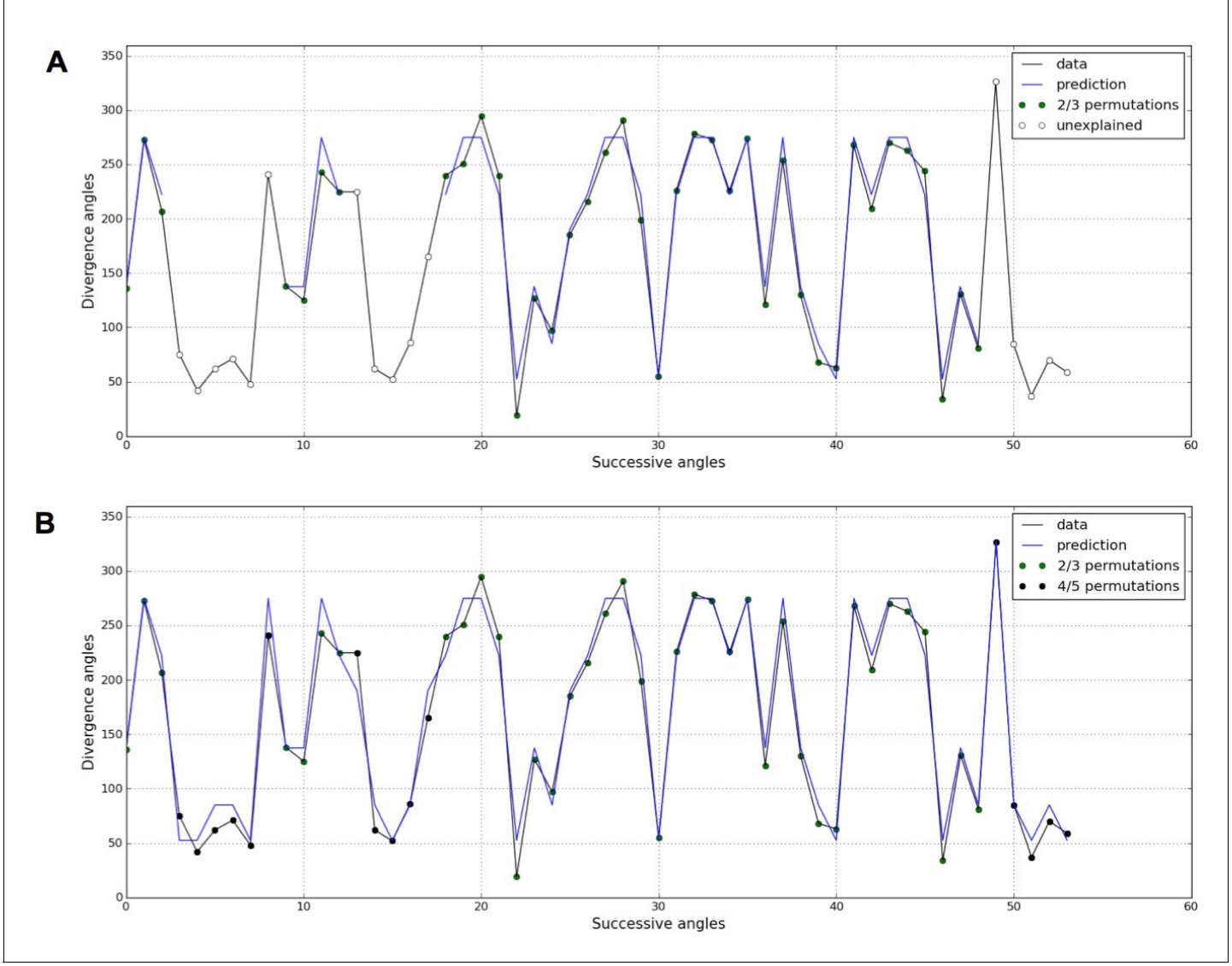

**Figure 7.** Detection of higher-order permutations in WS4. The detection algorithm (see ref [7] for details) searches plausible angle values, i.e. values within the 99% percentile given the Gaussian like distributions fitted in *Figure 1*, such that the overall sequence is *n*-admissible, i.e. composed of permuted blocks of length at most *n*. (**A**) When only 2- and 3-permutations are allowed, some angles in the sequences cannot be explained by (i.e. are not plausible assuming) permutations (the blue line of successfully interpreted angles is interrupted). (**B**) Allowing higher order permutations allows to interpret all the observe angles as stemming from 2-, 3- 4- and 5-permutations (the blue line covers the whole signal). Organs indexes involved in permutations: [3, 2], [5, 8, 6, 4, 7], [13, 12], [16, 14, 17, 15], [19, 18], [22, 21], [24, 25, 23], [27, 26], [30, 29], [32, 31], [35, 34], [39, 40, 38], [43, 42], [46, 45], [48, 49, 47], [52, 50, 53, 51].

increase of $\Gamma_D$ induces an increase of the plastochron. The model thus predicts that, in such a case, it is possible to expect an augmentation in the disorder correlated with a decrease in organ initiation frequency. To date, such a fact has not yet been observed and constitutes a testable prediction of our model.

## Discussion

We present here a multi-scale stochastic model of phyllotaxis driven by inhibitory fields and focusing on the locality of cellular decisions. A stochastic process models the perception of inhibitory fields by individual cells of the organogenetic domain and, at a higher scale, the initiation of primordia (*Figure 8A*). This process is continuous in essence and its results are independent of the time

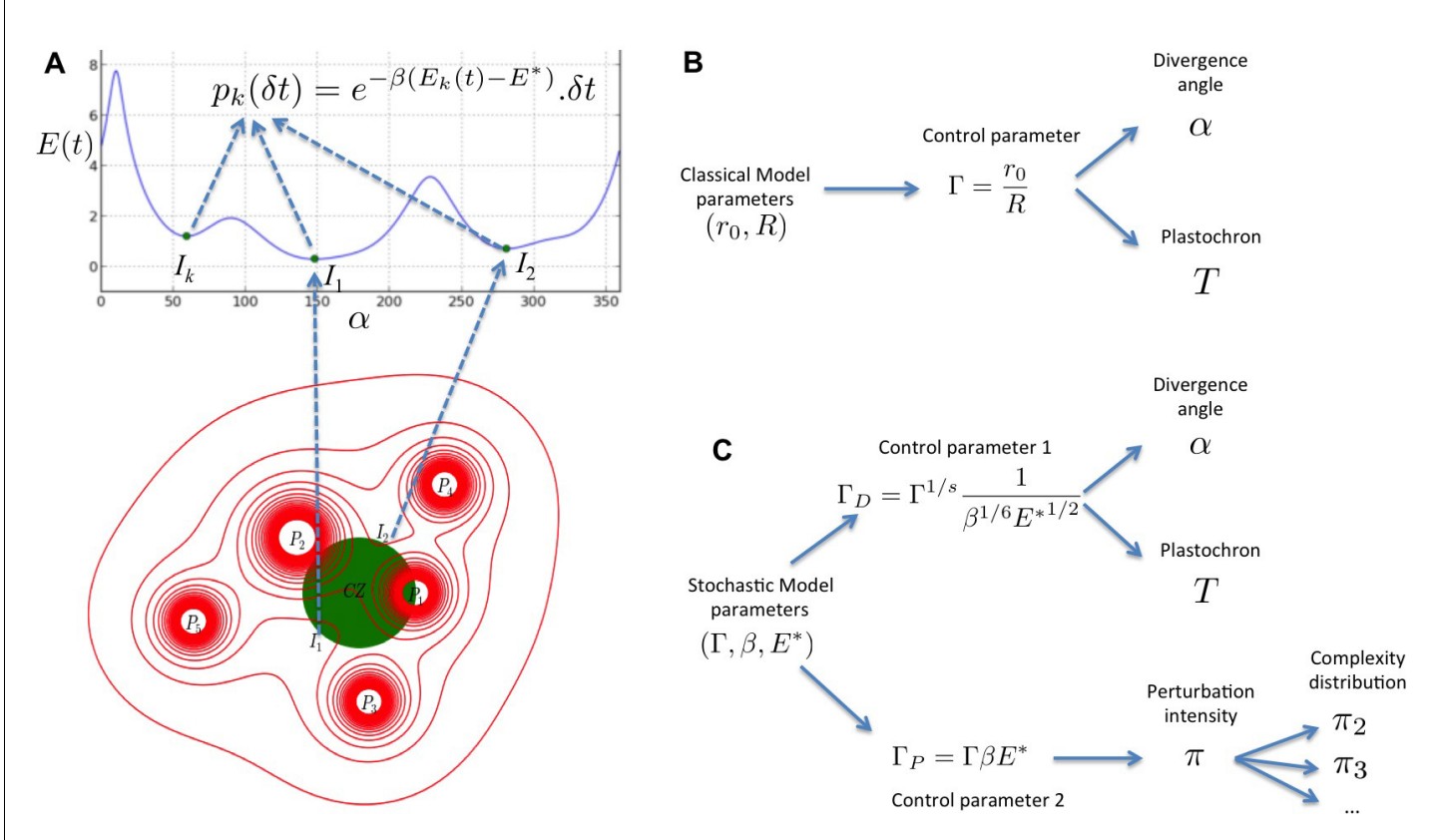

**Figure 8.** Structure of the stochastic model. (**A**) Inhibitory fields (red), possibly resulting from a combination of molecular processes, are generated by primordia. On the peripheral region of the central zone (CZ, green), they exert an inhibition intensity $E(t)$ that depends on the azimuthal angle $\alpha$ (blue curve). At any time $t$, and at each intensity minimum of this curve, a primordium can be initiated during a time laps $\delta t$ with a probability $p_k(\delta t)$ that depends on the level of the inhibition intensity at this position. (**B**) Relationship between the classical model parameters and its observable variables. A single parameter $\Gamma$ controls both the divergence angle and the plastochron. (**C**) Relationship between the stochastic model parameters and its observable variables. The stochastic model of phyllotaxis is defined by 3 parameters $\Gamma, \beta, E^*$. The observable variables $\alpha$, $T$ and $\pi, \pi_2, \pi_3 \dots$ are controlled by two distinct combinations of these parameters: $\Gamma_D = \frac{\Gamma^{1/s}}{\beta^{1/6} E^{*1/2}}$ controls the divergence angle and plastochron while $\Gamma_P = \Gamma \beta E^*$ controls the global percentage of permutated organs $\pi$, which in turns controls the distribution of permutation complexities: $\pi_2, \pi_3 \dots$

discretization chosen for the simulation. In contrast to previous models, the stochastic model does not use any inhibition threshold to decide either to produce an initiation or not. Instead, at each moment there is a non-zero probability to trigger initiation in any cell but this probability depends on the inhibition level in that cell, providing a realistic abstraction of the underlying signaling mechanism (**Figure 8A**).

## Noise on the timing as an intrinsic property of self-organization driven by lateral inhibitions

While the stochastic model is able to reproduce the major spiral and whorl phyllotaxis patterns, stochasticity induces alterations in the patterning process mainly affecting the plastochron i.e. the timing of organ initiation. These alterations take the form of permutations of the order of organ initiation in the meristem. If the plastochron is small as frequently observed in real sequences of permuted organs, permutations in the model can be considered equivalent to co-initiations that have been identified in the *Arabidopsis* meristem as the main source of permutations observed on the inflorescence stem. Less frequently, simulated permutations can have longer plastochrons. In this case, they can be interpreted as true permutations of the order of organ initiation in the meristem, consistently with the low frequency of such meristematic permutations observed also in *Arabidopsis* (***Besnard et al., 2014***). These results are in line with a previous attempt at introducing stochasticity

in the classical deterministic model that could also induce permutations (*Mirabet et al., 2012*). However in this latter work only a limited frequency of defects could be induced (even when the noise was fixed at high levels), while requiring time discretization and post-meristematic randomization of organ order when more than one organ initiation were detected in the same simulation time step. By contrast, the capacity of the stochastic model to reproduce faithfully perturbed sequences as observed in Arabidopsis indicates that the model captures accurately the dynamics of the phyllotactic system. Taken with the fact that permutations are observed in a variety of species and genotypes from a given species, our theoretical results identify noise on the plastochron as a common characteristic of phyllotaxis systems that may generate disorders in these developmental systems.

It is important to note that our work points at stochasticity in signaling mechanisms allowing perception of inhibitory fields as the most likely origin of this developmental noise but does not entirely rule out the idea that other phenomenon might contribute (see Appendix section 6). A major contribution of stochasticity in signaling is supported by the robustness of the model to changes in different assumptions and parameters. However we have observed that spatial discretization for example can modify the frequency of permutations in the model, although this effect is limited (see Appendix section 6.2). A possible interpretation is that changes in the size of cells could also contribute to a certain point to noise on the plastochron, an idea that could be further explored.

Importantly, phyllotaxis dynamics in the stochastic model relies not only on the geometry of the inhibitory fields captured by the $\Gamma$ parameter as in previous deterministic models (*Douady and Couder, 1996b*; *Richards, 1951*) (*Figure 8B*), but also on two new parameters $E^*$ and $\beta$ (*Figure 8C*). $E^*$ and $\beta$ describe respectively the sensitivity of cells to the inhibitory signal and the acuity of their perception, i.e. their capacity to differentiate close signal values. Our work thus suggests that a robust self-organization of a 3D developmental system driven by lateral inhibition depends both on the geometr of inhibitory fields in a tissue but also on the signaling capacities of cells in tissues. These theoretical observations are consistent with the key role of signaling in phyllotaxis (*Vernoux et al., 2011*), and with the setting of patterning dynamics in animal systems, downstream of morphogenetic signals (*Kutejova et al., 2009*). This pinpoints the interplay between global information provided by signal distribution and local interpretation of the information as a general principle for patterning emergence. In addition, we predict that due to pre-specification of initiation sites, noise in phyllotaxis is expected mainly on the timing of patterning. This might explain the selection through evolution of genetic mechanisms, such as the one recently described implicating the AHP6 protein (*Besnard et al., 2014*), to diminish noise on plastochron and disorders in phyllotaxis.

## Developmental disorders reveal biological watermarks

In biological systems, disorders are frequently viewed as a result of biological or environmental noise that mainly alters systems function or development. It is in this sense for instance that noise on phyllotaxis patterns had previously been analyzed (*Itoh et al., 2000*; *Jeune and Barabé, 2006*; *Peaucelle et al., 2008*). Here we show that biological noise at microscopic scale may be revealed at macroscopic scale in the form of organ disorders, the permutations. Our stochastic model of phyllotaxis suggests that these disorders bear information on the more profound, hidden variables that control the phyllotaxis patterning. Much like digital watermarks that represent a copyright or any information to be hidden in images or audio signals, the actual variables $(\Gamma, \beta, E^*)$ representing the state of the phyllotaxis system are not directly apparent in the plant phenotype (i.e. in the sequence of lateral organ angles and its dynamics). However, by scrutinizing carefully the image or, here, the phyllotaxis pattern and their perturbations with adequate decoding algorithms, it is possible to reveal the hidden information that was "watermarked" in the original signal. In this way, permutations together with divergence angle $\alpha$ and plastochron $T$ reveal key information about the state of the system that has produced them, as their knowledge drastically reduces the set of possible $\Gamma$, $\beta$ and $E^*$ values. Reciprocally, any experimental alteration of these values modifies the biological watermark. Our model suggests that this change is reflected in macroscopic alterations of the phyllotaxis patterns that convey information about their possible molecular origin.

To illustrate this, we used our stochastic model to confirm that changes in permutation frequencies due to changes in growth conditions are most likely explained by a specific modulation of $\Gamma$, as previously proposed (*Landrein et al., 2015*) (Appendix section 4.1.5). Such a biological watermarking also suggests a different interpretation of the function of AHP6 in phyllotaxis (*Besnard et al.,*

*2014*). Movement of AHP6 from organs has been proposed to generate secondary inhibitory fields that filter co-initiation at the meristem, decreasing the frequency of permutations. Based on permutation modifications, our theoretical framework suggests that AHP6 effect on the phyllotactic system does not need to be viewed as an additional specific mechanism acting on plastochron robustness but could be simply interpreted as a mechanism increasing $\Gamma$ slightly. As no differences in auxin-based inhibitory fields could be detected between *ahp6* mutant and wild-type plants (*Besnard et al., 2014*) (Appendix section 4.2), our model thus leads to a vision with composite inhibitory fields resulting at least from the combined effect of auxin-based and AHP6-based subfields. Conversely, this predicts that inhibitory fields in general cannot be explained only by auxin-based mechanisms as previously proposed.

Combined with data on divergence angles and on the plastochron, we further predict that the phyllotactic disorders could be used to identify mutants affected in biological mechanisms that controls $\beta$ and/or $E^*$. The model indicates that such mutations would have an opposite effect on frequency of permutations and on plastochron. Mutants behaving accordingly would allow not only to test this prediction of the model but also to dissect the molecular mechanisms at work. Precise and automated quantifications of the permutation, divergence angles and plastochron would allow for screening for such mutants and should become feasible with the fast development of phenotyping tools (*Dhondt et al., 2013*; *Granier and Vile, 2014*).

## Using stochastic models to understand multicellular development at multiple scales

Our model only takes into account stochasticity in the perception of inhibitory fields by cells and is based on two biologically plausible assumptions: that this perception is mostly cell autonomous and that it only depends on the local level of the inhibitory signal. This provides a reasonable abstraction of local stochastic fluctuations in *i*) hormonal concentrations related to inhibition produced by each primordium *ii*) in the activity of the signal transduction pathway leading to initium creation. The detailed molecular mechanisms controlling organ initiation are for the moment only partially known. However the capacity of the stochastic model to capture accurately phyllotaxis suggests that it also captures plausible emergent properties of the underlying molecular mechanisms. This model thus not only provides a framework to understand the dynamics of patterning in the meristem but also the properties of the signaling mechanisms that process the different signals involved. Note also that the predictive capacities of our model suggest that noise on perception could be the most influential source of noise in the system. However demonstrating this would require further exploration of other potential sources of stochasticity acting at different scales, such as growth variations, spatial discretization of the peripheral zone (to account for the real size of plant cells), in order to assess their relative contribution to disorders. Moreover, similarly to the classical deterministic model of phyllotaxis, our stochastic model does not explicitly account for the cascade of molecular processes that participate to the establishment of new inhibitory fields at the location of incipient primordia. This might limit the ability of these models to fully capture the dynamics of the self-organization of the system. To do so, more mechanistic versions of this stochastic model could be developed in the future, combining more detailed cellular models of hormone-based fields, e.g. (*Jönsson et al., 2006*; *Smith et al., 2006a*; *Stoma et al., 2008*), and stochastic perception of these hormonal signals in 2D or 3D models with cell resolution.

Heterogeneity of biological systems at all scale has attracted an ever-growing attention in the recent years (*Oates, 2011*). Deterministic models do not account for the high variability that can be observed in systems behaviors, indicating that they fail to capture some key characteristics of biological systems (*Wilkinson, 2009*). While more demanding computationally, stochastic models are required in such cases, *e.g.* (*Greese et al., 2014*; *Uyttewaal et al., 2012*; *Wennekamp et al., 2013*), and our work illustrates how dynamic stochastic modeling can help understanding quantitatively self-organization and more broadly patterning in higher eukaryotes.

## Material and methods

### Stochastic model formalization

Based on the classical model of phyllotaxis (Appendix section 2), a complete and formal presentation of the stochastic model is described in the Appendix section 3. In particular, it is shown how the exponential form of the intensity law can be derived from basic model assumptions and how different observable quantities can be expressed using the model parameters.

### Computational implementation of the stochastic model

A computational version of the stochastic model SMPmacro-max was implemented in Python programming language using Numpy and SciPy . Similarly to (*Douady and Couder, 1996b*), unless otherwise stated, the stiffness parameter was fixed in all simulations to $s = 3$. The non-homogeneous Poisson process was simulated using the algorithm described in (*Ross, 2012*). A pseudo-code version of the stochastic model algorithm is given in the Appendix section 3.2.

### Estimation of phyllotaxis variables

To estimate the value of the different variables characterizing phyllotaxis $\alpha$, $T$ and $\pi$, $\pi_2$, $\pi_3$, etc. in either simulated or observed sequences, we used a method based on the algorithm developed in (*Refahi et al., 2011*) and described in the Appendix section 4. As this algorithm is central to the identification of permutations, we additionally tested its ability to detect correctly permutations on synthetic data in which known permutation patterns were introduced (Appendix section 5). Results show that the algorithm is able to detect permutations with a success rate of 98% on average.

### Sensitivity analysis

The parameter space of the stochastic model was explored by varying values of $\Gamma$, $\beta$ and $E^*$. 60 stochastic runs have been made for each 3-tuple of the parameter values. Each simulation run generated a sequence of 25 divergence angles and corresponding plastochrons. The different observable variables have been extracted from these simulations. Results are reported in Tables 1 and 2 of the *Figure 5—source data 1*.

### Statistical models

The models describing the different non-linear relationships between the observable variables and the control parameters $\Gamma_D$ and $\Gamma_P$ were fitted with Gauss-Newton non-linear least-squares method (*Bates and Watts, 2007*). Approximate 95%-prediction bands of the response variables were computed by assuming random errors of the models independent and identically normally distributed.

## Acknowledgements

The authors would like to thank Olivier Hamant and Benoit Landrein for making original data from paper (*Landrein et al., 2015*) available to them, Arezki Boudaoud, Henrik Jönsson, Olivier Hamant and Jan Traas for their insightful comments on the manuscript, and to acknowledge funding from the following grants: HFSP grant RGP0054-2013 BioSensors (Human Frontier Science Program) to TV and CG, Inria Project Lab Morphogenetics and ANR-funded Institute of Computational Biology (IBC) to CG, Jan Traas's ERC Morphodynamics & INRA CJS to YR.

## Additional information

### Funding

| Funder | Grant reference number | Author |
| --- | --- | --- |
| Human Frontier Science Program | RGP0054-2013 | Teva Vernoux<br>Christophe Godin |
| Inria Project-Lab Morphogenetics | | Yassin Refahi<br>Teva Vernoux<br>Christophe Godin |

| | |
|---|---|
| ANR Institute of Computational Biology | Christophe Godin |
| ERC Morphodynamics | Yassin Refahi |

The funders had no role in study design, data collection and interpretation, or the decision to submit the work for publication.

## Author contributions

YR, Implemented the stochastic model, Conception and design, Acquisition of data, Analysis and interpretation of data, Drafting or revising the article; GB, Acquisition of data; EF, Conception and design, Analysis and interpretation of data; AJ-M, Conception and design; MP, Analysis and interpretation of data; TV, Conception and design, Analysis and interpretation of data, Drafting or revising the article; CG, Conception and design, Acquisition of data, Analysis and interpretation of data, Drafting or revising the article

## Author ORCIDs

Teva Vernoux, http://orcid.org/0000-0002-8257-4088
Christophe Godin, http://orcid.org/0000-0002-1202-8460

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

**Appendix 1**

## 1 Definitions related to phyllotaxis

Phyllotactic patterns emerge from the arrangement of organs on concentric circles, each circle corresponding to a "wave" of organ (or primordia) production, which is driven centrifugally by growth. Each circle contains the same number, called *jugacy*, of regularly spaced primordia, and is rotated with respect to the previous circle by a constant angle, called the *divergence angle*. Denoting the divergence angle by $\phi$ and the jugacy by $j$, phyllotactic patterns are traditionally classified as follows:

$$j=1: \begin{cases} \text{distichous, if } \phi = \pi \\ \text{spiral, if } \phi \neq \pi. \end{cases} \qquad j \geq 2: \begin{cases} \text{whorled, if } \phi = \pi/j, \\ \text{multijugate, if } \phi \neq \pi/j. \end{cases}$$

Note that to completely characterize a phyllotactic pattern, one also needs to specify the radii of the concentric "co-initiation" circles mentioned above. A first parameter needed for this is the radius of the *central zone*, a circular region at the centre of the apical meristem where no organ is ever produced. This radius is thus a lower bound for the radii of the co-initiation circles. Some regularity has been observed (or assumed) in these radii, and they often are specified thanks to a single additional parameter, which can be either

- the *plastochron*, defined as the time between two successive waves of organ formations, or

- the *plastochron ratio*, defined as the ratio between the radii of two successive co-initiation circles.

The use of these quantities as characteristic parameters implies that they are constant, which is only partially corroborated by the observation. In particular, permutations of organs correspond to irregularities of the plastochron. They allow to complete the description of "ideal" phyllotactic patterns such as those depicted in *Appendix 1—figure 1*.

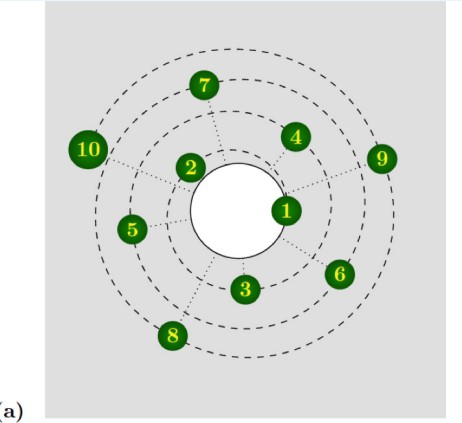 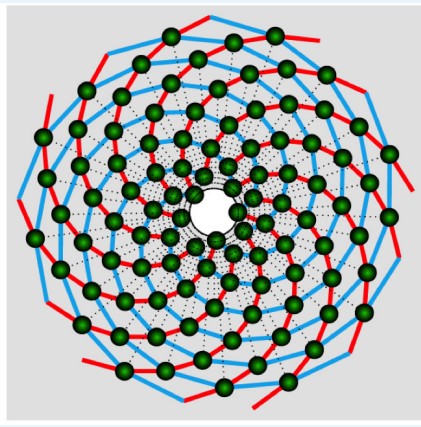

(a) (b)

**Appendix 1—figure 1.** Spiral phyllotaxis with divergence angle $\phi = 137.5°$ and different distributions of radial positions. (**a**) The first 10 primordia are depicted, with the generative spiral as a dashed line. The angle between successive organs is always equal to $\phi$. The nearest neighbours of primordium $i$ are $i+3$ and $i+5$, hence the mode of this pattern is $(3, 5)$. (**b**) 100 primordia are depicted, along with the 8 (resp. 13) parastichies oriented anticlockwise (resp. clockwise) indicated in cyan (resp. red), hence the mode of this pattern si $(8, 13)$.

In the spiral case (which is the most widespread in nature), as the name indicates successive organs are arranged along a *generative spiral*, see *Appendix 1—figure 1a*. In multijugate

cases, $j$ copies of this spiral are superimposed, each rotated by an angle multiple of $\pi/j$. When a sufficient number of organs are present, other spirals naturally occur to the eye. These spirals are formed by series of nearest neighbours, or second nearest neighbours. Such spirals are called *parastichies*. Because they are more apparent than the generative spiral, these parastichies are often used as characteristics of a phyllotactic pattern. More precisely, the parastichies formed by sequences of nearest and second nearest neighbours appear as spiralling outwards in opposite directions. One can thus count the number of spirals oriented in each direction, which determines a pair of integers $(p, q)$ called the *mode* of the phyllotactic pattern. A well-known observation is that the divergence angle is often close to the golden mean angle ($\approx 137.51°$), which leads to phyllotactic modes $(p, q)$ where $p$ and $q$ are successive numbers from the Fibonacci sequence $(F_n)_{n \geq 1} = (1, 1, 2, 3, 5, 8, 13, 21, ...)$ is defined by $F_1 = F_2 = 1$ and $F_{n+2} = F_n + F_{n+1}$. Other irrational numbers are also observed, giving rise to sequences closely related to the Fibonacci sequence. The most frequent pattern after Fibonacci is probably the so-called Lucas phyllotaxis, where $\phi \approx 99.5°$ and modes are successive terms in the Lucas sequence $(L_n)_{n \geq 1} = (2, 1, 3, 4, 7, 11, 18, 29, ...)$ is defined by $L_1 = 2$, $L_2 = 1$, and $L_{n+2} = L_n + L_{n+1}$.

To generate phyllotactic patterns two main mechanisms have been proposed. In 1868, Hofmeister proposed that a new primordium in generated periodically (the period being the plastochron defined above) in the largest available space on the central zone periphery. Later in 1962, Snow & Snow suggested that a new primordium is generated on the central zone periphery where and when there is available space, given the previous history of the system. In this view, phyllotaxis (and in particular the plastochron) *emerges* as a consequence of a local production rule.

Before describing models in more details, let us simply incorporate the notion of permutations in the description of phyllotaxis above. The permutations observed in plants and discussed in the text concern the order of organs along the generative spiral (hence they are defined for spiral phyllotactic patterns). We call $n$-permutation a series of $n$ organs whose order along the generative spiral is permuted. The order on the spiral corresponds to the order of appearance of primordia, and is materialized geometrically by the radial position which reflects the passage of time (see next section for more details). Hence, an $n$-permutation corresponds to a situation where $n$ successive organs as shown on *Appendix 1—figure 1a* have non-increasing radial positions: for instance, there is a $3$ permutation whenever the radial coordinates of organs $3$, $4$ and $5$ verify $r_3 > r_4$ and $r_3 > r_5$ (with the obvious notation). A more formal definition of this notion of $n$-permutations was published in (*Refahi et al., 2011*).

# 2 The classical model of phyllotaxis: a brief recap

## 2.1 Model description

We implemented the dynamical system introduced in (*Douady and Couder, 1996b*) based on the rules put forward by Snow & Snow for creation of a new primordium. According to these rules, the existing primordia inhibit primordia initiation in their vicinity. A new primordium is initiated at the periphery of the central zone of the meristem where and when the sum of the inhibition generated by pre-existing organs is below a threshold. In this dynamical system the plastochron, being an emergent property of the system and not an a priori parameter, is not necessarily a constant quantity.

In the simulations, the function that models the inhibition generated by primordium $q$ at sampling point $x$ on the central zone periphery decreases with the Euclidean distance $d(q, x)$ of $x$ from the center of primordium $q$ (by default we use polar coordinates for $x$ and $q$). Unless stated otherwise, we have used the function

$$E^{(q)}(x) = \left(\frac{r_0}{d(q,x)}\right)^s, \tag{1}$$

where $s$ denotes the stiffness of the inhibition and $r_0$ the radius of primordium $q$. Following (**Douady and Couder, 1996b**), we used $s = 3$ in all the simulations.

The total inhibition at a sampling point $x$ is the cumulative inhibition of pre-existing primordia $q \in Q$:

$$E(x, Q) = \sum_{q \in Q} E^{(q)}(x).$$

In the following the set $Q$ of pre-existing primordia will not be written explicitly when it is clear from the context or arbitrary. A new primordium appears at some point $x$ of the central zone periphery when $E(x) < \epsilon$, where $\epsilon$ is the inhibition threshold. In the simulations, we used $\epsilon = 1$ which guarantees that no primordium can be generated at a distance less than $r_0$ from the center of the primordia.

Because of growth, the primordia drift away radially from the central zone at a velocity proportional to their distance from the meristem center, i.e. $\frac{dr(t)}{dt} = \gamma r(t)$, (**Douady and Couder, 1996b**), leading to an exponential variation of the distance of primordia to meristem center:

$$r(t) = Re^{\gamma t}, \tag{2}$$

where $R$ is the radius of the central zone. The angular position of primordia, on the other hand, is supposed to remain constant in time.

In all generality, to simulate the dynamical emergence of phyllotactic patterns one needs a three-dimensional coordinate system of the form "time, radius, angle". However, with the choice of radial growth above, time and radial position are in fact equivalent coordinates, as detailed below, allowing us to represent the system using two coordinates only (time and angular position).

With the chosen radial growth, if there is a plastochron $T$ the plastochron ratio is constant, equal to $e^{\gamma T}$. The radial distance between successive organs is of the form $Re^t(e^{\gamma T} - 1)$ and is thus time dependent. The equation for $r(t)$ provides us with a bijection between the radial position and time of initiation of primordia. At time $t$ the radial position of a primordium that appeared at time $\tau_q$ is

$$r_q(t) = Re^{\gamma(t - \tau_q)}.$$

Reciprocally, a primordium with radial coordinate $r_q$ at time $t$ must have appeared at time

$$\tau_q = t - \frac{1}{\gamma}\ln\left(\frac{r_q}{R}\right).$$

The principle of our simulations is then to iteratively recalculate the positions of existing primordia for increasing time values, updating the value of inhibition functions at each time step and checking whether $E(x) < \epsilon$ for some coordinates $x$ at the boundary of the central zone. In simulations, primordia are represented by two coordinates: their date of initiation $\tau_q$ and their angular position $\theta_q \in [0, 2\pi)$. Note that by construction, at time $t = \tau_q$ we have $r_q = R$, i.e. primordia appear at the boundary of the central zone.

## 2.2 Initial conditions

Depending on the context, different archetypal initial conditions where used to run simulations. These initial conditions are given in terms of the coordinates of pre-existing primordia, which determine the initial value of the inhibition function $E$.

- Starting from cotyledons: one primordium is generated at the boundary of the central zone with a random azimuth $\theta_0$ (angular position). Then in the case of a dicotyledon plant, a second primordium is generated at the same distance of the apex center on the opposite side, i.e. at angle $\theta_0 + \pi$.

- Starting from a spiral pattern with divergence angle $\phi \in (0, 2\pi)$ and a constant plastochron $T > 0$. The radial and angular positions (used to calculate $E(x)$ ) of primordia are respectively of the form $Re^{nT}$ and $n\phi$ for successive integers $n$.

- Starting from a whorled pattern, with jugacy $j$ and constant plastochron $T > 0$. The radial and angular positions of primordia are respectively of the form $Re^{nT}$ and $\frac{n\pi}{j} + \frac{2k\pi}{j}$, where $0 \leq k < j$ spans the $j$ co-initiated primordia, and $n$ are successive integers spanning a finite number of co-initiation circles, or "whorls".

- Starting from a multijugate pattern of divergence angle $\phi$, jugacy $j$ and constant plastochron $T > 0$. The radial and angular positions of primordia are respectively of the form $Re^{nT}$ and $n\phi + \frac{2k\pi}{j}$, where $0 \leq k < j$ spans the $j$ co-initiated primordia, and $n$ are successive integers spanning a finite number of co-initiation circles.

## 2.3 Implementation

We implemented the classical model in Python programming language using NumPy and SciPy packages. A major issue in the classical model consists in deciding the moment and the place of the next initiation. In Snow and Snow simulations, time progression is continuous and the initiation time is unknown in contrast to the Hofmeister model where time periodicity is imposed. To simulate continuous time progression, small time steps $dt$ must be used, which may lead to markedly long simulation times.

To speed up the simulation time, we implemented a dichotomic initiation time search. Based on a initially large time step $\Delta t$, we first calculate the inhibition field every $\Delta t$ unit of time. Let $E_{min}$ denote the minimum value of inhibition at the periphery of the central zone at time $t$. Once the inhibition minimum at the periphery of the central zone is below the inhibition threshold, i.e. $E_{min} < \epsilon$, we take smaller time steps until the time precision $dt$ is reached. In the following, we present the pseudo code of the algorithm to generate $n$ primordia.

Here, the first primordium is generated at a random place on the periphery of the central zone. This can be replaced by other initial conditions as mentioned above.

**Classical Model of Phyllotaxis: Pseudo code**

```
Begin
    t ← 0
    Generate the first primordium randomly on the periphery of the central zone.
    Calculate the function E given this primordium.
    p ← 1
    while p<n :
        while E_min>ε :
            Calculate the primordia positions at t (radial growth).
            Calculate E given primordia positions.
            t ← t + Δt
        t₁ ← t − Δt
        while (t − t₁)>dt :
            t_mid ← t₁ + (t − t₁)/2
            Calculate the primordia positions at t_mid (radial growth).
            Calculate E given the primordia positions at time t_mid.
            if E_min>ε :
                t₁ ← t_mid
            else :
                t ← t_mid
        p ← p + 1
        Generate primordium at t on the periphery of the central zone where E = E_min
End
```

Note that not only time, but space also has to be discretized in the simulations. Formally, the boundary of the central zone is thus discretized into $K$ creation sites, corresponding to positions

$$C_k = (R, \theta_k) = \left(R, \frac{2k\pi}{K}\right), \quad 0 \le k \le K-1$$

in polar coordinates. Let $Q = Q(t)$ denote the number of pre-existing primordia at time $t$, and denote by $\tau_q$, the "date of birth" of primordium $q$ for $q \in \{1, ..., Q\}$. Primordia are supposed given in order of appearance, i.e.

$$\tau_1 \le \tau_2 \le ... \le \tau_Q.$$

Given $Q(t)$, we denote by $E_k(t)$ the value of the inhibition function at site $k$, i.e. with the previous notations

$$E_k(t) = E(C_k, Q(t)).$$

Then, we denote by $\kappa_q \in \{0, ..., K-1\}$ the creation site index at which primordium $q$ has appeared. The position of primordium $q$ at time $t \ge \tau_q$ is therefore

$$P_q = \left(Re^{\gamma(t-\tau_q)}, \theta_{\kappa_q}\right).$$

Now, the inhibition field at position $k$ is given more explicitly by the following expression:_

$$E_k(t) = \sum_{q=1}^{Q(t)} \left(\frac{r_0}{d(C_k, P_q)}\right)^s = \sum_{q=1}^{Q(t)} \left(\frac{r_0}{\sqrt{R^2(1 - e^{\gamma(t-\tau_q)})^2 + 2R^2 e^{\gamma(t-\tau_q)}(1 - \cos(\theta_k - \theta_{\kappa_q}))}}\right)^s,$$

using the Euclidean distance in polar coordinates. The expression can be simplified by introducing the geometric scaling parameter $\Gamma = r_0/R$:

$$E_k(t) = \Gamma^s \sum_{q=1}^{Q(t)} \left(1 + e^{2\gamma(t-\tau_q)} - 2e^{\gamma(t-\tau_q)} \cos(\theta_k - \theta_{\kappa_q})\right)^{-s2}. \tag{3}$$

Because $\Gamma^s$ appears as a global factor, the threshold parameter $\epsilon$ is defined up to a factor for any fixed choice of the stiffness $s$: for any $\rho > 0$ the exact same simulation would result from either $(\Gamma, \epsilon)$ or $(\rho\Gamma, \rho^s\epsilon)$. For this reason, we always fix $\epsilon = 1$ in the following.

## 2.4 Bifurcation diagram

Douady and Couder (*Douady and Couder, 1996b*) showed that the divergence angle and the phyllotactic mode reached by the classical model at equilibrium (if any) is controlled by the parameter $\Gamma$ appearing in the inhibition function, *Equation 3*. Similarly to their work, we observed stable spiral patterns, whose characteristics (divergence angle and mode) varied depending on $\Gamma$ and the initial condition.

To assess the consistency of our implementation with the study in (*Douady and Couder, 1996b*), we re-calculated the bifurcation diagram obtained by these authors. Current computational hardware allowed us to refine considerably the precision of the diagram. The variations of the divergence angles obtained by varying $\Gamma$ in the classical model form a characteristic tree like bifurcation diagram of phyllotaxis, shown in *Appendix 1—figure 2*.

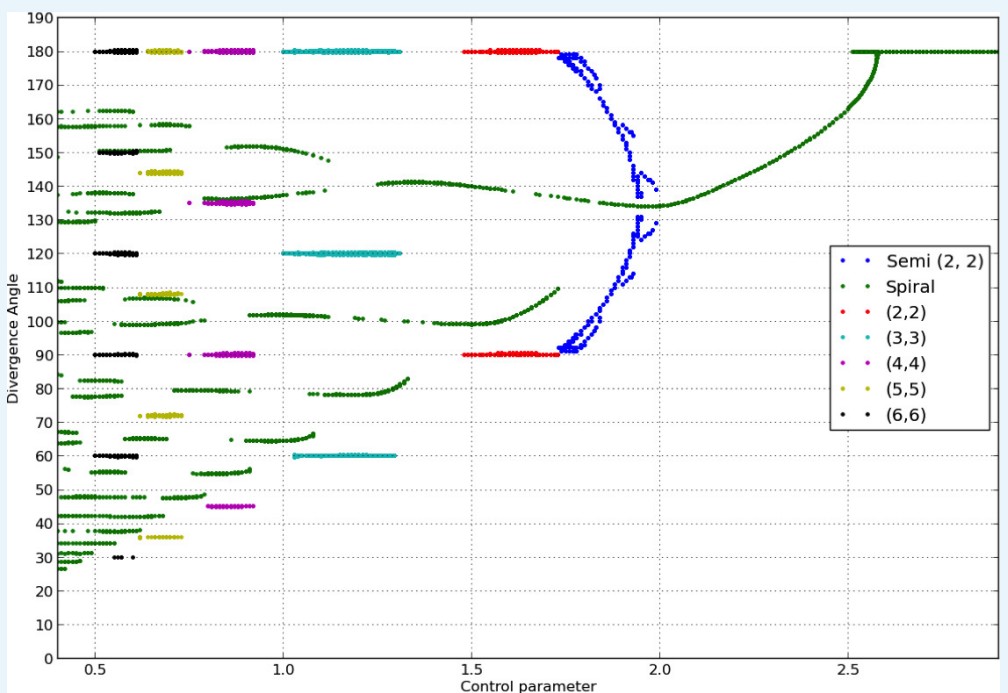

**Appendix 1—figure 2.** Bifurcation diagram in the Snow and Snow model. We used a sample of the interval $0.04 \leq \Gamma' \leq 2.9$ with steps of $0.01$ or less (refinements were performed in areas with higher numbers of branches). For each value of $\Gamma'$, we ran simulations of the classical model with (i) spiral initial conditions for divergence angles taking integer values in $[20°, 180°]$, and plastochrons taking values in a sample of 128 points between $0.05$ and $8.00$, (ii) whorled initial conditions for the same samples of divergence angles and plastochrons and all jugacies $2 \leq j \leq 7$. For each simulation, we estimated the final divergence angle $\phi$ and phyllotactic mode, and reported these in the graph above (abscissa: $\Gamma'$, ordinate: $\phi$, color code: mode).

Note that for consistency with (*Douady and Couder, 1996b*), the model had to be slightly altered compared to the description given in the previous section. First, the local inhibition function took the form below instead of a power function as in (*Bates and Watts, 2007*):

$$E^{(q)}(x) = \frac{-1 + (\tanh s \frac{d(x,q)}{r_0})^{-1}}{-1 + (\tanh s)^{-1}},$$

where $\tanh$ stands for hyperbolic tangent, and $s$ denotes the inhibition stiffness, $s$ was set to 8 in the simulations of the bifurcation diagram.

Secondly, the meristem was not considered as a flat domain but as being dome shaped. This was performed by using of a distance function of the form below, where $P_0$, $P_1$ are two points of coordinates $(r_0, \theta_0)$ and $(r_1, \theta_1)$ respectively

$$d(P_0, P_1) = \sqrt{\frac{(r_0 - r_1)^2}{N} + 2N r_0 r_1 (1 - cos(\theta_0 - \theta_1))},$$

where $1/N$ is called conicity (*Douady and Couder, 1996b*) and is equal to $3$ in *Appendix 1—figure 2*. This choice was only made for this particular situation, and in the remainder of the text all simulations were made with the model strictly as defined in the previous section. The expression above does not allow to factor $\Gamma$ out as in (*Cinlar, 1975*). Instead, following (*Douady and Couder, 1996b*) the new control parameter below was used:

$$\Gamma' = \frac{d_0}{N^{1/2} R_0}.$$

The caption of *Appendix 1—figure 2* describes the default sampling of parameters and initial conditions. Note for complete precision that for regions including many branches, additional sampling points were calculated a posteriori to reduce the number and width of gaps. Also, for small values of $\Gamma'$ were simulations are more expensive computationally, higher plastochron values ($T \geq 3$) were not included. Each initial condition comprised $150$ pre-defined primordia and was followed by $250$ primordia generated by the simulation algorithm. To assess the convergence to a stable phyllotactic patterns, the last $50$ divergence angles were considered. If all these $50$ divergence angle were within $5°$ from their average, the pattern was considered steady.

## 3 Stochastic model

### 3.1 Model description and derivation

In the stochastic model, we modify from the classical model the rule driving primordia initiation. Instead of a deterministic and global threshold value $\epsilon$, we assume that each cell is able to decide in a cell autonomous manner to trigger initiation of not. This decision is not taken on the basis of a threshold being reached or not, but on a local probability to trigger initiation at this cell site. This probability depends naturally on the inhibition level $E_k(t)$, *Equation 3* (the lower the inhibition, the higher the probability value) and on the amount of time one waits for an initiation at this site $k$ (the longer one waits, the higher the probability).

If $X_k(t, \delta t)$ denotes the number of cells initiated at a location $C_k$ in the time interval $[t, t + \delta t]$ ($X_k(t, \delta t)$ will typically take values $0$ or $1$, as a cell can initiate at most one primordium at a time), the probability of observing one initiation during a small time interval of length is assumed nearly proportional to $\delta t$:

$$P(X_k(t,\delta t)=1)=\lambda(E_k(t)).\delta t+o(\delta t)=\lambda_k(t).\delta t+o(\delta t), \tag{4}$$

where $\lambda_k(t)=\lambda(E_k(t))$ is a rate parameter that can be interpreted as a temporal density of events (and $o(\delta t)/\delta t \to 0$ as $\delta t \to 0$). The function $\lambda$ expresses the ability of any cell to respond to the inhibitory signal. If we assume in addition that probabilities to observe an initiation over non-overlapping time-spans are independent, the previous expression (4) defines an *non-homogeneous Poisson process* (**Cinlar, 1975**; **Ross, 2014**) of rate $\lambda_k(t)$.

## Derivation of the $\lambda$'s expression

As explained above, the function $\lambda$ must obviously decrease with the inhibition level at site $k$. We assume in addition that the ratio between probabilities at any two different sites $k$ and $l$ only depends on the difference between their respective inhibitory energies:

$$\frac{P(X_k(t,\delta t)=1)}{P(X_l(t,\delta t)=1)}=\frac{\lambda(E_k(t))}{\lambda(E_l(t))}=f(E_k(t)-E_l(t)), \tag{5}$$

where $f$ is an unknown function to be determined. Based on the previous assumptions, it is possible to find an explicit expression for $\lambda$ as a function of the inhibition. For sake of notation simplicity, let us omit the time dependence $t$ in the following expressions.

For sites $k$ and $l$ with the same energies, the rates $\lambda(E_k)$ and $\lambda(E_l)$ must be equal: $\frac{\lambda(E_k)}{\lambda(E_l)}=1=$ $f(0)$. Taking the specific value $E_l=0$, **Equation 5** becomes: $\frac{\lambda(E_k)}{\lambda(0)}=f(E_k)$, which by denoting $\alpha=\lambda(0)$, leads to:

$$\lambda(E_k)=\alpha.f(E_k). \tag{6}$$

Therefore from **Equation 5**, we deduce: $\frac{\lambda(E_k)}{\lambda(E_l)}=\frac{f(E_k)}{f(E_l)}=f(E_k-E_l)$, and by adding $E_l$ to argument $E_l$ in the above expression, we get:

$$f(E_k+E_l)=f(E_k).f(E_l),$$

showing that, if $f$ is continuous, $f$ must have the form of an exponential. Taking in addition that $f(0)=1$ and $f$ is decreasing, we get:

$$f(x)=e^{-\beta.x}$$

where $\beta$ is a positive scalar parameter. Finally, by replacing the above expression of $f$ in the expression of $\lambda$ of **Equation 6** and denoting $E^*=\frac{\ln\alpha}{\beta}$, we get the final expression of $\lambda$:

$$\lambda_k(t)=\lambda(E_k(t))=e^{-\beta(E_k(t)-E^*)}, \tag{7}$$

where $\beta$ and $E^*$ are two real parameters, with $\beta>0$. The probability of an initiation during a small time lapse $\delta t$ is thus:

$$P(X_k(t,\delta t)=1)=\lambda_k(t).\delta t \tag{8}$$

From the theory of non-homogeneous Poisson processes (**Cinlar, 1975**; **Ross, 2014**), we have that the number of initiations is also a Poisson random variable, with rate

$$\mu_k(t, \Delta t) = \int_t^{t+\Delta t} \lambda_k(u) du, \tag{9}$$

i.e. $\mu_k(t, \Delta t)$ is the mean number of initiations between $t$ and $t + \Delta t$ at site $k$.

Independent Poisson processes can be superposed to yield another Poisson process, and the instantaneous rates are additive. If $V \subset \{0 \ldots K-1\}$ is any set of creation sites (not necessarily contiguous; including the set $\{0 \ldots K-1\}$ itself), the global creation process for sites of $V$ is a non-homogeneous Poisson process with instantaneous rate

$$\lambda_V(t) = \sum_{k \in V} \lambda_k(t).$$

For $V = \{0 \ldots K-1\}$ in particular we denote

$$\lambda(t) = \sum_{k=0}^{K-1} \lambda_k(t).$$

This additivity carries over to the average rate functions, giving

$$\mu(t, \Delta T) = \sum_{k=0}^{K-1} \mu_k(t, \Delta T), \tag{10}$$

and $\mu_V(t, \Delta T)$ is defined similarly.

The creation process at a given initiation site is a non-homogeneous Poisson process as just described, but its rate is actually stochastic and depends on all the primordia which have been previously (randomly) created. Properly speaking, the process is *conditionally* (on previous creation events) a Poisson process. More precisely, from the properties of non-homogeneous Poisson processes we can derive the probability of occurrence of an initiation at site $k$, *conditionally on the previous creation events*, which with our notations are encoded by their times and sites of appearance, $\tau_q$ and $\kappa_q$ respectively (for $1 \leq q \leq Q = Q(t)$). The next initiation is the $(Q+1)^{th}$ organ and, at time $t \geq \tau_Q$, the probability that it occurs at site $k \in \{0 \ldots K-1\}$ can be written as:

$$P(\kappa(Q+1) = k \,|\, E(\tau_Q)) = \int_0^\infty \lambda_k(t+u) \, e^{\mu(\tau_Q, u)} du, \tag{11}$$

where we write the conditioning using $E(t) = (E_k(t))_{0 \leq k \leq K-1}$ for brevity.

In the classical model, the inhibition function (3) is directly compared to the threshold $\epsilon = 1$, so that all the parameters are those appearing in (3), i.e. $\Gamma$, $\gamma$ and the initiation times $\tau_q$, which are emerging from the model's dynamics and are thus not parameters strictly speaking. Note also that $\gamma$ is essentially a "time unit" parameter and is thus not crucial to the geometry of phyllotactic patterns (in other words, a change of time scale with a factor $\gamma$ would allow us to remove this parameter from **Equation 3**). In summary, $\Gamma$ is truly the only control parameter in the standard, deterministic model.

In the stochastic formulation above, the $\Gamma$ dependent inhibition $E_k$ is used via the rate $\lambda$, **Equation 7**. This introduces the two additional parameters $\beta$ and $E^*$. As the $\lambda$ is used in the integral expressions (10) and (11), there is no obvious way to reduce the dimensionality of the control space, with its coordinates $(\Gamma, \beta, E^*)$. See the main text for a heuristic approach, where some lower dimensional parametrizations are obtained empirically.

## Probability of $n$-permutations

A number of theoretical derivations can be carried out from the model. In particular, some properties of the probability that an $n$-permutation occurs can be related to the model parameters. For simplicity we focus on 2-permutations. Let us assume that previous initiations have occurred close to a "normal" spiral, with no permutations at least in the most recent organ creations. Without loss of generality, the last organ creation is supposed to have occurred at location $\theta_0 = 0$. According to our simulations, the minimum and second minimum of the inhibition profile $\{E_k(t) \mid 0 \leq k \leq K-1\}$ are then typically situated at the "next" positions along the spiral, i.e. at sites $k_1$ and $k_2$ respectively very close to $\phi$ and $2\phi$, with $\phi$ the divergence angle. Then, a 2-permutation happens if the next organ creation takes place at site $k_2$ rather than $k_1$. Conditionally on the next initiation taking place at one of the two sites $k_1$, $k_2$, **Equation 11** gives the probability of $k_2$ occurring first as

$$\Pi_2(t) = \int_0^\infty \lambda_{k_2}(t+u) \exp\left(-\mu_{k_1}(t,u) - \mu_{k_2}(t,u)\right) du.$$

Similarly, the (conditional) probability that the next creation occurs at site $k_1$ is given by replacing $\lambda_{k_2}$ by $\lambda_{k_1}$ in the integral above. Since, as just mentioned, we observe that typically $E_{k_1} < E_{k_2}$ and the intensity $\lambda$ is a decreasing function of $E$, we deduce that the next initiation is more likely to occur at site $k_1$, i.e. without a 2-permutation.

These expressions make it possible to investigate the variation of probability due to a change in model parameters. If for example the sensitivity of the system $E^*$ is increased by $\delta E^* > 0$, how does the 2-permutation probability vary? Remembering from (7) that we can write $\lambda_k = e^{\beta E^*} e^{-\beta E_k}$, we obtain

$$\Pi_2(t) = e^{\beta E^*} \int_0^\infty \exp(-\beta E_k(t+u) - \mu_{k_1}(t,u) - \mu_{k_2}(t,u)) du.$$

This indicates that an increase in the sensitivity $E^*$ will tend to increase $\Pi_2$. However things are more complicated, since from (9) we have $\mu_k(t,u) = e^{\beta E^*} \int_t^{t+u} e^{-\beta E_{k_2}(u)} du$ and thus the terms $\mu_{k_1}$ and $\mu_{k_2}$ in the integral above also depend on $E^*$, as below:

$$\Pi_2(t) = e^{\beta E^*} \int_0^\infty \exp\left(-\beta E_{k_2}(t+u) - e^{\beta E^*}\left(\int_t^{t+u} e^{-\beta E_{k_1}(u)} du + \int_t^{t+u} e^{-\beta E_{k_2}(u)} du\right)\right) du.$$

Hence, despite the first factor being an increasing function of the sentitivity $E^*$, it may happen in general that the probability of a 2-permutation decreases as the sensitivity of the system is increased, through the non-trivial formula above. To proceed further, the relationship between $E^*$, $\beta$, $\Gamma$ and permutations is studied empirically using simulations.

## 3.2 Implementation

We developed two different versions of the model.

- In the first version, called SMPmicro, every cell on the peripheral zone is considered as a potential initiation site and therefore the Poisson processes are simulated for each cell.

- In the second version of the model, SMPmacro-max, the local minima indicate the potential initiation sites and therefore the Poisson processes are simulated at these sites assuming that a primordium is triggered if at least one of the cells spanned by the local minimum valley switches its identity to primordium.

The models are implemented using the Python programming language using the NumPy and SciPy packages. In the following, we present the pseudo code of the model to generate $n$ primordia. We use the non-homogeneous Poisson simulation algorithm as presented in

(**Ross, 2012**). To encompass both versions above in a single description, we use the notation $S$ to describe either the whole set $\{0, ..., K-1\}$ if all sites are tested, or the subset $S \subset \{0, ..., K-1\}$ of all local minima of the function $E_k$ for the second version.

The variables appearing in the pseudo-code below are:

$n$: number of primordia generated by simulation

$t$: system time

$p$: current number of primordia

$\Delta t$: time step

$E_k$: inhibition level at sampling point $k$

$U(0, 1)$: the uniform distribution between 0 and 1

$\beta$ and $E^*$: parameters of the model

**Stochastic Model of Phyllotaxis: generic SMP Pseudo code**

```
Begin
    t ← 0
    Generate the first primordium randomly on the central zone boundary.
    Calculate the function E given this primordium.
    p ← 1
    while p<n :
        Compute the primordia positions at t (radial growth).
        Compute E_k(t) for k ∈ {0,...,K−1}
        initiation = false
        for k ∈ S : ▶ simulate non-homogenous Poisson process at each cell (or each local minimum)
            δ_k ← 0
            while true :
                Draw r ~ U(0,1)
                δ_k ← δ_k − ln(r)/(e^{βE^*})
                if δ_k > Δt :
                    break
                Compute primordia positions at t (radial growth).
                Compute E_k(t).
                Draw s ~ U(0,1)
                if s ≤ e^{−βE_k} :
                    initiation = true
                    break
        if initiation :
            t ← t + min{δ_1,...,δ_k}
            i = argmin{δ_1,...,δ_k}
            Generate a new primordium at cell number i (or local minimum)
            p ← p + 1
        else :
            t ← t + Δt
End
```

In the implementation, the simulation of non homogeneous Poisson processes in cells (or in local minima) is parallelized. This speeded up the simulations and allowed us to carry out an extensive sensitivity analysis.

# 4 Interpreting phyllotaxis phenotypes using the stochastic model

## 4.1 Parameter estimation

This section details how values for the control parameters $\Gamma_D$ and $\Gamma_P$ of the stochastic model were estimated from observed data for various genotypes or growth conditions. When similar data can also be produced by simulations (e.g. divergence angle sequences, permutations) the same estimation procedures are applied to the simulated data.

### 4.1.1 Permutation intensities

To estimate the value of the different permutation intensities, $\pi, \pi_2, \pi_3, \ldots$, in either simulated or observed sequences, we used the algorithms described in **Refahi et al. (2011)**. The algorithms allows identification of permutation patterns of bounded length (at most $n$ for a fixed $n \geq 2$) on spiral patterns with different canonical divergence angle values (e.g. $\varphi = 137.5$ for Fibonacci phyllotaxis, $\varphi = 99.5$ for Lucas phyllotaxis). The algorithm can detect truncated permutation patterns at the end of the sequences. It also allows the analysis of the sequences in the reverse direction. Most importantly in this study, it allows to characterize permutations in sequences of angles which are not strictly equal to multiples of $\varphi$, but are "close" to these, as happens in real experimental data or our stochastic simulations.

More precisely, let $g(x; \mu_q, \kappa)$ denote the probability density function of von Mises distribution (the standard circular analogue of a Gaussian distribution) with parameters $\mu_q \in D_n$ (mean direction) and $\kappa$ (concentration parameter), where $D_n$ is the set of the theoretical angles (multiples of $\varphi$ occurring with permutations of length at most $n$). We used $\kappa = 10.4$ which was estimated in (**Guédon et al., 2013**) by fitting von Mises distributions on experimental data from *A. thaliana*. For a measured (simulated ) angle $x_i \in [0, 360)$ and for a candidate theoretical angle, $\mu_q \in D_n$ we calculated the posterior probability :

$$\omega(x_i, \mu_q) = \frac{g(x_i; \mu_q, \kappa)}{\sum_{\mu_r \in D_n} g(x_i; \mu_r, \kappa)}$$

The permutation detecting algorithm seeks candidate angles in $D_n$ such that the sequence in composed of permutations, but only accepts them if the posterior probability is higher than a threshold whose value was set to $0.001 \times |D_n|$. This threshold was determined empirically based on sequences whose permutation patterns were confirmed by a human expert. The factor $|D_n|$ stands for the implicit hypothesis that the theoretical angles were a priori equally probable.

### 4.1.2 Average divergence angles

For a given sample of divergence angle sequences (i.e. repetitions of the same genotype in the same growing conditions, repetitions of stochastic model simulation for a given set of parameters $\Gamma, E^*, \beta$), divergence angles were estimated as the mean value of the divergence angle over the total set of divergence angles in the whole set of sequences. The estimation takes into account potentially detected permutations (meaning that, in case of permutation detection, organs are re-ordered in the sequences to remove the permutations prior to estimating average angles).

### 4.1.3 Plastochrons

In some cases plastochrons were directly estimated from counting initiations of organs over different time spans. In such cases plastochrons are estimated in terms of hours (see table below). However, to compare the experimental plastochron with simulated ones, we rather estimated plastochrons ($T$) from experiments from plastochron ratios ($\rho$), themselves estimated in (**Besnard et al., 2014**; **Landrein et al., 2015**) from measurements of ratios between distances of consecutive primordia. According to section 1, the plastochron ratio $\rho = e^{\gamma T}$ is constant for a constant plastochron $T$. Corresponding plastochrons $T$ can thus be estimated

(in units corresponding to that of the model parameter $\gamma$) from measured plastochron ratios as:

$$T = \frac{\ln \rho}{\gamma}, \qquad (12)$$

where $\gamma$ is a constant fixed in the model to $5s^{-1}$. For the wild type for example, the measure plastochron ratio is $\rho = 1.20$, leading to an estimated plastochron $T = 0.0227$ expressed in the time units defined in the simulation.

### 4.1.4 Control parameters of the stochastic model

Using the characteristic curves of **Figure 5A,C,E** (main text), it is possible to estimate values or ranges of values for the stochastic model control parameters from values of the observable phyllotaxis variables: $\pi$, $\alpha$ and $T$.

Let us illustrate the estimation method on the wild type. A value of $\Gamma_P$ can be estimated from a parametric regression of the could of points obtained from the stochastic model simulations **Appendix 1—figure 3A**. The estimated total permutation intensity is $\pi = 14.20\%$, which leads to a possible value of $\Gamma_P$ in $[8.30, 11.49]$ according to the curve of **Appendix 1—figure 3A**.

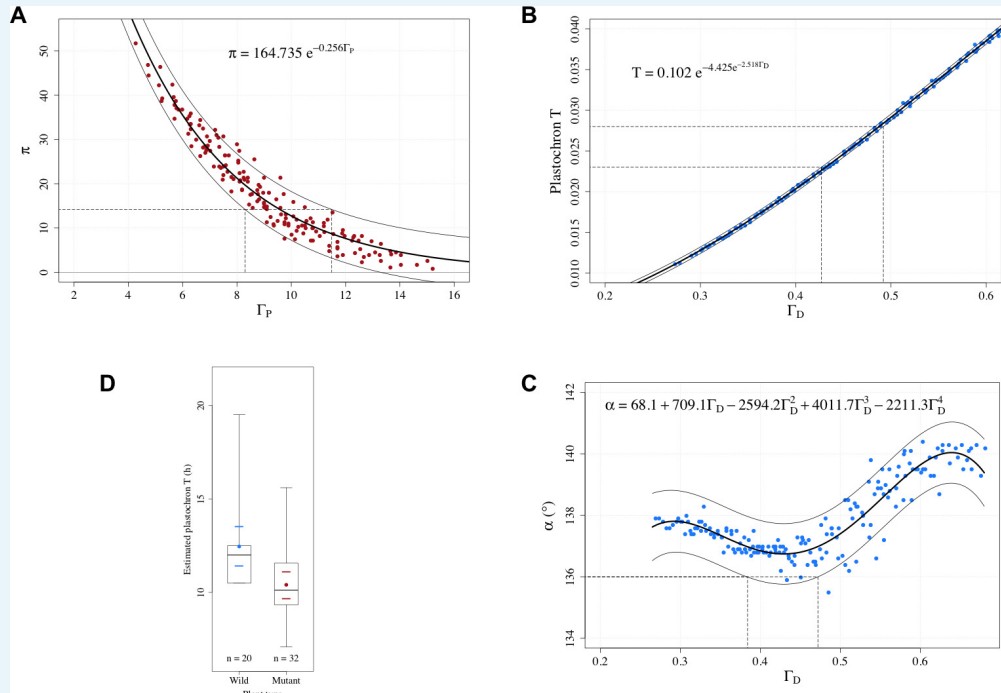

**Appendix 1—figure 3.** Estimation of control parameters from observable phyllotaxis variables. (**A**) An exponential model was fitted to the simulated data of $\pi$ and $\Gamma_P$ using the Gauss-Newton least squares method (**Bates and Watts, 2007**); for the fitted model, an approximate 95% prediction band was then computed by assuming the random error terms additive and i. i.d. normally distributed. The range of possible $\Gamma_P$ values $[8.30, 11.49]$ that could yield the observed $\pi$ value 14.2 was determined by the prediction band. (**B**) A Gompertz function was fitted to the simulated data of plastochron and $\Gamma_D$ using the Gauss-Newton least squares method (**Bates and Watts, 2007**); for the fitted model, an approximate 95% prediction band was then computed by assuming the random error terms additive and i.i.d. normally distributed. The range of possible $\Gamma_D$ values $[0.427, 0.492]$ that could yield the observed range of plastochron values $[0.023, 0.028]$ was determined by the prediction band. (**C**) A 4th degree polynomial was fitted to the simulated data of angle and $\Gamma_D$ using the least squares method; for the fitted model, an approximate 95% prediction band was then computed by assuming

the random error terms additive and i.i.d. normally distributed. The range of possible $\Gamma_D$ values $[0.384, 0.472]$ that could yield the observed angle value of 136° was determined by the prediction band. (**D**) Distributions of the estimated plastochrons in the groups of wild-type and mutated Arabidopsis plants in the experiments of **Besnard et al., 2014**. The box depicts the inter-quartile range bisected by the median, and the whiskers reach out to the extreme values in the group; the colored point denotes the arithmetic mean, and the colored dashes indicate twice the standard error of the mean; $n$ stands for the number of plants in the group.

We then estimate the plastochron as explained in the preceding section. For the wild type, $T \in [0.023, 0.0227]$ (**Landrein et al., 2015**). By using a parametric regression of the simulated cloud of points ($\Gamma_D, T$ predicted by the stochastic model, **Appendix 1—figure 3**, we derive an estimate value of $\Gamma_D$ in $[0.427, 0.492]$. Now using the curve of **Appendix 1—figure 3**, and the average divergence angle of 136° measured for the WT (**Besnard et al., 2014**), we deduce that $\Gamma_D$ should be included in $[0.384, 0.472]$. These two sets of estimates for $\Gamma_D$ are consistent with each other and lead to a consolidated estimate for its value of $\Gamma_D$ in $[0.427, 0.472]$.

Similar estimations for $\Gamma_D$ and $\Gamma_P$ of the wild-type and other plants are reported in the next section.

### 4.1.5 Estimated values

The following tables summarize estimated observable variables ($\alpha, T, \pi, \pi_2, \pi_3$) based on data from (**Besnard et al., 2014**; **Landrein et al., 2015**) and the inferred model control parameters $\Gamma_P$ and $\Gamma_D$. Two different day length conditions were used to affect the size of the meristem of different phenotypes and mutants (**Landrein et al., 2015**). In the first setting, plants were first grown in short-day conditions for one month after germination and then passed in long-day conditions, **Appendix 1–table 1**. In a second setting, plants were growth only in long-day conditions from germination, **Appendix 1–table 2**. To complete the published results, we grew *ahp6* plants in long day conditions as in (**Landrein et al., 2015**). Results are reported in the last column of **Appendix 1–table 2**.

**Appendix 1-table 1.** Observed phyllotaxic variables on plants grown in short day and then in long day conditions. % permuted organs is shown as a number followed by 2 other numbers in parentheses, i.e. $\pi_{2,3}(\pi_2, \pi_3)$

| Short day - Long day | Col0 | WS4 | *clasp*-1 | *ahp6* |
|---|---|---|---|---|
| #angles/#sequences | 704/29 | 1046/25 | 619/17 | 2815/89 |
| Estimated # 2-permutations | 49 | 154 | 28 | 297 |
| Estimated # 3-permutations | 2 | 52 | 1 | 53 |
| % unexplained angles | %2.7 | %7.7 | %1.1 | %2 |
| Estimated % permuted organs | 14(13.37,0.82) | 43.3(29.45,14.91) | 9.2(8.81, 0.47) | 25.9(20.45,5.47) |
| Estimated average $\tilde{\alpha}$ | 136.4 | 136 | 138.0 | - |
| Estimated $\tilde{\alpha}$ standard deviation | 5.5 | 2.3 | 6 | - |
| # Lucas sequences | 0 | 0 | 0 | 0 |
| Average measured plastochron (h) | 7.5 | 5.1 | 19 | 10.6 |
| Plastochron ratio | 1.12 | 1.08 | 1.135 | - |
| Estimated plastochron (equ. 12) | 0.0227 | 0.0154 | 0.0253 | - |
| Estimated $\Gamma_D$ | 0.430 | 0.325 | 0.450 | - |
| Estimated $\Gamma_P$ (in $[\Gamma_{Pmin}, \Gamma_{Pmax}]$) | [8.7,11.4] | [5.2,5.8] | [9.6,12.3] | [6.7,8.7] |

**Appendix 1–table 2.** Observed phyllotaxic variables on plants grown in long day conditions only. % permuted organs is shown as a number followed by 2 other numbers in parentheses, i.e. $\pi_{2,3}(\pi_2, \pi_3)$.

| Long day only | Col0 | WS4 | *clasp*-1 | *ahp6* |
|---|---|---|---|---|

*Appendix 1–table 2 continued on next page*

| #angles/#sequences | 1193/34 | 487/15 | 667/19 | 965/27 |
|---|---|---|---|---|
| Estimated # 2-permutations | 13 | 22 | 14 | 22 |
| Estimated # 3-permutations | 0 | 1 | 1 | 0 |
| % unexplained angles | %2.6 | %3.6 | %4.3 | %0.1 |
| Estimated % permuted organs | 2.12(2.12,0) | 9.3(8.76, 0.6) | 4.52(4.08, 0.44) | 4.4(4.44,0) |
| Estimated average $\tilde{\alpha}$ | 136.4 | 141.4 | 128.1 | - |
| Estimated $\tilde{\alpha}$ standard deviation | 10.7 | 6.8 | 20.46 | - |
| # Lucas sequences | 0 | 0 | 6 | 0 |
| Average plastochron (h) | 10 | 6.3 | 21 | 9.5 |
| Plastochron ratio | 1.15 | 1.115 | 1.145 | - |
| Estimated plastochron (equ. 12) | 0.0280 | 0.0218 | 0.0271 | - |
| Estimated $\Gamma_D$ | 0.485 | 0.410 | 0.475 | - |
| Estimated $\Gamma_P$ (in $[\Gamma_{Pmin}, \Gamma_{Pmax}]$) | [12.8,15.1] | [10.0,12.0] | [11.7,14.2] | [11.8,14.3] |

## 4.2 In comparison to wild-type, plastochrons are reduced in *ahp6*

The stochastic model suggests that a comparison between plastochrons of the wild type and the mutant plants would be informative about the potential influence of the mutation on key phyllotactic geometrical or biochemical processes (see main text). We tested this idea on *ahp6* mutants. For this, we re-analysed the measurements of number of organ initiations over different time-laps carried out by (*Besnard et al., 2014*). These measurements were made for three experiments on organ initiation with wild-type and two mutation-types (ahp6-1) of Arabidopsis. We pooled the experiments together and combined the two mutation groups. For each plant in each group, we estimated the plastochron by dividing the total observation time of the plant with the total number of organs initiated in it within this time. The distributions of the plastochrons in the two groups (wild type, mutant) are shown in *Appendix 1—figure 3*. The groups differ from each other statistically significantly (in Kruskall-Wallis rank sum test, the null hypothesis "mean ranks of plastochron values in the groups are the same" is rejected with p-value 0.01417; in one-way Welch ANOVA test, the null hypothesis "means of the plastochrons in the groups are the same" is rejected with p-value 0.01055), and the distributions suggest that plastochrons tend to be smaller in mutant plants than in wild plants.

## 4.3 Interpretation of phenotypes with modified meristem size

We revisited the results from (*Landrein et al., 2015*) to check whether the observed correlation between meristem size and perturbation intensity could be quantitatively explained by the stochastic model.

For each genotype, (*Landrein et al., 2015*) assumed that the change of growth conditions induces a change in size of the central zone while the radius of the primordia inhibitory fields is mainly not modified. Therefore, for plants grown in long-day conditions, the size of the meristem decreases, and supposedly induces a decrease of the size of the central zone, thus leading to an increase of $\Gamma$. Here, as inflorescences develop in identical conditions, we assume in addition that the parameters $E^*$ and $\beta$ reflecting the signal sensitivity and acuity of the biological system are not affected by the change in growth conditions. For all the plants, the different observable variables $\alpha$, $T$, $\pi_2$ and $\pi_3$ were measured (*Landrein et al., 2015*). From these data, we can estimate for each genotype and growth condition the corresponding values of the control parameters $(\Gamma_D, \Gamma_P)$ as described in the previous section. Let us denote $G$ a particular genotype, we can derive from the model that the ratio of central zone size due to a change in growth condition is given by:

$$\frac{R(G_{SL})}{R(G_{LL})} = \left(\frac{\Gamma_P(G_{LL})}{\Gamma_P(G_{SL})}\right)^{\frac{1}{7}}\left(\frac{\Gamma_D(G_{LL})}{\Gamma_D(G_{SL})}\right)^{\frac{6}{7}} \quad (13)$$

If it is assumed that this change in central zone sizes is induced by the change in meristem size due to different growth conditions, this ratio is also the ratio of meristem sizes. From this relationship, it is possible to predict whether an increase or a decrease in the meristem size can be expected based on the measured values of the other observed variables. For the tested plants, the model predicted in all the cases a decrease of the meristem size with an error of respectively 4%, 5% and 1% for Col0, WS4 and clasp-1.

# 5 Assessment of pattern identification accuracy

As mentioned above our analysis of permutations and the resulting tables relied on our previously published algorithms (*Refahi et al., 2011*). These algorithms are based on the construction of a set of candidate labellings (organized as a tree), within which sequences containing permutations are selected provided they are close enough to the data given as input. Because these algorithms require some parametrization (to measure "closeness to the data" in particular) and have been used extensively to generate the tables above, we performed a robustness analysis.

Namely, we randomly generated sets of sequences of divergence angles involving 2-permutations and 3-permutations with von Mises noise (circular version of the Gaussian distribution) to model uncertainty. We tested the combinatorial models pattern detection precision for different noise amplitude values as well as different permutations frequency, chosen in adequacy with the available experimental data.

Let $X = \{X^1, X^2, ..., X^{60}\}$ be a set of generated sequences of multiples of $\phi = 137.5°$ involving 2- and 3-permutations, where $X^k = (x_1^k, ..., x_\ell^k)$, $\ell = 25$ (a typical length in experimental data sets), and $1 \leq k \leq 60$. We then introduced noise, leading to $S = \{S^1, ..., S^{60}\}$, $S^k = (s_1^k, ..., s_\ell^k)$, such that $s_i^k \sim P(\mu, \kappa)$, where $P(\mu, \kappa)$ indicates the von Mises distribution with mean value $\mu = x_i^k$, and the concentration parameter $\kappa$. Both $\kappa$ and the permutation frequencies were estimated from the measured sequences in (*Refahi et al., 2011*; *Besnard et al., 2014*; *Guédon et al., 2013*; *Landrein et al., 2015*).

We generated four sets of sequences for different values of permutation frequency and $\kappa$ as described above, see *Appendix 1—figure 4* for the histogram of angles. We then used the combinatorial model to analyse the sequences. By careful human inspection, we then counted the correctly labelled noisy angles and divided them by the number of all divergence angles. The resulting values are reported in *Appendix 1–table 3*.

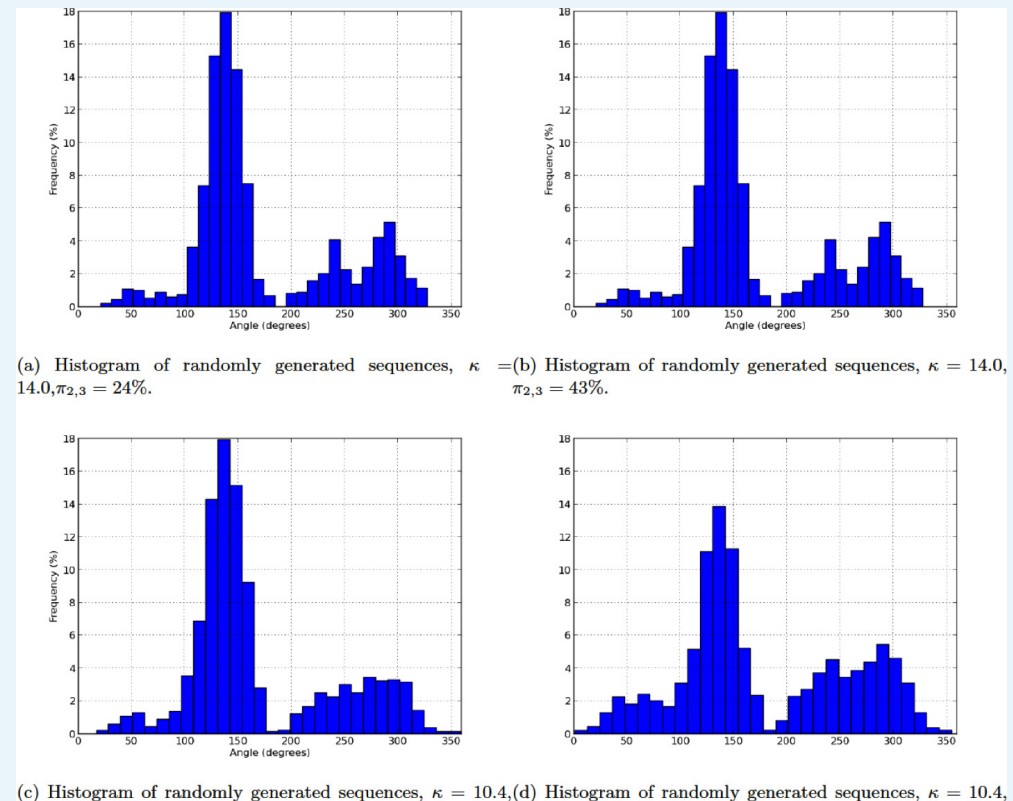

(a) Histogram of randomly generated sequences, $\kappa = 14.0, \pi_{2,3} = 24\%$.

(b) Histogram of randomly generated sequences, $\kappa = 14.0$, $\pi_{2,3} = 43\%$.

(c) Histogram of randomly generated sequences, $\kappa = 10.4$, $\pi_{2,3} = 24\%$.

(d) Histogram of randomly generated sequences, $\kappa = 10.4$, $\pi_{2,3} = 43\%$.

**Appendix 1—figure 4.** Histogram of randomly generated sequences. (**a**) Histogram of randomly generated sequences, $\kappa = 14.0, \pi_{2,3} = 24\%$. (**b**) Histogram of randomly generated sequences, $\kappa = 14.0$, $\pi_{2,3} = 43\%$. (**c**) Histogram of randomly generated sequences, $\kappa = 10.4$, $\pi_{2,3} = 24\%$. (**d**) Histogram of randomly generated sequences, $\kappa = 10.4$, $\pi_{2,3} = 43\%$.

**Appendix 1—table 3.** Combinatorial models permutation detection precision on randomly generated noisy data.

| $\kappa \setminus \pi_{2,3}$ | 24% | 43% |
| --- | --- | --- |
| 14 | 100% | 98.5% |
| 10.4 | 99% | 98% |

# 6 Assessing the model robustness by altering implementation choices

Even though the main ideas of our model are unambiguous, a number of parameters had to be specified for implementation purposes. This raises the question of whether some of our conclusions could be dependent upon these implementation choices. To ensure that this is not the case, we considered a number of alternatives and compared simulation results with our "main choices". The results are reported below and overall they confirm the robustness of our results, as they are not significantly changed by alternative implementation choices.

## 6.1 Generating spiral and whorl patterns from random initial conditions

To test the ability of the model to initiate spiral or whorl patterns *de novo* from unpatterned meristems, we made simulations starting from a random inhibition field $E_k$ at each location $k$ of the central zone periphery, $k$ ranging over the discretized angular positions. $E_k$ is constructed as follows: $E_k = exp(15\sigma_k + 10)$, where $\sigma_k$ is drawn from the uniform distribution on the interval $[0, 1)$ and the overall distribution $(\sigma_k)_k$ is smoothed by applying a Gaussian filter. The value of the inhibitory field in each cell is then updated at each time step of the simulation until the first primordium initiation, by adding a random factor drawn from a uniform distribution on the interval $[-0.5, +0.5]$. For all the simulations, $\beta$ was fixed to $11.0$ and $E^*$ to $1.0$.

We observed that the model is able to generate phyllotaxis patterns from random initial conditions (see movies and their legends in the main text). Depending on the model control parameter $\Gamma_D$, the system was able to generate either spiral of whorl phyllotaxis. In the case of whorl patterns, the system was able to initiate the expected whorl motifs for 2 and 4 organs without any transition phase. This is due to the symmetry of the system combined with its geometry. Starting from a meristem with no preexisting organ, assume that the level of inhibition is low so that a random noise can potentially augment substantially the probability of an organ initiation at any position around the periphery. Soon, a first organ is initiated at a random location on the periphery. This breaks the circular symmetry of the system and favours slightly the site at 180 degrees. As the inhibition level is low, a second primordium is initiated immediately at this new position with a plastochron close to 0. This in turn creates a new inhibitory component to the global inhibitory field. Note that the symmetry of the system has been partially recovered as the second primordium is at the same distance of the center as the first. Therefore, if the range of the inhibition emitted by each primordium is large enough, it prevents for a while the initiation of new primordia. Atfer some growth, the two initial primordia get away from the meristem center. The inhibition decreases at the central zone periphery and after some time, new initiations will occur almost synchronously at the two possible locations perpendicular to the first organ pair direction where the inhibition field is minimal. This initiates a 2-whorled phyllotaxis.

Along similar lines, a 4-whorled pattern can be initiated if, starting from the two initial organs as above, the inhibition does not cover the entire peripheral circle. Then another pair of two primordia can immediately be generated, with plastochrons close to zero, in the direction perpendicular to the first pair and on opposite sides of the central zone. The inhibition produced by the 4 organs covers the entire meristem and another 4 organs will be generated only after some time, when the 4 initial ones have drifted away, sufficiently far from the center. In this case the phyllotaxis is 4-whorled.

## 6.2 Changing spatial discretization

We considered different spatial discretizations. These can be described in terms of the angle at the central zone boundary which corresponds to possible positions for organ initiations. The default discretization, used in the main document, uses a 1-degree resolution. The 2-degree and 5-degree discretizations generate roughly the same permutation rates, whereas the 10-degree discretization leads to an increase of the permutation rates, see *Appendix 1– tables 4–7*.

**Appendix 1–table 4.** 1 degree (in the paper), permutation rates and average divergence angle.

| $\Gamma\|E^* \setminus \beta$ | 10 | 12 |
|---|---|---|
| 0.65, 1 | 30.0 (22.3, 7.7), 136 | 18.4(15.7, 2.8), 136.8 |
| 0.75, 1 | 30.7(21.2, 9.5), 136.2 | 15.5 (11.5, 0.9), 137.3 |
| 0.85, 1 | 17.6(13.5, 4.1), 138.5 | 15.5(11.9, 3.5), 137.3 |
| 0.95, 1 | 11.7(9.5, 1.8), 139.9 | 4.8(4.4, 0.4)140.1 |

**Appendix 1–table 5.** 2 degrees, permutation rates and average divergence angle.

| $\Gamma|E^* \setminus \beta$ | 10 | 12 |
|---|---|---|
| 0.65 , 1 | 31.8(24.3,7.4), 136.4 | 19.7(18.2,1.5), 136.6 |
| 0.75, 1 | 27.1(16.9,10.2), 136 | 11.8(10.2,1.6), 136.5 |
| 0.85, 1 | 19.8(14.7,5.2), 137.3 | 11.0(9.9,1.1), 137.7 |
| 0.95, 1 | 13.6(12.6,1.0), 138.7 | 5.3(5.0,0.3), 139.5 |

**Appendix 1–table 6.** 5 degrees, permutation rates and average divergence angle.

| $\Gamma|E^* \setminus \beta$ | 10 | 12 |
|---|---|---|
| 0.65, 1 | 32.9(25.1,7.8), 136.1 | 21.3(17.5,3.8), 136.1 |
| 0.75, 1 | 35.2(23.3,11.9), 135.7 | 18.1(16.0,2.1), 136.3 |
| 0.85, 1 | 16.6(12.9,3.7), 137.9 | 8.9(7.5,1.4), 137.9 |
| 0.95, 1 | 10.0(8.2,1.9), 139.4 | 4.3(4.3,0.0), 139.9 |

**Appendix 1–table 7.** 10 degrees, permutation rates and average divergence angle.

| $\Gamma|E^* \setminus \beta$ | 10 | 12 |
|---|---|---|
| 0.65, 1 | 36.7(24.7,12.0), 136.5 | 29.7(22.0,7.6), 136.2 |
| 0.75, 1 | 29.0(22.4,6.6), 136.9 | 25.3(20.8,4.5), 136.7 |
| 0.85, 1 | 24.2(17.4,6.8), 135.9 | 21.0(17.5,3.5), 134.8 |
| 0.95, 1 | 13.2(13.2,0.0), 139.3 | 9.4(8.8,0.6), 139.5 |

## 6.3 The effect of the inhibition function on the results and control parameters

We changed the inhibition function to the two following function, used e.g. in (**Douady and Couder, 1996b**)

$$E^{(q)}(x) = \frac{-1 + (\tanh s \frac{d(x,q)}{r_0})^{-1}}{-1 + (\tanh s)^{-1}},$$

while we repeat here our default choice (also considered in [**Douady and Couder, 1996b**]):

$$E^{(q)}(x) = \left(\frac{r_0}{d(q,x)}\right)^s,$$

where $\tanh$ stands for hyperbolic tangent, and for both functions $s$ denotes the inhibition stiffness. Note that though qualitatively controling the stiffness, the same quantitative value of $s$ is expected to give different results for both functions above. In the following **Appendix 1–table 8** we present permutation rates for the $\tanh$ based expression, with stiffness $s = 1$.

**Appendix 1–table 8.** Permutation rates for the $\tanh$ based inhibition rate, with $s = 1$; permutation rates and average divergence angle.

| $\Gamma|E^* \setminus \beta$ | 7 | 8 | 10 | 12 |
|---|---|---|---|---|
| 0.45, 1 | | | 35.9(25.9,10.0), 138.5 | 25.1(20.9,4.2), 138.3 |
| 0.45, 1.2 | 45.9(25.6,20.3), 138.8 | 42.4(30.1,12.3), 138.5 | 26.8(21.1,5.7), 138.1 | 17.7(16.0,1.6), 137.9 |
| 0.55, 1 | 44.3(27.4,16.8), 138.0 | 36.1(26.5,9.6), 138.2 | 23.3(20.1,3.1), 137.5 | 15.9(14.5,1.3), 137.4 |
| 0.55, 1.2 | 36.7(25.8,10.9), 138.0 | 28.0(20.4,7.6), 137.9 | 17.1(15.5,1.6), 137.7 | 13.9(13.9,0.0), 137.5 |

*Appendix 1–table 8 continued on next page*

*Appendix 1–table 8 continued*

| $\Gamma|E^* \setminus \beta$ | 7 | 8 | 10 | 12 |
|---|---|---|---|---|
| 0.65, 1 | 39.0(26.3,12.8), 135.7 | 30.4(23.7,6.7), 136.3 | 15.3(14.1,1.1), 136.3 | 10.3(9.7,0.6), 136.8 |
| 0.65, 1.2 | 27.7(22.2,5.6), 136.4 | 19.9(17.1,2.8), 136.7 | 10.6(10.2,0.4), 136.7 | 4.3(4.3,0.0), 136.7 |
| 0.75, 1 | 40.2(23.0,17.2), 135.2 | 32.2(20.9,11.3), 135.1 | 17.6(14.4,3.2), 135.8 | 10.9(10.3,0.5), 136.3 |
| 0.75, 1.2 | 31.1(22.6,8.5), 135.0 | 21.3(18.1,3.2), 136.0 | 11.1(10.0,1.1), 136.5 | 6.7(6.2,0.5), 137.1 |
| 0.85, 1 | 27.4(17.5,9.9), 137.6 | 19.2(14.4,4.8), 137.2 | 5.9(5.9,0.0), 138.0 | 4.4(4.2,0.2), 138.1 |
| 0.85, 1.2 | 21.5(15.8,5.8), 136.4 | 14.4(13.0,1.4), 136.9 | 5.6(5.6,0.0), 137.6 | 3.1(3.0,0.1), 138.0 |
| 0.95, 1 | 21.0(16.2,4.9), 138.9 | 14.8(10.7,4.0), 138.6 | 3.7(3.7,0.0), 139.6 | |
| 0.95, 1.2 | 15.4(12.4,3.1), 138.4 | 8.5(7.3,1.2), 138.9 | 2.9(2.9,0.0), 139.4 | |
| 1.05, 1 | 22.4(16.8,5.6), 139.6 | 13.1(11.0,2.1), 139.2 | 2.4(2.4,0.0), 139.6 | |
| 1.05, 1.2 | 10.2(9.3,0.9), 139.5 | 5.8(5.7,0.1), 139.7 | 2.1(2.1,0.0), 139.9 | |

The control parameters $\Gamma_P$ and $\Gamma_D$ turn out to be the control parameters of the system using this new inhibition function, as can be seen in *Appendix 1—figure 5*.

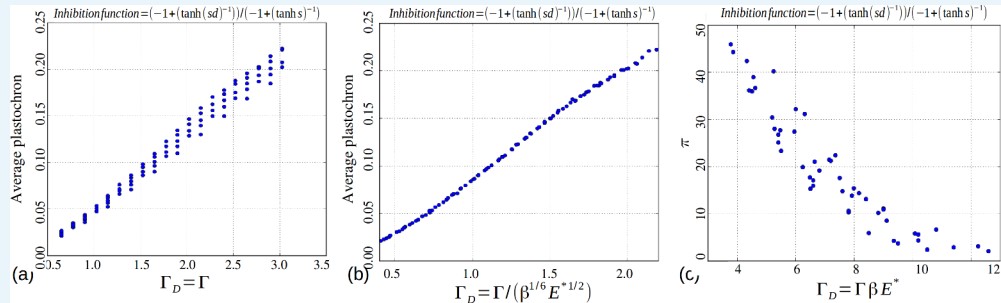

**Appendix 1—figure 5.** Role of the control parameters for the $\tanh$ based inhibition function. (**a–b**) Average plastochron ratio as a function of $\Gamma$ and $\Gamma_D = \frac{\Gamma}{\beta^{1/6}E^{*1/2}}$, respectively. (**c**) Number of permutations $\pi$ as a function of $\Gamma_P$.

When using our default (power law) inhibition function, we have fixed $s = 3$ in the main paper. We found the $\Gamma_D = \frac{\Gamma}{\beta^{1/6}E^{*1/2}}$ as the control parameter of plastochrons.

As shown in *Appendix 1—figure 6*, when using a ower law the parameter $\Gamma_D$ can in fact be related more explicitly to the stiffness $s$, by using

$$\Gamma_D = \frac{\Gamma^{s/3}}{\beta^{1/6}E^{*1/2}}$$

instead of the previous expression.

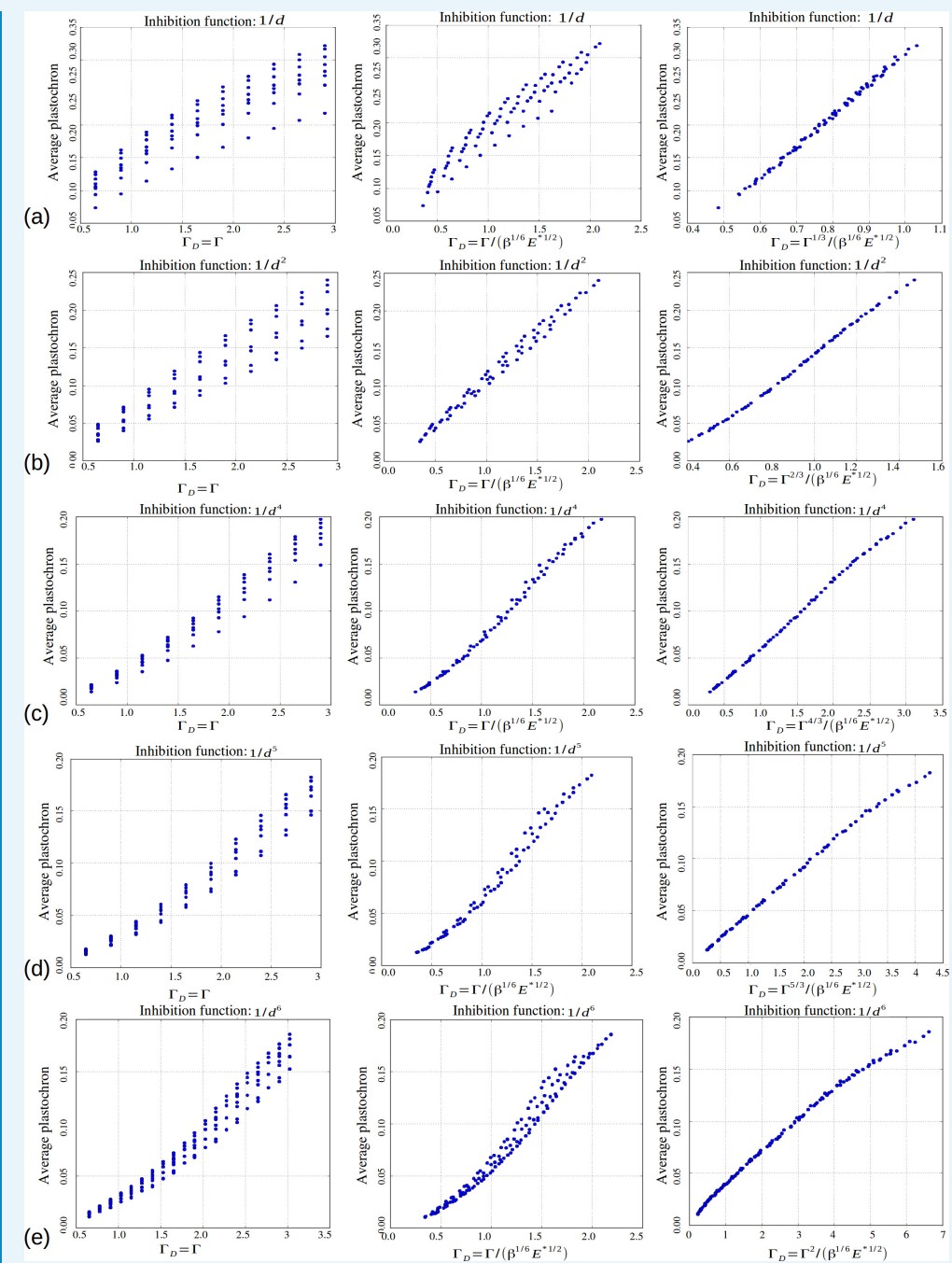

**Appendix 1—figure 6.** Role of the control parameters for the power law inhibition function, with rows (**a–e**) corresponding respectively to a steepness $s = 1, 2, 4, 5, 6$. First two columns: average plastochron ratio as a function of $\Gamma$ and $\Gamma_D = \frac{\Gamma}{\beta^{1/6}E^{*1/2}}$, respectively. Third column: average plastochron ratio as a function of $\Gamma_D = \frac{\Gamma^{s/3}}{\beta^{1/6}E^{*1/2}}$.

## 6.4 Changing the growth rate

Because of growth, the primordia drift away radially from the central zone at a velocity proportional to their distance from the center, i.e. $\frac{dr(t)}{dt} = \gamma r(t)$. In the simulations we have used $\gamma = 5$, we tested $\gamma = 3$, see the results in **Appendix 1–table 9**, which do not depart significantly from our main table, see e.g. **Appendix 1–table 4** for comparison.

**Appendix 1–table 9.** Growth rate changed, $\gamma = 3$.

| $\Gamma|E^* \setminus \beta$ | 10 | 12 |
|---|---|---|
| 0.65, 1 | 29.0(21.0,8.0), 136.2 | 17.7(16.2,1.5), 136.6 |
| 0.75, 1 | 27.4(19.8,7.6), 135.4 | 15.8(14.6,1.2), 136.2 |
| 0.85, 1 | 14.7(11.3,3.4), 137.7 | 10.5(9.8,0.6), 138.0 |
| 0.95, 1 | 8.0(6.9,1.1), 139.3 | 4.3(4.1,0.2), 139.3 |

## 6.5 Adding noise to the primordia initiation site on the periphery of central zone

To see the effect of the noise on the sampling of the peripheral zone, we added a random noise in $\{-2, -1, 0, 1, 2\}$ degrees to the initiation site of the primordium. The permutation rates are most of the time slightly increased.

**Appendix 1–table 10.** Noise added to the peripheral zone.

| $\Gamma|E^* \setminus \beta$ | 10 | 12 |
|---|---|---|
| 0.75, 1 | 29.2(19.5,9.7), 135.9 | 16.8(14.2,2.6), 136.7 |
| 0.75, 1.2 | 17.2(14.8,2.3), 136.4 | 14.6(12.5,2.1), 136.1 |
| 0.85, 1 | 23.6(16.0,7.7), 136.6 | 12.3(11.2,1.1), 137.7 |
| 0.85, 1.2 | 16.5(14.2,2.2), 135.7 | 9.2(8.3,0.9), 137.2 |
| 0.95, 1 | 12.5(11.4,1.1), 139.4 | 6.9(6.7,0.2), 139.4 |
| 0.95, 1.2 | 8.6(7.5,1.1), 138.9 | 2.4(2.4,0.0), 138.6 |

## 6.6 Introducing new primordia at arbitrary positions

As a control to the experiment reported in *Figure 2G*, we ran simulations where a new primordium is artificially introduced in an otherwise stable spiral pattern. While *Figure 2G* only considers the case where this "forcing" takes place at the angle corresponding to the second local minimum, we report here the system's response in cases of arbitrary angular positions.

These types of perturbations amount to exploring random initial conditions. In the bifurcation diagram of the "standard model" (cf. *Appendix 1—figure 2*), different initial conditions can lead to different phyllotactic patterns (the branches of the bifurcation diagram); each pattern is reached by a specific set of initial conditions called its *basin of attraction*. If the artificially created "initial condition" lies outside the basin of attraction of the current pattern, we expect to observe a change of phyllotactic pattern in the subsequent simulation. This intuition is confirmed by simulations: see a broad classification of the possible effects of different typical "new initial conditions" in *Appendix 1—figure 7*.

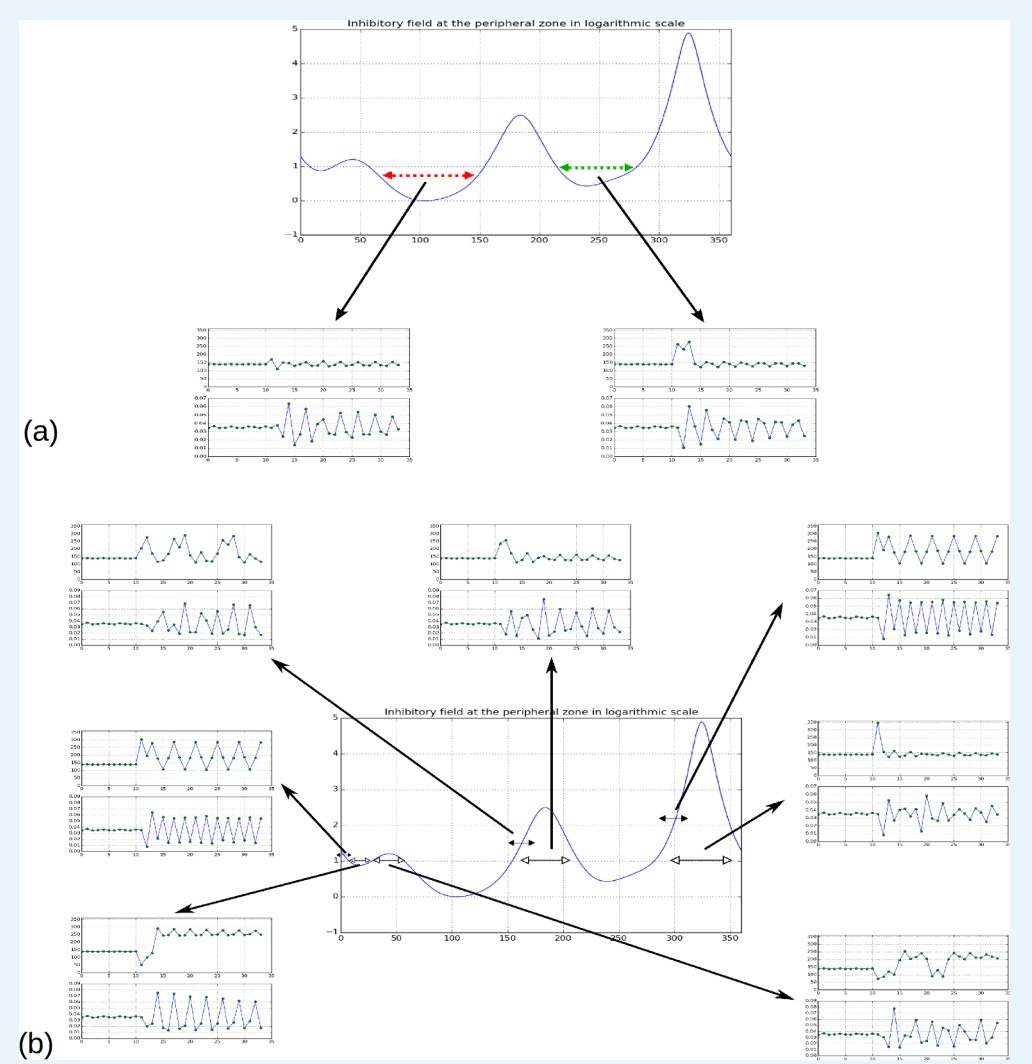

**Appendix 1—figure 7.** Effect of forcing initiation in different regions of the inhibitory field profile. (**a**) As reported in **Figure 2H**, forcing a primordium near the first or second local minima does not affect the phyllotactic mode permanently, but simply forces a 2-permutation in the case of the second minimum. (**b**) When primordia are introduced near the highest or second highest local maxima, the system returns to its previous pattern. Near the third maximum or the third minimum, the system changes pattern, in the shown simulation it converges to a spiral oriented in the opposite direction to the previous spiral (i.e. $\varphi = -137.5$ instead of $+137.5$); a 2-permutation can be seen in the new spiral for the third maximum. New primordia introduced far from any maximum or minimum (black headed arrows) tend to disrupt the phyllotactic mode more significantly, converging to whorled patterns (primordia forced near 0 or near 290), or patterns involving high numbers of successive permutations (primordium forced near 150).

## Appendix 2

### Additional videos and initial conditions for all videos

For all videos, unless otherwise stated, we used either of the following initial conditions:

Random: We assume that no organ preexists at the meristem surface and that the level of inhibition randomly fluctuates in the peripheral cells. We model this situation by defining a random inhibition field $E=(E_k)_k$ for $k$ ranging over the discretized angular positions as follows: $E_k=\exp(15\sigma_k+10)$, where $\sigma_k$ is drawn from the uniform distribution on the interval [0, 1) and the overall distribution $(\sigma_k)_k$ is smoothed by applying a Gaussian filter. The value of the inhibitory field in each cell is then updated at each time step of the simulation until the first primordium initiation, by adding a random factor drawn from a uniform distribution on the interval [-0.5, +0.5].

Empty: We assume that no organ preexists at the meristem surface and that the level of inhibition is zero in every cell at the beginning of the simulation (i.e. $E_k=0$ for all $k$ at $t=0$). The first primordium is generated at a random location on the peripheral zone.

For all the simulations, $\beta$ was fixed to 11.0 and $E^*$ to 1.0.

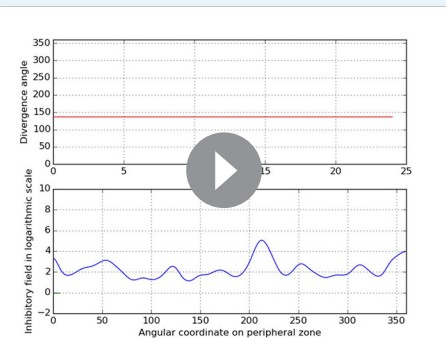

**Appendix 2—video 1.** Dynamics of the divergence angles as well as the inhibitory field at the peripheral zone. The stochastic model generates a Fibonacci spiral including two 2-permutations from a random initial inhibitory field. For more precise information about the initial condition see Supplementary information.

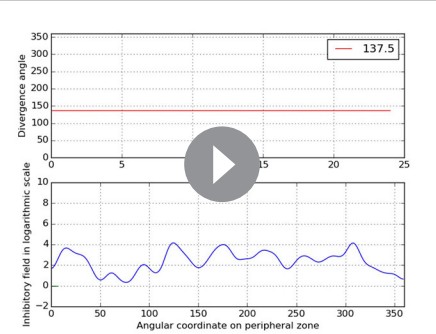

**Appendix 2—video 2.** Dynamics of the divergence angles as well as the inhibitory field at the peripheral zone. The stochastic model generates a Fibonacci spiral from a random initial inhibitory field. It takes some time to converge to 137 degrees. For more precise information about the initial condition see Supplementary information.

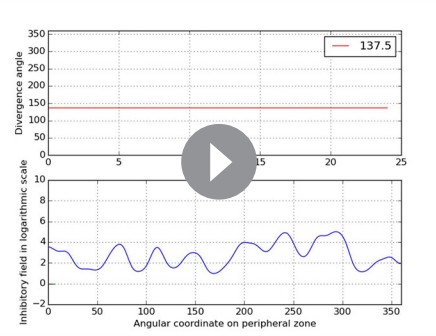

**Appendix 2—video 3.** Dynamics of the divergence angles as well as the inhibitory field at the peripheral zone. The stochastic model generates a Fibonacci spiral pattern including a 2-permutation pattern at the beginning before converging to 137 degrees. For more precise information about the initial condition see Supplementary information.

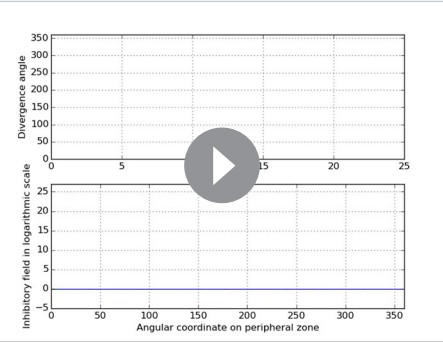

**Appendix 2—video 4.** Dynamics of the divergence angles as well as the inhibitory field at the peripheral zone.  The stochastic model generates a 2-whorled pattern from no pre-existing organs. For more precise information about the initial condition see Supplementary information.

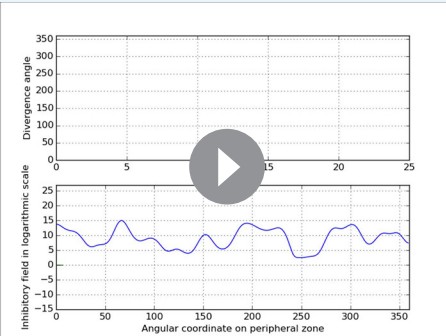

**Appendix 2—video 5.** Dynamics of the divergence angles as well as the inhibitory field at the peripheral zone. The stochastic model generates a 2-whorled pattern from a random initial inhibitory field. For more precise information about the initial condition see Supplementary information.

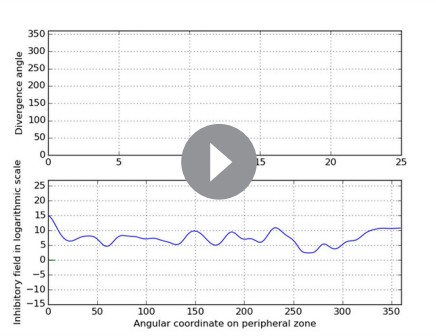

**Appendix 2—video 6.** Dynamics of the divergence angles as well as the inhibitory field at the peripheral zone. The stochastic model generates a 2-whorled pattern from a random initial inhibitory field (second run). For more precise information about the initial condition see Supplementary information.

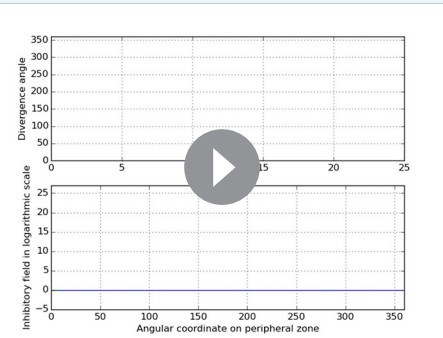

**Appendix 2—video 7.** Dynamics of the divergence angles as well as the inhibitory field at the peripheral zone. The stochastic model generates a 2-whorled pattern from no pre-existing organs (second run). For more precise information about the initial condition see Supplementary information.

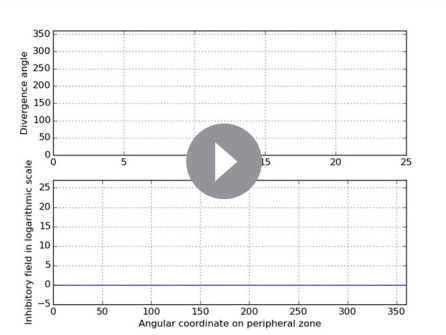

**Appendix 2—video 8.** Dynamics of the divergence angles as well as the inhibitory field at the peripheral zone. The stochastic model generates a 4-whorled pattern from no pre-existing organs. For more precise information about the initial condition see Supplementary information.

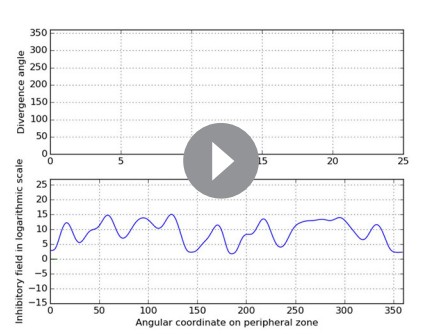

**Appendix 2—video 9.** Dynamics of the divergence angles as well as the inhibitory field at the peripheral zone. The stochastic model generates a 4-whorled pattern from a random initial inhibitory field. For more precise information about the initial condition see Supplementary information.

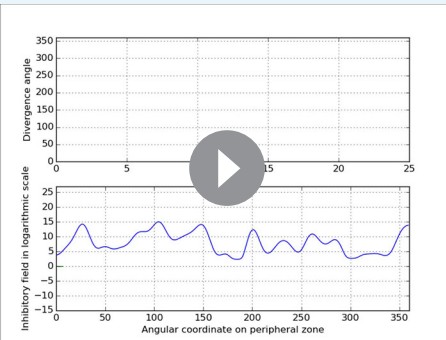

**Appendix 2—video 10.** Dynamics of the divergence angles as well as the inhibitory field at the peripheral zone. The stochastic model generates a 4-whorled pattern from a random initial inhibitory field (second run). For more precise information about the initial condition see Supplementary information.

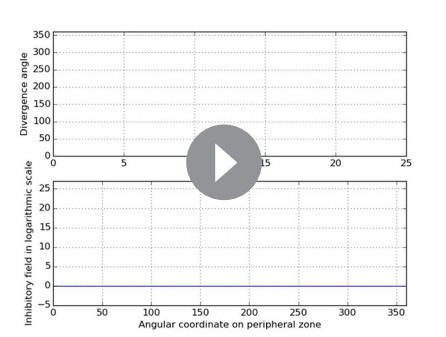

**Appendix 2—video 11.** Dynamics of the divergence angles as well as the inhibitory field at the peripheral zone. The stochastic model generates a 4-whorled pattern from no pre-existing organs (second run). For more precise information about the initial condition see Supplementary information.

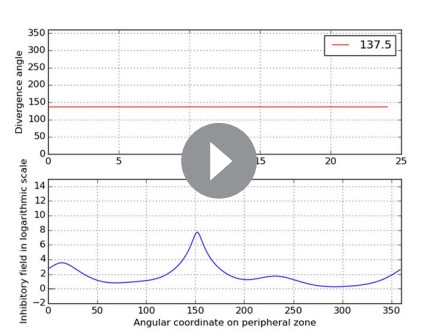

**Appendix 2—video 12.** Dynamics of the divergence angles as well as the inhibitory field at the peripheral zone showing multiple initiation sites. The SMPmicro variant was used instead of SMPmacro-max used in the previous simulations. With SMPmicro, a primodium may be initiated at any cell independently of the others. This makes it possible to two primordia initiation in the same valley. A single initiation site is marked in green and multiple simultaneous initiation sites are marked in red. An initial spiral pattern of 150 preexisting primordia has been used as initial condition.

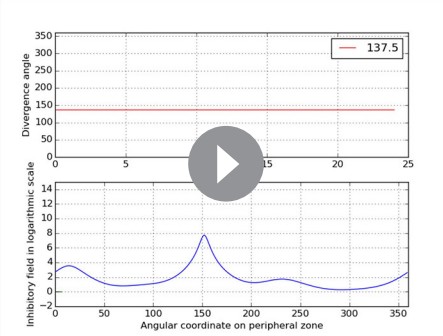

**Appendix 2—video 13.** Same video as *Appendix 2—video 12* with lower fps.

