## [Decision Letter]

Thank you for submitting your work entitled "A Stochastic Multicellular Model Identifies Disorders as Biological Watermarks in Self-organized Patterns" for consideration by *eLife*. Your article has been reviewed by three peer reviewers, and the evaluation has been overseen by a Reviewing Editor and Naama Barkai as the Senior Editor.

The reviewers have discussed the reviews with one another and the Reviewing Editor has drafted this decision to help you prepare a revised submission. All reviewers and the editors agree that your work is of potential interest, however they also identified a number of shortcomings that we would like to ask you to address. Please pay attention to the following essential revisions. The reviewers and editors agree that without them, and depending on their outcome, the paper could not be published. They are the following:

1) To convince your audience that your model is capable of the self-organization shown experimentally, you would have to demonstrate that it is able to create a whorled pattern from a random starting point. This is important because a random starting point may change the way the system responds to noise.

2) To demonstrate robustness of the model and its in vivo credibility, we believe it is important that you vary some of the assumptions and parameters in the model, such as the spatial and temporal discretization, the choice of inhibition function (which would vary the nature of the inhibition), and the presence/absence of a secondary mechanism for plastochron control.

Moreover, ideally, you could in addition demonstrate that you get similar results analyzing mistakes in an unrelated system. This would considerably strengthen the credibility of your approach.

Find more details in the reviews pasted below, and please address to your best capacity the other points raised in there.

*Reviewer #1:*

The authors propose that we can learn something about the mechanism that creates a pattern, not just by looking at the pattern itself, but also by looking carefully at the mistakes it makes. Motivated by recent work, the authors use phyllotactic patterns in order to explore this idea. I find the idea interesting. They suggest that the analysis of the mistakes can help us to understand the mechanism, and they give us an example, the *ahp6* mutant, that their analysis shows is more likely the result of a reduction in size of the inhibitory field about primordia, rather than a change in meristem size. This would allow us to distinguish between these two choices. This is not really proof that this is the case, but rather a proof of principle that the idea has discriminating power. In this case there are other possibilities that would need to be excluded, such as increased/decreased noise in location selection due to changes in cell size, or any other input that could affect noise in the system.

I would have like to have seen more exploration of the effect of other model formulations of the model on the results they observe. For example, does the rate of permutations depend on the choice of the inhibition function? What about the spatial and temporal discretization? The growth rate? What does the shape of the curve look like in response to noise in the sampling of the peripheral circle where primordia are formed?

Specific comments:

General: I am not really fond of the comparison to watermarks, I find it a bit misleading as the perturbations are random after all.

Introduction: The presentation in the Introduction suggests that Douady and Couder pioneered the inhibition field model of phyllotaxis. Although they contributed considerably to the field, the authors should perhaps mention at least a few of the others that came before, such as Thornley (1975), Mitchison (1977), Veen and Lindenmayer (1977), Young (1978), Schwabe and Clewer (1984), Chapman and Perry (1987), etc.

Results: The authors write that it has been shown that the central zone is high in auxin. Based on the data, I would say it has been "suggested" or "proposed". Data to support this idea is only from Arabidopsis inflorescense meristem, and from the DII reporter without the recent control construct added by Weijers. The data looks completely different in tomato, where DR5 works just fine in the central zone.

Results: The authors report dislocations in other species, and then immediately assume that it is due to variation in the plastochron. One other possibility could be variations in inhibition threshold. In most inhibition models of phyllotaxis these two are tied together, but that is not necessarily the case in planta. In fact, does not their previous work (Besnard et al. 2014) claim that the timing (i.e. plastochron) is controlled somewhat independently? To report that it is a variation in plastochron, analysis of the timing of primordium initiation at the meristem is required.

Also it is not clear to me from the side views in Figure 2 if these dislocations are just random positioning, or are permutations as stated in the figure caption and text. You would need an image or diagram from the top, or the angles plotted on a graph in order to see if this is the case.

Results: The authors state that the inhibition profile can be seen as an energy. In what sense is it an energy? I think it would be less confusing for the reader, and more accurate to refer to it as the inhibition profile.

Property 2 is reported in the caption for Figure 5 in Smith et al. (2006, CJB), you should probably reference that work in this context.

In a simulation experiment, they perturb the model by initiating primordia at a local inhibition minimum that is not the global minimum and observe recovery. They need the control experiment here. What happens when they initiate a primordia at a random location? Does not the pattern also recover (as it does in ablation experiments)? How does this compare?

Also the authors argue that noise in divergence angle order does not propagate far, but that it has a long effect on the timing, but they don't really give any proof this. We would need to see the time, divergence angle pairs produced by the model, and the model would have to have a similar discretization in time as in space (in most simulation models space it much more coarsely discretized than time). Finally, I might be convinced if they could show evidence of this in planta. It seems to me that after ablation experiments the timing recovers just as fast as the divergence angle.

Stochastic model section: Again they are assuming there is no separate control of the plastochron. I wonder if their stochastic model leads to a different outcome than just adding noise to the system? An example would be the Smith et al. (2006, CJB) model, where noise is added to the peripheral circle sampling point positions, which would also have the effect of adding noise to the inhibition threshold. Don't get me wrong, I appreciate a proper stochastic model, but the worth of such models becomes apparent when they demonstrate different behavior than a simple deterministic model + noise. Is this really the case here?

Speaking of the Smith et al. model, another major source of noise in this system is the fact that the resolution of the possible locations for primordia to initiate is very low. Fate change is a cellular phenomena, and there are really not very many cells around the peripheral zone, maybe 30-40 cells. This gives a resolution of around 10 degrees, a considerable source of noise. Given this it would seem that a perturbation of the possible initiation locations would be very relevant.

Materials and methods: The common phyllotaxis angles are irrational, and have the property that new primordia will be placed far from existing ones. It is therefore not surprising that if you include higher permutations, you start to obtain a roughly uniform sampling of the entire meristem. Consideration of 5 permutations gives 6 matching points, 52.5, 105, 137.5, 190, 275, 327.5 so random angles will always be close to one of these locations, which could explain the data in Figure 7 would need to see statistics on this to demonstrate that these angle matches are really closer than random.

*Reviewer #2:*

Phyllotaxis has fascinated philosophers and scientists for centuries being one of few known examples of biological patterns characterized by such regularity, and hundreds of papers were published focused on biological and mathematical interpretations of this unique trait. The submitted manuscript, however, shows that "the other side" of real phyllotactic patterns, namely their perturbations, can also be informative in terms of shoot development regulation. The authors present two models of phyllotactic pattern generation: in the first model the "classical" assumptions often used in phyllotactic pattern models are employed; in the second one stochastic factors are added. Patterns generated with the aid of two models are compared with one another as well as with the empirical data, mainly on Arabidopsis inflorescence phyllotaxis (wild type and mutants). I would like to point several advantages of the presented stochastic model: it accounts for expansion (growth) of the apex surface; primordia simulated in the model are generated by groups of "cells" which all contribute to the pattern formation; the model reproduces real phyllotactic patterns as shown by comparison of pattern parameters, in particular the disorders in the patterns; the biological meaning of the variables used in the model is thoroughly and critically discussed, and those that can have the biological meaning assigned are recognized.

I would like to point to three topics that could be addressed by the authors:

1) In the Introduction and Abstract, the authors state that developmental disorders are generally not considered as informative. I would not point it so strongly, since in fact studies on mutant phenotypes are often studies on disturbed patterns that lead us to conclusions on pattern generation mechanisms. Also some old literature was devoted to teratology, i.e. disturbed development, in order to speculate on developmental mechanisms.

2) From my knowledge on empirical and theoretical papers on phyllotaxis one of the discrepancies between the two is in that in nature, so-called fused or double primordia (or leaves) can be sometimes observed, while they are most often not generated by models. Would it be possible to generate such primordia in the stochastic model (decreasing the epsilon value)? This could provide one more example of pattern perturbation in support of the model.

3) Most reference to empirical data are done with Arabidopsis, obviously a good choice since some mechanisms on the phyllotaxis regulation are recognized based on the mutation. Nevertheless, the relative size of primordia in Arabidopsis is large, and they are not densely packed around its SAM (low numbers of contact parastichies), while in other species primordia are relatively small and densely packed. I understand that the phyllotaxis parameters used for comparison with model generated phyllotaxis, are not available for these species, but these cases could be to larger extent referred to. For example, shoots with relatively small primordia and short plastochrons (conifer twigs, Asteraceae capitulum) would support the model conclusion on relationships between plastochron duration and frequency of permutations (Results); just have a look on the already dead Christmas tree thicker twigs or trunk, and you will most likely find some permutations in support of the model.

*Reviewer #3:*

My ability to fully interpret this paper is very limited since it is mainly mathematical and my background is biology. However, I understand the main message to be that the authors take an inhibitory field model for phyllotaxis and integrate stochasticity into it. The stochastic model recapitulates the normal patterns of organogenesis as well as observed deviations. It also enables the authors to infer information about the sources of noise.

I think that the ability to help pin-point the root cause for why a pattern is disrupted is very useful and hence this paper is helpful to phyllotaxis researchers. However, I still have concern over the paper's general significance.

My other concern is how model-dependent the conclusions are in any case. Their model, based on abstracted inhibitory fields rather than self-organising auxin transport seems to me to lack a fundamental feature of the latter. For instance, auxin application to un-patterned meristems reveals that organ spacing is robust to random initial distributions of applied auxin (Reinhardt et al., 2003). From what I understand of the model in this paper (and I may be wrong) this fundamental property is not a feature. The ability of the system to form patterns dynamically from random starting points is however captured by less abstracted models based on polarity feed-back from auxin concentrations (Jonsson et al., 2006).

If the model used does not have this property I am worried that the conclusions are not relevant to the real plant.

[Editors' note: further revisions were requested prior to acceptance, as described below.]

Thank you for resubmitting your work entitled "A Stochastic Multicellular Model Identifies Biological Watermarks from Disorders in Self-organized Patterns" for further consideration at *eLife*. Your revised article has been favorably evaluated by Naama Barkai (Senior editor), a Reviewing editor, and three reviewers.

The manuscript has been improved but there are some remaining issues that need to be addressed before acceptance, as outlined below:

The reviewers and editors agree that you should change the manuscript title to reflect its more narrow proof of concept nature, for example "[…] from Disorders in the self-organizing pattern of phyllotaxis".

Please fix some of the wording, in particular the use of inhibition vs energy (see comments below), to avoid misunderstandings in the context of the wider literature.

Please address the issue of the dynamic definition of inhibitory fields. This appears to be an important limitation of your current model, and although we do not ask you to resolve it at this point, we would like to ask you to explicitly spell out this limitation If it is not captured in the model. That is, specification (or growth of the organ) and the generation of inhibition could occur simultaneously and dynamically rather than in a consecutive, step-wise fashion as is modelled here. Maybe you could also comment on how would this influence the way the system responds to noise?

Find more details in the reviewer comments below, but you only need to respond to the points raised above.

*Reviewer #1:*

I am mostly satisfied with the changes the authors have made to address the comments from the initial review. It is too bad that they were not able to apply their method to another patterning system, but the authors argue that it would be a complete work in itself, although I am not so sure. I think the application of the method to at least one other system would speak to the potential generality of their ideas, and without it the paper seems in a bit of a niche area (i.e. plant phyllotaxis specific).

*Reviewer #2:*

Judging from the answers to Editor's and Reviewers' comments provided by the authors and, most of all, from the changes introduced in the manuscript, it has been significantly improved and most of the suggested changes have been introduced: more modelling was performed widening the examined cases, and model interpretation has been improved, also the changes have been made in the main text as suggested. Nevertheless, the manuscript in the present form is for me more clear and representing a broader approach to the problem.

*Reviewer #3:*

The authors have now attempted to address my main point which was to test whether the model could create a whorled pattern from a random starting point. While I'm happy the authors have done this simulation I still have a remaining concern on this point. This is that in their description of this process they relate that the first primordium is specified stochastically.

The current models for phyllotaxis based on polarized auxin transport are able to self-organize auxin peaks (and hence primordia) simultaneously. Auxin is dynamically distributed over time to create spacing similar to a Turing mechanism. Again, the spacing process for all auxin peaks occurs spontaneously and dynamically. From the supplied description I am still not convinced this model captures this dynamic. In other words, in current models based on auxin transport (e.g. Johnson et al. (2006), an inhibitory field isn't specified only after a primordium is specified (as seems to be the case here), it is being generated dynamically during the process of specification i.e. auxin build-up. Hence the whole process can adjust to changing conditions as they occur – it is dynamic. In contrast, in the whorl simulations here, it seems an initial position is specified by noise and then others are specified in quick succession at a spacing that allows several to form around the apex in the same plastochron but not really in a spontaneous manner.

I guess it really comes down to the question: Can the authors map their model (with or without the stochasticity) on to the class of models represented by the current polarized transport models? For instance, as done by Newell, Shipman and Sun (2008) Journal Theoretical Biology 251, 421-439.

Overall I do think it is critical to show that the model they have chosen to use captures the same fundamental properties as exhibited by models that are more closely based on current experimental work.

---

## [Author Response]

*1) To convince your audience that your model is capable of the self-organization shown experimentally, you would have to demonstrate that it is able to create a whorled pattern from a random starting point. This is important because a random starting point may change the way the system responds to noise.*

As suggested, we completed our analysis of the self-organization of our model by testing its ability to generate both spiral and whorl patterns from a random starting point, i.e. a naked dome, where cells are interpreting a noisy background signal. We showed that these conditions are sufficient to initiate both types of patterns (spiral and whorls) in a robust manner. Movies showing these simulations are provided as supplementary material (see Appendix, section 7 for details about each movie).

*2) To demonstrate robustness of the model and its in vivo credibility, we believe it is important that you vary some of the assumptions and parameters in the model, such as the spatial and temporal discretization, the choice of inhibition function (which would vary the nature of the inhibition), and the presence/absence of a secondary mechanism for plastochron control.*

We are most grateful for this comment, as it allowed us to determine a more robust formula for the control parameter Γ*_D_* that includes our previous expression of Γ*_D_* as a particular case. It turns out indeed that this parameter is also dependent on the stiffness of the inhibition function.

More specifically, we have considered variations of the discretization and inhibition function as detailed below and in our specific answers to reviewers and in all cases our main conclusions remained unchanged (except for the precise form of Γ*_D_*). These new results are reported in detail in the appendix, section 6, and correspond to several additional in silico experiments:

Spatial discretization: We made simulations with different angular discretizations (1°, 2°,5°, 10°; 1° being the value initially used in the original manuscript) and show that these changes affect mostly the precision of the divergence angle and only marginally the levels of permutations in the simulated sequences, as seen in our new Appendix section 6.2.Inhibition function: We tested other types of inhibition function, either by varying thestiffness of the original function (values *s* = 1,2,3,5,6 have been tested) or by changing the form of the inhibition function itself (see Appendix section 6.3). This analysis revealed that the change in the inhibitory function per se did not modify the previous results (nature of the control parameters, permutation frequencies). However, the changes in stiffness were shown to modify the dynamics of the system, similarly to the previously defined model parameters (Γ, E*, *β*). This led us to modify slightly the definitions of the control parameters so that they account for the effect of *s* on the system. Equations in the text were changed accordingly. Importantly, this analysis led us to a more general formulation of the control parameters, including now stiffness.

Concerning temporal discretization, contrary to classical deterministic models, there is no discretization of time in our stochastic model. This is an advantage as parameters related to time discretization have been removed. The model simulates continuous time by drawing initiation events from continuous non-homogeneous Poisson process (see algorithm in Appendix section 3.2).

Concerning plastochron control, it is not necessary to integrate any secondary mechanism in the model. We did previously show that cytokinin-based secondary fields (encoded by the distribution of the AHP6 protein: Besnard et al. Nature 2014) regulate specifically the plastochron. However, reinterpretation of our previous data in the light of our new model indicates that such a field can be viewed as a component of the inhibitory fields whose effect is equivalent to an increase in Γ. Our model has the native capacity to integrate different components in the inhibition function. This aspect was discussed in the Discussion section of our original submission (see section of Developmental disorders as biological watermarks).

*Moreover, ideally, you could in addition demonstrate that you get similar results analyzing mistakes in an unrelated system. This would considerably strengthen the credibility of your approach.*

We agree that, ideally, similar results on other independent systems would be remarkable. We think indeed that our work lays the ground for such investigations in developmental biology. Our model is about patterning systems with lateral inhibition. We have developed it on a 2D circular system. This could similarly be developed on other developmental systems based on lateral inhibitions and that can be modelled in 2D (e.g. mouse hairs, trichomes) or 1-D (somites, anabaenae). However, such a work necessitates extensive quantitative data and analysis of the underlying structure of the data. To our knowledge, no other systems have been analysed at the moment in a way that permits such an exploration. In addition, providing a similar demonstration in any other system would be a complete work in itself (that necessitates modelling a new developmental system) and collect completely new datasets from various phenotypes and we believe this is out of the scope of the present work.

*Reviewer #1:*

*I would have like to have seen more exploration of the effect of other model formulations of the model on the results they observe. For example, does the rate of permutations depend on the choice of the inhibition function? What about the spatial and temporal discretization? The growth rate? What does the shape of the curve look like in response to noise in the sampling of the peripheral circle where primordia are formed?*

As pointed out by the reviewer, our study aims at showing the plausibility of a stochastic perception of the inhibition signal to explain observed phyllotaxis perturbation in wild-type and mutant plants. What we show here is that a noise in the perception of the signal satisfactorily explains all the data that have been recently observed about perturbations in phyllotaxis systems. However, it is true that other sources of noise could be imagined. To address this concern, we investigated various possibilities as suggested by the reviewer. We have addressed some of these in the section on essential revision and recall below our conclusions.

The choice of inhibition function: We tested other types of inhibition function, either byvarying the stiffness of the original function (values *s* = 1,2,3,5,6 have been tested, see Appendix section 6.3) or by changing the inhibition function itself. The main outcome of these new simulations is that our main results remain essentially unchanged, provided the control parameter is updated to include the stiffness parameter. This updated control parameter confirms our qualitative claims and introduce a new level of refinement, for which we are very grateful to Reviewer #1. Equations in the text were changed accordingly to include the stiffness parameter.The spatial discretization of the peripheral zone: We made simulations with different angular discretizations (1°, 2°, 5°, 10°; 1° being the value initially used in the original manuscript) and show that these changes affect the precision of the divergence angle rather than the levels of permutations in the simulated sequences, as seen in our new Appendix section 6.2.The temporal discretization: contrary to classical deterministic models, there is nodiscretization of time in our stochastic model. This is an advantage as parameters related to time discretization have been removed. The model simulates continuous time by drawing initiation events from continuous non-homogeneous Poisson process (see algorithm in Appendix section 3.2).Growth rate: Changing the growth range does not change divergence angle, and other curves, but accelerates or decelerates plastochrons (temporal scaling). This has a low effect on the frequency of permutations (see Appendix section 6.4), suggesting that, as expected, the growth rate is not determinant for the spatio-temporal patterning of the system.

Finally, as reported in Appendix section 6.6, we have tested simulations where a random *noise* is introduced in the sampling of the peripheral zone. More precisely, the precise initiation position was randomly shifted by a few positions compared to the exact position induced by the inhibition field. Perhaps not surprisingly, the results in Appendix 6.6 show that this leads to a moderate increase in the number of permutations.

We have updated the Results and Discussion to highlight these new results. We believe that this strengthens significantly our conclusions.

*Specific comments:*

*General: I am not really fond of the comparison to watermarks, I find it a bit misleading as the perturbations are random after all.*

We thank the reviewer for this remark. It is true that the perturbations are random but, they are not the watermark per se. Our previous formulation was indeed ambiguous. In our analogy, the biological watermark (the hidden information) is made of the state variables Γ, E*, β. What we say is that by looking in the actual phyllotactic signal, that contains noise, it is possible to compute a perturbation intensity, i.e. this is part of the algorithm to extract the watermark. This, together with the estimation of the other observable variables, plastochron and divergence angle, provides information on the actual values of the hidden state variables, and strongly restricts their range of values. This information is our biological watermark.

To correct this ambiguity, we slightly modified the wording of the paper’s title and clarified the description of the analogy in the Discussion.

*Introduction: The presentation in the Introduction suggests that Douady and Couder pioneered the inhibition field model of phyllotaxis. Although they contributed considerably to the field, the authors should perhaps mention at least a few of the others that came before, such as Thornley (1975), Mitchison (1977), Veen and Lindenmayer (1977), Young (1978), Schwabe and Clewer (1984), Chapman and Perry (1987), etc.*

We agree with the reviewer and we have now changed the wording of this passage in the Introduction and added a few key references as suggested.

*Results: The authors write that it has been shown that the central zone is high in auxin. Based on the data, I would say it has been "suggested" or "proposed". Data to support this idea is only from Arabidopsis inflorescense meristem, and from the DII reporter without the recent control construct added by Weijers. The data looks completely different in tomato, where DR5 works just fine in the central zone.*

We agree with the reviewer and we now say that this “has been suggested”.

*Results: The authors report dislocations in other species, and then immediately assume that it is due to variation in the plastochron. One other possibility could be variations in inhibition threshold. In most inhibition models of phyllotaxis these two are tied together, but that is not necessarily the case in planta. In fact, does not their previous work (Besnard et al. 2014) claim that the timing (i.e. plastochron) is controlled somewhat independently? To report that it is a variation in plastochron, analysis of the timing of primordium initiation at the meristem is required.*

Modeling stochasticity of inhibition perception is actually exactly the essence of our approach. An analysis of the effect of variations in inhibition threshold was previously carried out in Mirabet et al. 2012 and concluded that it induced specifically a variation in the plastochron leading to organ co-initiations. However, in this work permutations could be induced although at a much lower frequency than what is observed in planta. Also only simple perturbations (2-permutations) could be obtained. We developed this idea further by modelling the response of the system to the inhibition signal (using a probabilistic framework rather than a deterministic threshold).

Our work also does suggest that there is a relative independence between the emergent plastochron and the spatial patterning system, as reported previously in planta in Besnard et al. 2014. Such an analysis, comparing observed wild type to mutant plastochrons, is done in the appendix (see Appendix 4—figure 1D).

In any case, we agree that an analysis of the timing of primordium initiation at the meristem in other species than Arabidopsis would be important. However, this would require performing live imaging in these various systems, which is at the moment out of reach. For this reason, we have considered permutations as a plausible signature of plastochron variation, based on our previous work in Arabidopsis. We have thus rephrased our text in the following way in order to be more careful in our conclusions and to address the reviewer’s concern:

“As suggested by the results on Arabidopsis, these observations raise the possibility that these organ permutations might result from a noise on the plastochron and that such perturbation could be a common feature of phyllotactic systems that occurs in meristems with different geometries.”

*Also it is not clear to me from the side views in Figure 2 if these dislocations are just random positioning, or are permutations as stated in the figure caption and text. You would need an image or diagram from the top, or the angles plotted on a graph in order to see if this is the case.*

To clarify the 3D geometric interpretation of the 2D pictures, we added as suggested at the bottom of each image a small diagram showing how the branch angles and orders should be read from the top.

*Results: The authors state that the inhibition profile can be seen as an energy. In what sense is it an energy? I think it would be less confusing for the reader, and more accurate to refer to it as the inhibition profile.*

It is common in the literature about phyllotaxis to refer the inhibition created by each organ in its surrounding as an abstract repulsive energy by analogy with physical systems. This term was notably used in the work of Douady and Couder (1996). We added a couple of sentences to make this definition clearer.

*Property 2 is reported in the caption for Figure 5 in Smith et al. (2006, CJB), you should probably reference that work in this context.*

The reviewer is right that several works before have mentioned that in a stationary regime the position of next organ to be initiated is separated from the current initium by roughly a divergence angle (Smith et al. 2006 mention this in one figure legend as well as Mirabet et al. 2011, Figure 1 for instance). However, what we say here based on our study of the dynamics of inhibitory fields is more general: at any moment, *all* the energy minima are separated by multiples of the divergence angle. This can be observed for example in Figure 3 and in corresponding Video 1–Video 3.

*In a simulation experiment, they perturb the model by initiating primordia at a local inhibition minimum that is not the global minimum and observe recovery. They need the control experiment here. What happens when they initiate a primordia at a random location? Does not the pattern also recover (as it does in ablation experiments)? How does this compare?*

We agree with the reviewer that such a control would be informative. We have performed the control experiments suggested by this comment: interrupting an on-going simulation, we have forced the initiation of a primordium at various random locations, not only local minima. These types of perturbations amount to exploring random initial conditions. In the bifurcation diagram of the standard model (cf. Appendix, Figure 2), different initial conditions can lead to different phyllotactic patterns (the branches of the bifurcation diagram), each pattern having a basin of attraction composed of those initial conditions that converge to the corresponding phyllotactic pattern. From this general principle, we expect that forcing a primordium at an arbitrary position may lead to a permanent change of phyllotaxis (if the artificially created initial condition lies outside the basin of attraction of the current pattern). This expected behavior is exactly what we observed and in fact we were able to classify the initial conditions that typically lead to a change of phyllotactic mode and those that did not (including the previously treated case of a primordium introduced at the second local minimum), as reported in Appendix, Section 6.7 and Appendix Figure 5 and we mentioned this control experiment in the main text.

*Also the authors argue that noise in divergence angle order does not propagate far, but that it has a long effect on the timing, but they don't really give any proof this. We would need to see the time, divergence angle pairs produced by the model, and the model would have to have a similar discretization in time as in space (in most simulation models space it much more coarsely discretized than time). Finally, I might be convinced if they could show evidence of this in planta. It seems to me that after ablation experiments the timing recovers just as fast as the divergence angle.*

We are unsure of what is the concern of the reviewer here. We have not studied the effect of noise on divergence angle but the effect of introducing artificially noise on the timing through forcing permutations in the order of organ initiation. We then observe that temporal noise does not affect significantly the angular position of the other primordia but has a long term effect on the timing in the classical model. This was observed robustly on many repetitions. In this, we are not trying to demonstrate that noise on spatial position does not propagate but only to show that spatial positioning and timing are partly uncoupled in this system and that this drove us to develop a model of signal perception to explain the observed permutations. In addition, information concerning the divergence angles and the associated plastochron produced by the stochastic model are available in Figure 4, H.

Concerning the experimental evidence, we would like to point out that our previous work published in Besnard et al. 2014 (Figure 2) shows in planta using live imaging of wild-type meristems that angle specification is extremely stable over time despite a high plastochron variability with almost 30% of organ co-initiated. These observations are indeed fully consistent with the results of our simulations, showing a partial uncoupling of organ positioning and timing of initiation. To make this point absolutely clear we have indicated this at the end of the corresponding Results section.

*Stochastic model section. Again they are assuming there is no separate control of the plastochron. I wonder if their stochastic model leads to a different outcome than just adding noise to the system? An example would be the Smith et al. (2006, CJB) model, where noise is added to the peripheral circle sampling point positions, which would also have the effect of adding noise to the inhibition threshold. Don't get me wrong, I appreciate a proper stochastic model, but the worth of such models becomes apparent when they demonstrate different behavior than a simple deterministic model + noise. Is this really the case here?*

We assume that when mentioning “separate control of the plastochron” the referee has in mind again the mechanisms involving AHP6 (Besnard et al., 2014). We would like to point out that we didn’t say that AHP6 controls the plastochron but that it controls the robustness of plastochron. From our work and previous theoretical/experimental work on phyllotaxis, we rather believe that plastochron is an emergent property of the system, while having different properties compared to the divergence angles. We propose that what is regulated is the different parameters of the patterning (not directly in the form of a direct regulation of the plastochron) and consequently that all the observation variables (i.e. spatial positioning and plastochron but also perturbations) are linked through the 2 control parameters.

Concerning the added value of our model, Mirabet et al. did exactly what the reviewer suggests here and tried the option to explain permutation with a simple deterministic model + noise. Interestingly, the model is able to capture some features of the observed perturbation (as suggested by the reviewer) but was limited in the capacity to recapitulate satisfactorily the various experimental observations as discussed above. We show here that our model provides a much more plausible interpretation of the observed phenomena.

*Speaking of the Smith et al. model, another major source of noise in this system is the fact that the resolution of the possible locations for primordia to initiate is very low. Fate change is a cellular phenomena, and there are really not very many cells around the peripheral zone, maybe 30-40 cells. This gives a resolution of around 10 degrees, a considerable source of noise. Given this it would seem that a perturbation of the possible initiation locations would be very relevant.*

As pointed out above, a noise on the spatial resolution has only a limited effect on the dynamics of the system, supporting the idea that it unlikely explains the frequent perturbations of phyllotaxis observed in *Arabidopsis*. Also note that this paper studies the plausibility of a particular source for the observed phyllotactic perturbations but does not aim to claim that other sources should completely be excluded.

*Materials and methods: The common phyllotaxis angles are irrational, and have the property that new primordia will be placed far from existing ones. It is therefore not surprising that if you include higher permutations, you start to obtain a roughly uniform sampling of the entire meristem. Consideration of 5 permutations gives 6 matching points, 52.5, 105, 137.5, 190, 275, 327.5 so random angles will always be close to one of these locations, which could explain the data in Figure 7 would need to see statistics on this to demonstrate that these angle matches are really closer than random.*

We completely agree with the reviewer regarding the fact that multiples of an angle like the golden angle 137.5 will naturally tend to cover the full interval [0,360) densely and indeed match quite well any distribution of randomly drawn angles. However, this viewpoint does not take into account the temporal succession of angles, which is a crucial component of our fit: the data shown in Appendix Figure 7 is not only a fit of individual angles, but also of their sequence. In other words, the fit does not simply consist in finding the “closest match” for each individual angle but the closest match for the whole sequence *amongst* sequences that can be decomposed into permuted blocks of bounded length *n* (called *n-admissible sequences* in our previous publications). In particular, it is likely that a pure distance criterion would yield a quantitatively closer fit using only the closest match of individual angles, but given the variability of angle values observed experimentally (see for instance the spread of the fitted Gaussian-like distributions shown in Figure 1), the sequences we propose are *equally plausible*: by design, our algorithms only select angles which are within the center part of Gaussian distributions fitting angle distributions. Given that *n-admissible* sequences have a very low probability of being obtained from a random draw (when *n* is smaller than the length of the whole sequence) we claim that it is striking to find plausible matches with a high frequency as we did.

In summary, we believe that the answer to this comment was already present in our manuscript, although we apologize that it was not explained with enough clarity. Additional explanations are now provided in the caption for Appendix Figure 7.

*Reviewer #2:*

*I would like to point to three topics that could be addressed by the authors:*

*1) In Introduction and Abstract, the authors state that developmental disorders are generally not considered as informative. I would not point it so strongly, since in fact studies on mutant phenotypes are often studies on disturbed patterns that lead us to conclusions on pattern generation mechanisms. Also some old literature was devoted to teratology, i.e. disturbed development, in order to speculate on developmental mechanisms.*

We agree with the reviewer and reformulated the corresponding passages in a weaker form.

*2) From my knowledge on empirical and theoretical papers on phyllotaxis one of the discrepancies between the two is in that in nature, so-called fused or double primordia (or leaves) can be sometimes observed, while they are most often not generated by models. Would it be possible to generate such primordia in the stochastic model (decreasing the epsilon value)? This could provide one more example of pattern perturbation in support of the model.*

This is an interesting suggestion. To implement it, we have considered the microscopic, most computationally expensive version of our model, SMPmicro, where the stochastic process of primordium initiation takes place in parallel at each discretization point on the central zone boundary (as opposed to processes only at local inhibition minima, in SMPmacro-max, which effectively precludes fused primordia). In this model, with low values of the sensitivity parameter E*, we indeed observed simultaneous formations of two primordia near the same local minimum. Examples of such simulations are now included in the form of animations (revision_multipleInitiation_slow.wmv and revision_multipleInitiation_fast.wmv), see Appendix section 7 for more details.

*3) Most reference to empirical data are done with Arabidopsis, obviously a good choice since some mechanisms on the phyllotaxis regulation are recognized based on the mutation. Nevertheless, the relative size of primordia in Arabidopsis is large, and they are not densely packed around its SAM (low numbers of contact parastichies), while in other species primordia are relatively small and densely packed. I understand that the phyllotaxis parameters used for comparison with model generated phyllotaxis, are not available for these species, but these cases could be to larger extent referred to. For example, shoots with relatively small primordia and short plastochrons (conifer twigs, Asteraceae capitulum) would support the model conclusion on relationships between plastochron duration and frequency of permutations (Results); just have a look on the already dead Christmas tree thicker twigs or trunk, and you will most likely find some permutations in support of the model.*

We agree with the reviewer that it is important to show permutations in species other than *Arabidopsis*. It is for this very reason that we included evidence of permutations in a variety ofspecies that are expected to have different meristem geometries. Optimally we would need to use live-imaging in these species to analyze the dynamics of organ initiation in these species but this is at the moment technically out of reach. Our work is however laying the ground by demonstrating the important of performing such analyses in the future.

*Reviewer #3:*

*My ability to fully interpret this paper is very limited since it is mainly mathematical and my background is biology.*

*[…]*

*If the model used does not have this property I am worried that the conclusions are not relevant to the real plant.*

We agree with Reviewer #3 and it is important that our model be able to generate phyllotactic patterns from a wide range of initial conditions, including randomly chosen ones. To clarify that this is indeed the case, we have performed simulations with initial conditions where the inhibition field is randomly defined, as well as with a naked meristem with no initial primordium. The results are reported in the form of animations, whose details are shown in Appendix, section 7. In all cases, the system robustly self-organizes and converges to a plausible phyllotactic pattern. We thus believe that this provides further support to the idea that our conclusions are relevant to real plant.

Regarding our paper’s general significance, phyllotactic patterning is a prototypical example of a developmental system driven by lateral inhibitions (the inhibitory fields) as pointed out in our manuscript. Lateral inhibitions have been implicated in a large number of patterning processes in both plant and animals (see before). With the ongoing development of live-imaging in a variety of developmental system, we expect that irregularities in such systems will be reported and we believe that we provide a conceptual framework for understanding how such irregularities can be used to understand the molecular mechanisms at work. We also think that this framework will be useful for the analysis of noise in developmental systems in general by providing a vision of how noise propagates through scale to generate specific developmental signatures.

[Editors' note: further revisions were requested prior to acceptance, as described below.]

*The manuscript has been improved but there are some remaining issues that need to be addressed before acceptance, as outlined below:*

*The reviewers and editors agree that you should change the manuscript title to reflect its more narrow proof of concept nature, for example "…from Disorders in the self-organizing pattern of phyllotaxis".*

We have changed the title of the paper as suggested.

*Please fix some of the wording, in particular the use of inhibition vs energy (see comments below), to avoid misunderstandings in the context of the wider literature.*

We have changed the wording as suggested by you and reviewer 1 throughout the main text, figures and appendixes. In addition, we have corrected a few typos in the manuscript.

*Please address the issue of the dynamic definition of inhibitory fields. This appears to be an important limitation of your current model, and although we do not ask you to resolve it at this point, we would like to ask you to explicitly spell out this limitation If it is not captured in the model. That is, specification (or growth of the organ) and the generation of inhibition could occur simultaneously and dynamically rather than in a consecutive, step-wise fashion as is modelled here. Maybe you could also comment on how would this influence the way the system responds to noise?*

This is an important point that was not completely clear to us from the original review comments. We thank Reviewer 3 and editors’ new comments that clarify the concern and we do agree with you that our stochastic system, similarly to the classical deterministic model, does not model the details of the molecular processes involved in the formation of the inhibitory fields and that this may impact its ability to capture self-organizing dynamic aspects of field formation. We have thus added a paragraph in the last section of the Discussion stating clearly this potential limit of our model and how to address it in the future:

“Moreover, similarly to the classical deterministic model of phyllotaxis, our stochastic model does not explicitly account for the cascade of molecular processes that participate to the establishment of new inhibitory fields at the location of incipient primordia. This might limit the ability of these models to fully capture the dynamics of the self-organization of the system. To do so, more mechanistic versions of this stochastic model could be developed in the future, combining more detailed cellular models of hormone-based fields, e.g. (Jönsson et al., 2006; Smith et al., 2006; Stoma et al., 2008), and stochastic perception of these hormonal signals in 2D or 3D models with cell resolution.”